# Inflammasome-mediated GSDMD activation facilitates escape of *Candida albicans* from macrophages

Xionghui Ding[1,2,7], Hiroto Kambara[1,7], Rongxia Guo[1,3,7], Apurva Kanneganti[1], Maikel Acosta-Zaldívar [4], Jiajia Li[3], Fei Liu[3], Ting Bei [1], Wanjun Qi [4], Xuemei Xie[1], Wenli Han[1], Ningning Liu[4], Cunling Zhang[1], Xiaoyu Zhang[1], Hongbo Yu[5,6], Li Zhao[1], Fengxia Ma[3], Julia R. Köhler[4] & Hongbo R. Luo [1✉]

*Candida albicans* is the most common cause of fungal sepsis. Inhibition of inflammasome activity confers resistance to polymicrobial and LPS-induced sepsis; however, inflammasome signaling appears to protect against *C. albicans* infection, so inflammasome inhibitors are not clinically useful for candidiasis. Here we show disruption of GSDMD, a known inflammasome target and key pyroptotic cell death mediator, paradoxically alleviates candidiasis, improving outcomes and survival of *Candida*-infected mice. Mechanistically, *C. albicans* hijacked the canonical inflammasome-GSDMD axis-mediated pyroptosis to promote their escape from macrophages, deploying hyphae and candidalysin, a pore-forming toxin expressed by hyphae. GSDMD inhibition alleviated candidiasis by preventing *C. albicans* escape from macrophages while maintaining inflammasome-dependent but GSDMD-independent IL-1β production for anti-fungal host defenses. This study demonstrates key functions for GSDMD in *Candida*'s escape from host immunity in vitro and in vivo and suggests that GSDMD may be a potential therapeutic target in *C. albicans*-induced sepsis.

[1] Department of Pathology, Dana-Farber/Harvard Cancer Center, Harvard Medical School; Department of Laboratory Medicine, Boston Children's Hospital, Enders Research Building, Room 814, Boston, MA 02115, USA. [2] Department of Burn and Plastic Surgery, Children's Hospital of Chongqing Medical University, National Clinical Research Center for Child Health and Disorders, Ministry of Education Key Laboratory of Child Development and Disorders, Chongqing 400014, China. [3] State Key Laboratory of Experimental Hematology, National Clinical Research Center for Blood Diseases, Institute of Hematology & Blood Diseases Hospital, CAMS Key laboratory for prevention and control of hematological disease treatment related infection, Chinese Academy of Medical Sciences & Peking Union Medical College, 288 Nanjing Road, Tianjin 300020, China. [4] Division of Infectious Diseases, Boston Children's Hospital/Harvard Medical School, Boston, MA 02115, USA. [5] VA Boston Healthcare System, Department of Pathology and Laboratory Medicine, 1400 VFW Parkway West Roxbury, Boston, MA 02132, USA. [6] Department of Pathology, Brigham and Women's Hospital and Harvard Medical School, Boston, MA 02115, USA. [7] These authors contributed equally: Xionghui Ding, Hiroto Kambara, Rongxia Guo. ✉email: hongbo.luo@childrens.harvard.edu

Candida albicans is the most common opportunistic fungal pathogen, and it can cause life-threatening disseminated candidiasis and sepsis in immunocompromised patients. Despite the availability of antifungal agents, there are an estimated 400,000 Candida bloodstream infections each year worldwide, with a mortality rate of up to 50%[1,2]. Sepsis, including fungal sepsis, is generally defined as life-threatening organ dysfunction caused by dysregulated systemic host inflammatory responses to microbial infection[3–6]. Sepsis arises when an overwhelming response to infection leads to tissue and organ injury[4,5,7,8]. Sepsis is a major health threat and the primary cause of morbidity and mortality in critically ill patients with infection or trauma[9–15], and it is characterized by both inflammatory and immunosuppressive responses[4,6,16]. During the early stages of sepsis, microbes or microbe-derived molecules trigger the production and release of pro-inflammatory mediators such as TNF-α, ICAM-1, IL-6, and IL-1, also known as a "cytokine storm". Overwhelming and sustained release of proinflammatory mediators exaggerates systemic inflammation and leads to vascular injury and multiple organ damage in severe cases (septic shock)[16–21]. Patients surviving this phase become increasingly immunosuppressed and may die from secondary infections despite the use of appropriate antibiotics[7,16,22,23]. Sepsis-related deaths can therefore occur at either pathophysiological stage[4,6,16].

While pro-inflammatory cytokines are a necessary component of the host response to invading pathogens, excessive proinflammatory cytokine production can cause exaggerated and deleterious tissue inflammation in sepsis. IL-1β is a prototypic proinflammatory cytokine and key initiator of the host inflammatory response, which itself can trigger systemic inflammation and consequent tissue and organ damage[24,25]. IL-1β mediates a wide repertoire of expression programs involved in secondary inflammation, on the one hand activating monocytes, macrophages, and neutrophils and on the other inducing adaptive Th1 and Th17 cellular responses[24,25]. Caspase-1-mediated canonical and caspase-11-mediated noncanonical inflammasomes directly or indirectly control pro-IL-1β cleavage and the consequent production and secretion of mature IL-1β[26,27]. Mice lacking key inflammasome components such as NLRP3 or caspase-1/11 are more resistant to polymicrobial and LPS-induced sepsis than wild-type mice and are less likely to die from lethal endotoxemia[28–36].

Paradoxically, inflammasome signaling appears to be protective in C. albicans infection despite Candida-induced sepsis being associated with excessive cytokine production and organ dysfunction[37]. The pathophysiology of Candida sepsis is therefore significantly different from bacterial sepsis. In contrast to bacterial sepsis, cytokines released by macrophages, such as IL-1β increase host resistance to C. albicans and alleviate disseminated C. albicans infection[38,39]. In several mouse candidiasis models, including a tail vein systemic candidiasis model and a model of oral infection with C. albicans, mice lacking key inflammasome components are hypersusceptible to Candida infection[40–46]. Therefore, inhibition of inflammasome-mediated cytokine production has been dismissed as a strategy to prevent C. albicans-induced sepsis.

Gasdermin D (GSDMD) was recently identified as a key factor responsible for the inflammatory form of lytic cell death (pyroptosis) in macrophages, a critical antibacterial innate immune defense mechanism[47–49]. GSDMD is inactive in its steady state due to autoinhibition by the C-terminal domain[47,49,50], with activation dependent on cleavage of the full-length protein to generate the N-terminal active fragment (GSDMD-NT), which has a poreforming function through oligomerization at the plasma membrane[47,48,51,52]. GSDMD-NT-mediated plasma membrane perforation triggers membrane rupture and subsequent lytic death, leading to the release of pro-inflammatory cytokines such as IL-1β.

Since GSDMD is a direct inflammasome target for execution of this pathway, as observed in inflammasome-deficient mice, Gsdmd disruption also inhibits bacterial sepsis in which cytokines facilitate septic shock[47–49,53–55]. However, the physiological function of GSDMD in response to fungal pathogens and fungal-induced sepsis remains elusive.

Here we show that Gsdmd disruption unexpectedly protects against C. albicans-induced sepsis, contrary to observations in inflammasome-deficient mice. Casp1/11 disruption completely abolished, while Gsdmd disruption only reduced, IL-1β production, which is essential for anti-Candida host defenses, providing an explanation for the discrepancy between Gsdmd and Cap1/11 knockout mice. Moreover, we discover a GSDMD-mediated mechanism, involving hyphae and candidalysin that enables C. albicans to escape from macrophages. Finally, we demonstrate that the same protective effect can be achieved by applying a GSDMD inhibitor, NSA, to C. albicans-infected hosts. Thus, GSDMD is critical in impeding clearance of C. albicans and is a potential therapeutic target in Candida sepsis.

## Results

**Gsdmd disruption alleviates, while Casp1/11 disruption aggravates, C. albicans infection.** IL-1β production is required for an effective host response to Candida infection, and mice lacking critical inflammasome components such as NLRP3, ASC, or caspase-1 are hypersusceptible to C. albicans-induced sepsis[38–46]. In agreement with these studies, in a mouse model of systemic candidiasis, Casp1/11[-/-] mice (Fig. S1a) were more susceptible to C. albicans infection, displaying higher death rates (Fig. S1b), worse clinical scores (Fig. S1c), more severe organ damage (Fig. S1d), and increased fungal burden (Fig. S1e, f) compared to wild-type (WT) mice.

Infection with C. albicans triggered robust cleavage of full-length GSDMD and generated the active N-terminal fragment in mouse bone marrow-derived macrophages (BMDMs), indicating direct activation of GSDMD by C. albicans (Fig. 1a). GSDMD cleavage occurred as early as 4 h after the infection and continued to increase thereafter (Fig. S1g). We next investigated the host response to intravenously injected C. albicans in Gsdmd-deficient mice. Surprisingly, as a well-known downstream target of caspase-1/11, GSDMD disruption led to the completely opposite phenotype to Casp1/11 disruption in C. albicans-challenged mice. Gsdmd deletion, without affecting the cleavage and activation of caspase 1 (Fig. 1a), conferred partial protection from C. albicans-induced sepsis and disease, as shown by increased survival (Fig. 1b), alleviation of infection-induced body weight loss (Fig. 1c), and improved clinical outcomes (Fig. 1d) in Gsdmd[-/-] mice. WT mice infected with C. albicans also developed larger kidneys than Gsdmd[-/-] mice (Fig. 1e–g), with histopathological examination revealing large areas of necrosis and hemorrhage in C. albicans-infected WT kidneys (Fig. 1h). By contrast, kidneys from C. albicans-infected Gsdmd[-/-] mice showed significantly less necrosis and less severe tissue damage (Fig. 1h). Mice with hematogenous disseminated candidiasis die from progressive sepsis, and kidney fungal burden is correlated with the severity of renal failure and systemic acidosis, hallmarks of severe sepsis. We therefore investigated the fungal loads in several organs, including the kidneys. As expected, C. albicans organisms were mainly lodged in the kidneys. In line with resistance to disseminated candidiasis, Gsdmd[-/-] mice displayed a ten-fold decrease in kidney fungal burden as calculated by the total number of colony-forming units (CFUs) in kidneys on day 2 compared to WT mice (Fig. 1i). Thus, the alleviated C. albicans infection in Gsdmd[-/-] mice appears to be at least partially due to better clearance of C. albicans by the host defense system. Consistently, methenamine silver staining

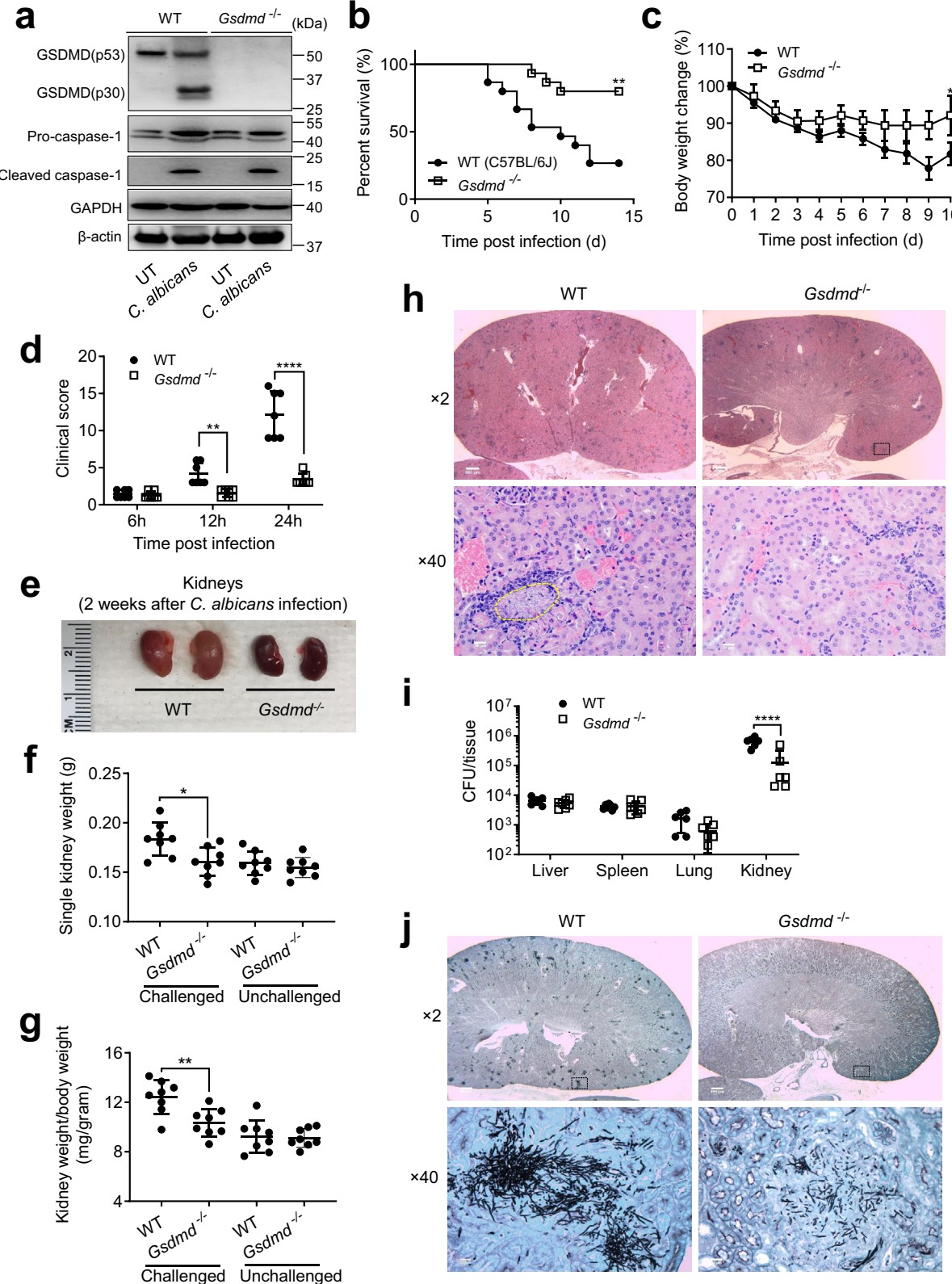

revealed large numbers of fungal hyphae growing in necrotic areas in the collecting ducts of kidneys from *C. albicans*-infected WT mice, while the number of hyphae in the kidneys was markedly decreased in *Gsdmd*<sup>-/-</sup> mice (Fig. 1j). Collectively, these results demonstrate that GSDMD plays a key role in *C. albicans* infection and may represent a potential therapeutic target for *C. albicans*-induced sepsis.

***Casp1/11* disruption completely abolishes, while *Gsdmd* disruption only reduces, IL-1β production.** We next explored the mechanism responsible for the discrepant outcomes of candidiasis in *Casp1/11*- and *Gsdmd*-deficient mice. IL-1β plays an important role in anti-*C. albicans* host defenses. The impaired host response to invading *C. albicans* in *Casp1/11*<sup>-/-</sup> mice was thought to be due to their inability to produce IL-1β[38–46]. Due to the critical roles of

**Fig. 1 Gsdmd disruption alleviates C. albicans infection and associated kidney damage. a** *C. albicans*-triggered GSDMD cleavage in mouse bone marrow-derived macrophages (BMDMs). BMDMs from WT (C57BL/6 J) or *Gsdmd*[-/-] (C57BL/6 J) mice were untreated or infected with WT *C. albicans* for 6 h at MOI 50. Full-length (p53) GSDMD, cleaved (p30) GSDMD, pro-caspase-1, and cleaved caspase-1 in the cell lysates were detected by western blotting. The figure shows the result of a representative experiment that was repeated three times. Source data are provided as a Source Data file. **b** Kaplan-Meier survival plots of *C. albicans*-challenged WT ($n = 15$) and *Gsdmd*[-/-] ($n = 15$) mice. Age- and sex-matched (10-week-old female) WT and *Gsdmd*[-/-] mice were intravenously challenged with $2 \times 10^5$ CFU *C. albicans* and monitored for 15 days. Survival rates were analyzed using Kaplan-Meier survival curves and log-rank testing ($p = 0.0031$ *vs.* WT). **c** Comparison of body weight changes of *C. albicans*-challenged WT and *Gsdmd*[-/-] mice. Two-tailed unpaired Student's t test was used for statistical comparisons. Data presented are the means ± SD ($n = 9$ for WT, $n = 4$ for *Gsdmd*[-/-]) of 3 independent experiments. $p = 0.0443$ *vs.* WT at day 10. **d** Clinical score of *C. albicans*-challenged WT and *Gsdmd*[-/-] mice. Data presented are the means ± SD ($n = 7$ mice per data point) of three independent experiments. Comparison was made by two-way ANOVA followed by Sidak' multiple comparisons test. $p = 0.0443$ (12 h), $p < 0.0001$ (24 h) vs. WT. **e** Comparison of the kidneys of *C. albicans*-challenged WT and *Gsdmd*[-/-] mice. Mice were sacrificed 2 weeks after the *C. albicans* infection. Shown are representative images from three independent experiments. **f** Kidney weights of *C. albicans*-challenged WT and *Gsdmd*[-/-] mice. Two-tailed unpaired Student's *t* test was used for statistical comparisons. Data presented are the means ± SD ($n = 8$ kidneys per group) of three independent experiments. $p = 0.012$ (Challenged) vs. WT. **g** Kidney weights are shown as the weight per gram body weight of each mouse. Two-tailed unpaired Student's *t* test was used for statistical comparisons. Data presented are the means ± SD ($n = 8$ kidneys per group) of three independent experiments. $p = 0.0049$ (Challenged) vs. WT. **h** Histopathologic assessment of the kidneys of WT and *Gsdmd*[-/-] mice 2 days after intravenous injection of $1 \times 10^6$ CFU *C. albicans*. Shown are representative H&E-stained sections of kidney tissues from three independent experiments. **i** Fungal burden in the kidneys, livers, spleens, and lungs of WT and *Gsdmd*[-/-] mice. Mice were sacrificed 2 days after intravenous injection of $1 \times 10^6$ CFU *C. albicans*. Whole organs were homogenized, and live *C. albicans* in the indicated organs were quantified as CFU per tissue. Data presented are the means ± SD ($n = 7$ mice per group) of three independent experiments. Comparison was made by two-way ANOVA followed by Sidak' multiple comparisons test. $p < 0.0001$ (kidney) *vs.* WT. **j** *C. albicans* in the kidneys were identified by Grocott methenamine silver staining. Mice were sacrificed 2 days after intravenous injection of $1 \times 10^6$ CFU *C. albicans*. The figure shows the result of a representative experiment that was repeated three times.

caspase-1/11 and GSDMD in regulating IL-1β processing and secretion in macrophages, we next measured *C. albicans*-induced IL-1β production in macrophages. Mature IL-1β was not detectable by western blotting in the cell lysates or supernatants of *Casp1/11*[-/-] BMDM cultures after *C. albicans* infection, confirming that *C. albicans*-triggered IL-1β processing was completely caspase-1/11-dependent (Fig. 2a); similar results were observed by ELISA (Fig. 2b). In contrast, *C. albicans*-induced IL-1β release was only partially inhibited in *Gsdmd*[-/-] BMDM cultures (Fig. 2c, d), with more mature IL-1β sequestered in the cytosol compared to WT macrophages (Fig. 2c and e), suggesting that both GSDMD-dependent pyroptosis and GSDMD-independent mechanisms are involved in mature IL-1β secretion following caspase-1/11-mediated pro-IL-1β processing. *C. albicans*-induced pro-IL-1β processing was completely independent of GSDMD; the total mature IL-1β (intracellular and extracellular) was unaltered in *Gsdmd*[-/-] BMDM cultures (Fig. 2e). Consistent with the ex vivo results, *C. albicans*-induced IL-1β production was completely inhibited in *Casp1/11*[-/-] mice (Fig. 2f), while *C. albicans*-challenged *Gsdmd*[-/-] mice produced reduced but still significant amounts of IL-1β (Fig. 2g). At 24 h after *C. albicans* infection, the serum level of IL-1β in *Gsdmd*[-/-] mice was still >100 pg/ml compared to <10 pg/ml in *C. albicans*-challenged *Casp1/11*[-/-] mice or unchallenged WT mice. Collectively, these results support a model in which pro-IL-1β processing is absolutely caspase-1/11 dependent, while the secretion of processed mature IL-1β is mediated by both GSDMD-dependent and -independent mechanisms. Thus, *Casp1/11* disruption completely abolishes, while *Gsdmd* disruption only partially reduces, IL-1β production, which is essential for anti-*Candida* host defenses, providing an explanation for the discrepancy between the *Casp1/11*- and *Gsdmd*-deficient mice. Supporting this hypothesis, in the presence of anakinra, a commonly used interleukin 1 receptor antagonist, the protective effect triggered by GSDMD disruption was abolished (Fig. 2h–j and Fig. S2). Anakinra-treated WT and GSDMD-deficient mice both displayed more severe organ damage and reduced survival after *C. albicans* infection compared to untreated GSDMD-deficient mice.

**GSDMD facilitates *C. albicans* escape from macrophages.** *Gsdmd* disruption not only failed to impair *C. albicans* clearance

but also promoted host defenses against *C. albicans* infection (Fig. 1). We next explored the mechanism by which *Gsdmd* disruption alleviated *C. albicans* infection. GSDMD is a key mediator of pyroptosis, a form of lytic cell death. *C. albicans*-induced macrophage lytic cell death is an important cellular event for both the fungal pathogen and the host. *C. albicans* uses this mechanism to escape from macrophages and disseminate[45,46], while macrophages use this mechanism to secrete IL-1β to mount innate and adaptive anti-fungal immune responses[38,39]. In an in vitro phagocytosis assay, unicellular yeast-form *C. albicans* were efficiently engulfed by both the WT and *Gsdmd*[-/-] BMDMs (Fig. 3a). The number of yeast particles engulfed by 100 BMDMs (phagocytosis index), the percentage of BMDMs engulfing at least one yeast particle (phagocytosis efficiency), and the percentage of *C. albicans* engulfed by BMDMs were quantified, and there were no differences between WT and *Gsdmd*[-/-] BMDMs (Fig. 3b). To maximize the efficiency of phagocytosis[56–60] and most importantly to mimic the serum-containing physiological environment, we used serum-opsonised *C. albicans* in this study (Fig. S3a, b). Under this condition, the phagocytosis of *C. albicans* by BMDMs was extremely efficient, with most of the opsonized *C. albicans* engulfed within 1 h (Fig. 3b).

To further investigate *C. albicans*-macrophage interactions, we employed time-lapse live imaging to observe the behavior of *C. albicans*-infected BMDMs (Fig. 3c and Supplementary Movie 1). Engulfed *C. albicans* quickly grew hyphae inside cells, breaking the plasma membrane to escape from BMDMs. The average number of hyphae escaping from *Gsdmd*[-/-] BMDMs was significantly lower than that from WT BMDMs (Fig. 3d). We used propidium iodide (PI), a membrane-impermeable dye, to detect the loss of plasma membrane integrity caused by lytic cell death. The percentage of PI[+] cells increased gradually after *C. albicans* infection, with *Gsdmd*[-/-] BMDMs displaying significantly reduced levels at each time point after 3 h (Fig. 3e). By assessing macrophage lytic death through lactate dehydrogenase (LDH) release, *C. albicans* infection induced robust LDH release from WT BMDMs over time. LDH release was suppressed in *Gsdmd*[-/-] BMDMs, with a 50% reduction detected 12 h after *C. albicans* infection, again demonstrating that *C. albicans*-induced macrophage death was partially blocked in *Gsdmd*-deficient BMDMs (Fig. 3f).

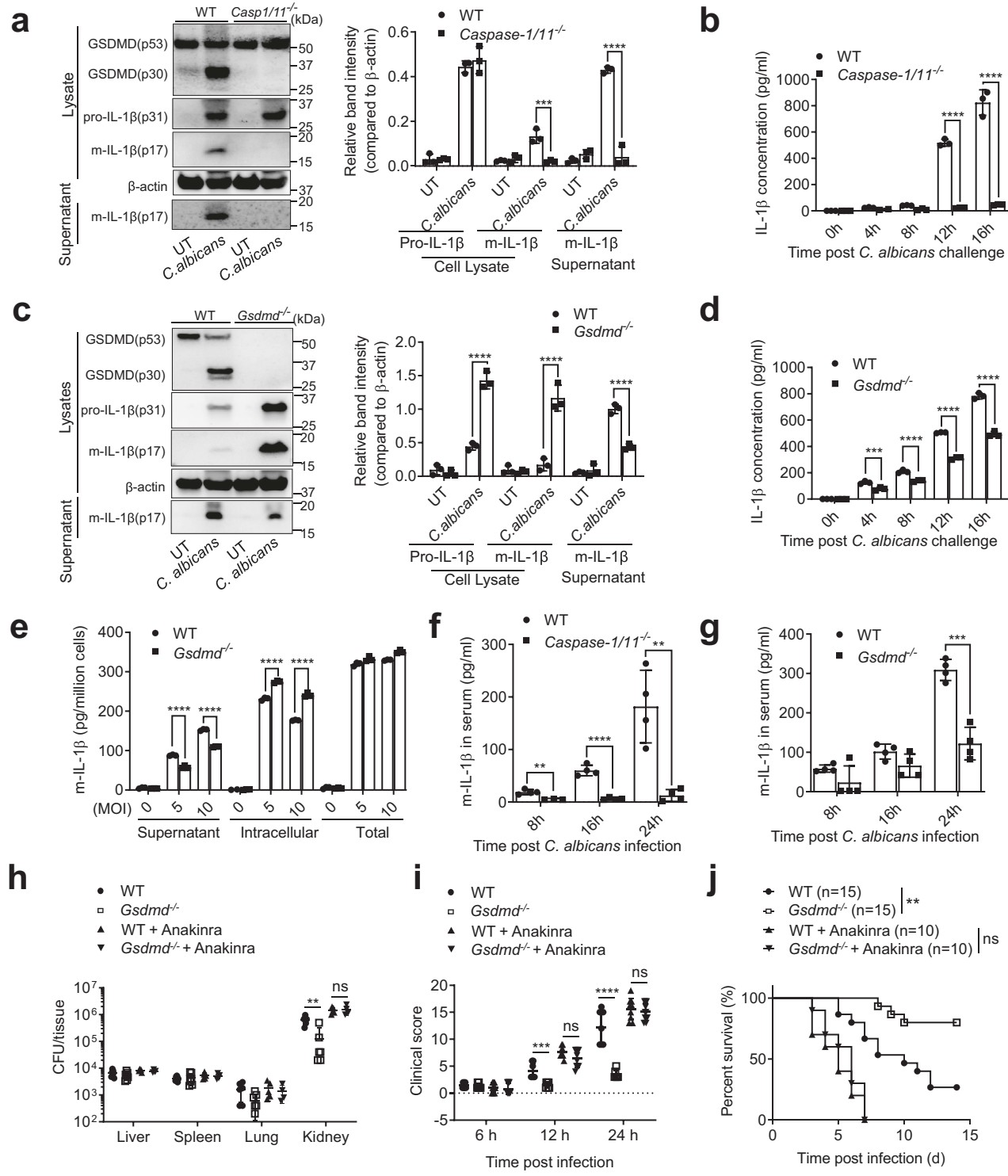

Finally, we measured the effect of macrophages on the overall growth and survival of *C. albicans*. First, GFP-expressing *C. albicans* were grown in the presence or absence of macrophages, and the *C. albicans* mass was calculated from the fluorescence intensity of each culture. Compared to macrophage-free cultures, *C. albicans* mass was reduced by 25% in the presence of WT macrophages but by >50% in the presence of *Gsdmd*⁻/⁻ macrophages, indicating that *Gsdmd* disruption elevated the *C. albicans* clearing capability of macrophages (Fig. 3g). Similar results were obtained when *C. albicans* growth and survival were assessed by measuring the number of colony-forming units

(CFUs) in culture (Fig. 3h). In summary, *Gsdmd* disruption promoted macrophage survival and attenuated the escape of *C. albicans* from macrophages, thereby enhancing the *C. albicans* clearing capability of macrophages.

Of note, *Gsdmd*⁻/⁻ macrophages still displayed *C. albicans*-induced lytic death (Fig. 3d–f). The fact that significant macrophage death still occurred in the absence of GSDMD suggests that *C. albicans*-induced macrophage death could be mediated by both GSDMD-dependent pyroptosis and GSDMD-independent mechanisms. Thus, significant amounts of IL-1β can still be released in *Gsdmd*⁻/⁻ mice (Fig. 2) via

**Fig. 2 Gsdmd disruption reduces, while Casp-1/11 disruption completely abolishes, IL-1β secretion. a** *C. albicans*-triggered GSDMD cleavage and IL-1β production in *Casp-1/11⁻/⁻* mice. BMDMs from WT and *Casp-1/11⁻/⁻* mice were infected with *C. albicans* for 6 h at MOI 50. GSDMD (cleaved p30 and full-length p53), pro-IL-1β, and mature IL-1β (m-IL-1β) in the cell lysates and m-IL-1β in the supernatants were assessed by western blotting. Results are representative of data from at least three biological replicates. The intensities of protein bands were quantified using ImageJ. Comparison was made by two-way ANOVA followed by Sidak' multiple comparisons test. Data shown are means ± SD ($n = 3$ biologically independent samples). $p = 0.003$ (cell lysate-m-IL-1β), $p < 0.0001$ (supernatant-m-IL-1β) *vs*. WT. Source data are provided as a Source Data file. **b** IL-1β secretion by BMDMs from WT and *Casp-1/11⁻/⁻* mice. BMDMs were infected with *C. albicans* (MOI = 10) for the indicated time periods. IL-1β in culture supernatants was measured by ELISA. Comparison was made by two-way ANOVA followed by Sidak' multiple comparisons test. Data shown are means ± SD ($n = 3$ mice per group). $p < 0.0001$ (12 h, 16 h) *vs*. WT. **c** *C. albicans*-triggered IL-1β production in WT and *Gsdmd⁻/⁻* mice assessed by western blotting. Comparison was made by two-way ANOVA followed by Sidak' multiple comparisons test. Data shown are means ± SD ($n = 3$). $p < 0.0001$ vs. WT. Source data are provided as a Source Data file. **d** IL-1β secretion by BMDMs from WT and *Gsdmd⁻/⁻* mice measured by ELISA. Comparison was made by two-way ANOVA followed by Sidak' multiple comparisons test. Data shown are means ± SD ($n = 3$ mice per group). $p = 0.0003$ (4 h), $p < 0.0001$ (8 h, 12 h, 16 h) *vs*. WT. **e** *Gsdmd* disruption did not alter the total amount of m-IL-1β produced by *C. albicans*-challenged BMDMs. BMDMs from WT and *Gsdmd⁻/⁻* mice were infected with *C. albicans* for 12 h at the indicated MOIs. The amounts of m-IL-1β in culture supernatants and cell lysates (intracellular) were measured by ELISA. Comparison was made by two-way ANOVA followed by Sidak' multiple comparisons test. Data shown are means ± SD ($n = 3$ mice per group). $p < 0.0001$ *vs*. WT. **f** *C. albicans*-induced IL-1β production in *Casp-1/11⁻/⁻* mice. WT and *Casp-1/11⁻/⁻* mice were intravenously challenged with $1 \times 10^6$ CFU *C. albicans*, and the level of IL-1β in the serum was assessed at the indicated time points. Two-tailed unpaired Student's t test was used for statistical comparisons. Data shown are means ± SD ($n = 4$ mice per group). $p = 0.0022$ (8 h), $p < 0.0001$ (16 h), $p = 0.0029$ (24 h) *vs*. WT. **g** *C. albicans*-induced IL-1β production in *Gsdmd⁻/⁻* mice. WT and *Gsdmd⁻/⁻* mice were intravenously challenged with $1 \times 10^6$ CFU *C. albicans*. Two-tailed unpaired Student's *t* test was used for statistical comparisons. Data shown are means ± SD ($n = 4$ mice per group). $p = 0.1542$ (8 h), $p = 0.0881$ (16 h), $p = 0.0003$ (24 h) *vs*. WT. **h** Fungal burden in the kidneys, livers, spleens, and lungs of untreated or anakinra-treated WT and *Gsdmd⁻/⁻* mice. Mice were sacrificed 2 days after intravenous injection of $1 \times 10^6$ CFU *C. albicans*. Two-tailed unpaired Student's t test was used for statistical comparisons. Data presented are the means ± SD ($n = 7$ for Control, $n = 5$ for Anakinra-treated mice) of three independent experiments. $p = 0.0014$ (kidney without anakinra), $p = 0.8061$ (kidney with anakinra) *vs*. WT. **(i)** Clinical score of *C. albicans*-challenged untreated or anakinra-treated WT and *Gsdmd⁻/⁻* mice. Two-tailed unpaired Student's t test was used for statistical comparisons. Data presented are the means ± SD ($n = 7$ for Control, $n = 10$ for Anakinra-treated mice) of three independent experiments. $p = 0.0009$ (12 h without anakinra), $p = 0.0551$ (12 h with anakinra), $p < 0.0001$ (24 h without anakinra), $p = 0.5316$ (24 h with anakinra) *vs*. WT. **j** Kaplan-Meier survival plots of *C. albicans*-challenged untreated or anakinra-treated WT and *Gsdmd⁻/⁻* mice. Mice were intravenously challenged with $2 \times 10^5$ CFU *C. albicans* and monitored for 15 days. Log-rank analysis was used. $p = 0.0031$ (without anakinra), $p = 0.4337$ (with anakinra) *vs*. WT.

GSDMD-independent macrophage lytic death to maintain host responses to *C. albicans*, thereby alleviating infection in these mice. Candida infection also directly triggered TNFα and IL-6 production in macrophages. However, WT and *Gsdmd⁻/⁻* macrophages generated the same amount of TNFα and IL-6 after Candida infection (Fig. S3c), supporting the notion that the expression and secretion of these cytokines were GSDMD-independent. Thus, the elevated antifungal response observed in the GSDMD-deficiency mice was unlikely mediated by overall upregulation of pro-inflammatory cytokine production.

**Hyphae formation is essential for *C. albicans*-induced GSDMD cleavage and macrophage death, facilitating *C. albicans* escape from macrophages.** GSDMD-mediated pyroptosis facilitates *C. albicans* escape from macrophages (Fig. 3). We next explored the mechanism triggering GSDMD cleavage and activation. It has been reported that *C. albicans* must switch from a unicellular yeast form into a filamentous hyphae form for inflammasome activation and macrophage lytic cell death[44,45]. Thus, using a yeast-locked strain (*cph1Δ/Δ efg1Δ/Δ*) defective in filamentous growth and unable to accomplish yeast-to-hyphae or -pseudo-hyphae transition in response to stimuli such as serum or macrophages[61], we investigated the role of hyphae in *C. albicans*-induced GSDMD cleavage and IL-1β processing by western blotting. Yeast-locked *C. albicans* failed to efficiently trigger GSDMD cleavage and IL-1β production by BMDMs (Fig. 4a, b). Consistent with previous findings, pro-IL-1β processing was significantly suppressed in mutant *C. albicans*-challenged BMDMs, suggesting that hyphae formation was critical for inflammasome activation and consequent pro-IL-1β processing (Fig. 4a, b). We therefore measured secreted IL-1β levels in the supernatant by ELISA (Fig. 4c). WT *C. albicans* induced robust IL-1β release from BMDMs, which was mostly hyphae dependent, since IL-1β release was significantly blocked when macrophages were treated with yeast-locked *C. albicans* (Fig. 4c).

Hyphae-dependent IL-1β secretion was suppressed by ~50% in *Gsdmd⁻/⁻* BMDMs, again indicating a critical role for GSDMD in this process (Fig. 4d). However, consistent with the data presented in Fig. 2 and 3, this result also suggested that *C. albicans*-induced IL-1β release might also be mediated by GSDMD-independent mechanisms, since *Gsdmd⁻/⁻* BMDMs could still generate a significant amount (~50%) of IL-1β (Fig. 4d). Time-lapse live imaging revealed that yeast-locked *C. albicans* were phagocytosed as efficiently as WT *C. albicans* (Supplementary Movie 2, Fig. 4e). However, mutant *C. albicans* failed to significantly rupture macrophage membranes, as evidenced by the reduced number of PI⁺ macrophages (Fig. 4e, f). Similar to IL-1β release, *C. albicans*-induced LDH release was largely hyphae dependent (Fig. 4g) and partially mediated by GSDMD (Fig. 4h). Finally, the overall growth and survival of *C. albicans* in the presence of macrophages was measured by CFU counting. Yeast-locked *C. albicans* displayed significantly reduced survival compared to WT *C. albicans* (Fig. 4i). Thus, hyphae formation is essential for *C. albicans*-induced GSDMD cleavage and macrophage death. Yeast-locked *C. albicans* were trapped in intact macrophages, preventing their escape (Fig. 4j).

**GSDMD plays critical roles in both candidalysin-independent and -dependent macrophage death and *C. albicans* escape from macrophages.** GSDMD cleavage and activation in macrophages is mediated by caspase-1/11. *C. albicans*-induced GSDMD cleavage was abolished in *Casp1/11*-deficient macrophages (Fig. 2a). Recent studies have revealed that a hyphae-associated fungal toxin, candidalysin[62,63], can trigger inflammasome activation and caspase-1 activation in macrophages[64,65]. Accordingly, we examined whether candidalysin is required for GSDMD-mediated macrophage death and *C. albicans* escape from macrophages during *C. albicans* infection. Treatment with candidalysin-deficient (*ece1Δ/Δ*) *C. albicans* still triggered GSDMD cleavage in BMDMs but at significantly lower efficiency

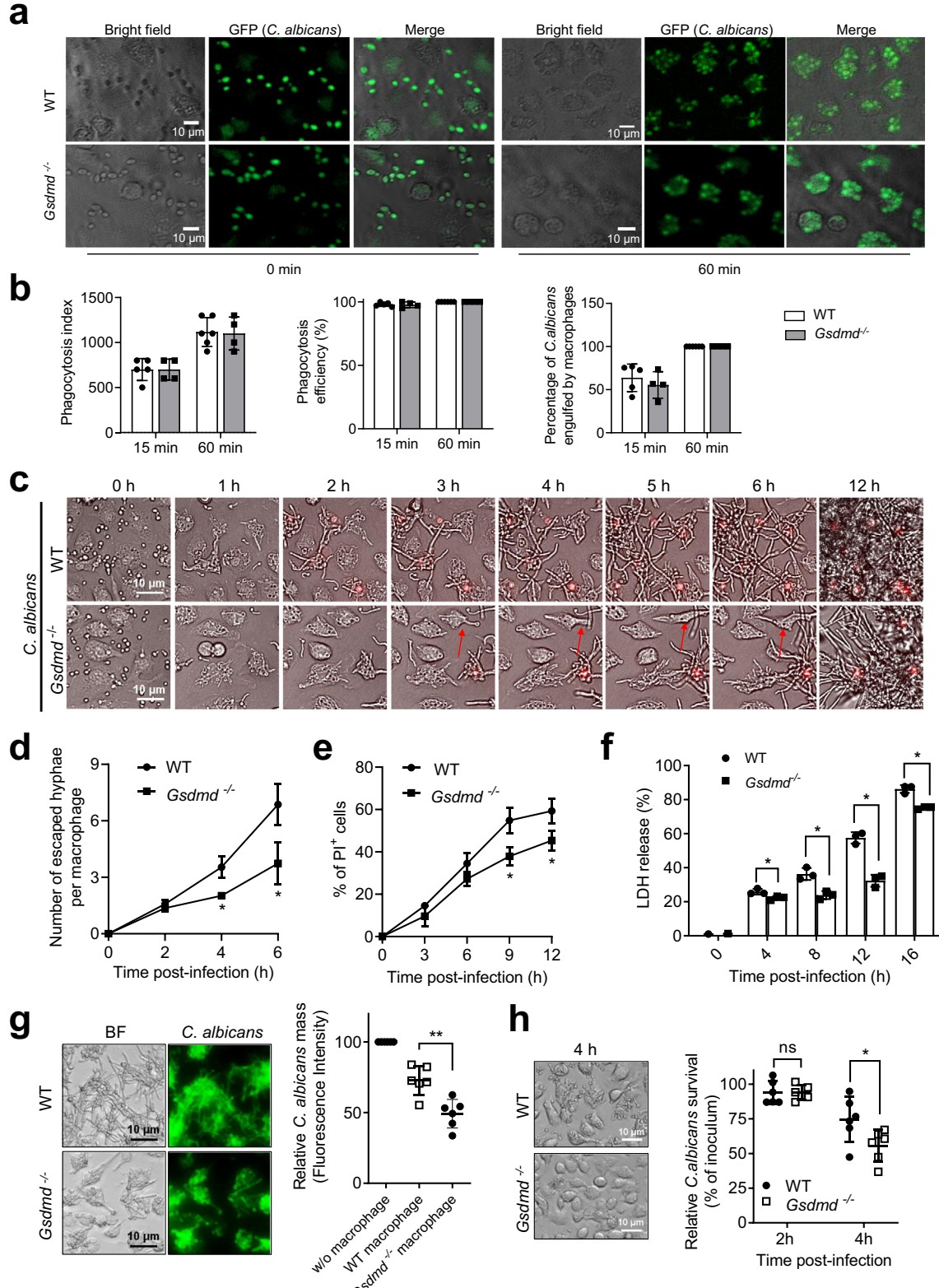

(>50%) than treatment with WT *C. albicans* (Fig. 5a, b). Similarly, IL-1β release was significantly reduced when macrophages were treated with candidalysin-deficient *C. albicans*. However, candidalysin disruption in *C. albicans* did not completely abolish IL-1β release from BMDMs, indicating the presence of candidalysin-independent IL-1β production (Fig. 5c). IL-1β release induced by candidalysin-deficient *C. albicans* was significantly inhibited in

*Gsdmd*[−/−] BMDMs, confirming the essential role of GSDMD in candidalysin-independent IL-1β release (Fig. 5c). Based on the amounts of IL-1β release triggered by WT and *ece1Δ/Δ C. albicans*, we also calculated candidalysin-dependent IL-1β release. On one hand, this process was also suppressed in *Gsdmd*[−/−] BMDMs, demonstrating that GSDMD was important for both the candidalysin-independent and -dependent IL-1β release (Fig. 5d).

**Fig. 3 GSDMD facilitates C. albicans escape from macrophages. a** *Gsdmd* disruption did not affect phagocytosis of *C. albicans* by macrophages. WT and *Gsdmd$^{-/-}$* BMDMs were incubated with GFP-expressing *C. albicans* for 60 min at MOI 10. Shown are the representative images. Scale bars represent 10 μm. **b** Quantification of in vitro phagocytosis capacity of BMDMs. Phagocytosis efficiency was expressed as the percentage of BMDMs that engulfed at least one *C. albicans*. Phagocytosis index was expressed as the average number of internalized *C. albicans* per 100 cells. At least 200 cells were assessed for each sample. Data shown are means ± SD (*n* = 5 for WT 15 min, *n* = 6 for WT 60 min, *n* = 4 for *Gsdmd$^{-/-}$* biologically independent samples per group). **c** Escape of engulfed *C. albicans* from macrophages. WT and *Gsdmd$^{-/-}$* BMDMs were incubated with *C. albicans* at MOI 10. Dying cells were stained with propidium iodide (PI) (red) (200 ng/mL). Images were acquired every 5 min for 12 h using a 20× dry lens. Shown are representative images at the indicated time points. Scale bars represent 20 μm. Experiments were repeated three times. The related time-lapse video is included in the **Supplemental Materials**. **d** The number of escaped hyphae per macrophage. After 6 h, there were too many hyphae to be counted accurately. At least 200 cells were assessed for each sample. Data shown are means ± SD (*n* = 5 biologically independent samples per group). Comparisons were conducted with a two-tailed unpaired student's t test. *p* = 0.0101 (4 h), *p* = 0.0257 (6 h) vs. WT. **e** The percentage of PI-positive BMDMs calculated at each time point. At least 200 cells were assessed for each sample. Data shown are means ± SD (*n* = 5 biologically independent samples per group). Comparisons were conducted with a two-tailed unpaired student's *t* test. *p* = 0.0171 (9 h), *p* = 0.0306 (12 h) *vs.* WT. **f** *C. albicans*-induced BMDM death assessed by a lactate dehydrogenase (LDH) cytotoxicity assay. WT and *Gsdmd$^{-/-}$* BMDMs were incubated with *C. albicans* at MOI 10. Relative LDH release was expressed as the percentage LDH activity in supernatants of cultured cells (medium) compared with total LDH (from the medium and cells) and used as an index of cytotoxicity. All values represent mean ± SD of three independent experiments. Comparisons were conducted with a two-tailed unpaired student's t test. *p* = 0.0371 (4 h), *p* = 0.0070 (8 h), *p* = 0.0008 (12 h), *p* = 0.0014 (16 h) vs. WT. **g** Growth of GFP-expressing *C. albicans* (JKC2078-GFP) in the presence of WT and *Gsdmd*-deficient macrophages. WT and *Gsdmd$^{-/-}$* BMDMs were incubated with GFP-expressing *C. albicans* for 6 h at MOI 5. The relative *C. albicans* mass was measured as the percentage fluorescence intensity of macrophage-containing *C. albicans* cultures compared with *C. albicans* cultures in the absence of macrophages. All values represent mean ± SD of three independent experiments. Comparisons were conducted with a two-tailed unpaired student's t test. *p* = 0.0025 (16 h) *vs.* WT. **h** Growth and survival of *C. albicans* in the presence of WT and *Gsdmd*-deficient macrophages assessed by a colony formation assay. WT and *Gsdmd$^{-/-}$* BMDMs were incubated with *C. albicans* at MOI 2 for the indicated time periods. Relative *C. albicans* survival was measured as the percentage CFU in macrophage-containing *C. albicans* cultures compared with *C. albicans* cultures in the absence of macrophages. All values represent mean ± SD of three independent experiments. Comparisons were conducted with a two-tailed unpaired student's t test. *p* > 0.9999 (2 h), *p* = 0.0424 (4 h) *vs.* WT.

On the other hand, GSDMD disruption did not completely abolish either candidalysin-independent or -dependent IL-1β release, suggesting that both processes could also be mediated by GSDMD and pyroptosis-independent mechanisms.

Consistent with the attenuated IL-1β release observed in *ece1Δ/ΔC. albicans*-treated macrophages, time-lapse live imaging showed significantly lower numbers of escaped *ece1Δ/Δ C. albicans* compared to WT counterparts (Supplementary Movie 3, Fig. 6a, b). Most opsonized WT and mutant *C. albicans* cells were phagocytosed within 1 h, and both WT and *ece1Δ/Δ C. albicans* quickly grew hyphae inside cells. However, candidalysin-deficient *C. albicans* could not penetrate BMDM membranes as efficiently as WT *C. albicans*, as indicated by fewer PI-positive cells being detected in candidalysin-deficient *C. albicans*-treated BMDMs compared to WT *C. albicans*-treated BMDMs (Fig. 6c). Infection with candidalysin-deficient *C. albicans* also induced significantly less LDH release, again suggesting that candidalysin partially contributed to macrophage death (Fig. 6d). Similarly, *C. albicans* escape from macrophages and *C. albicans*-induced macrophage death were mediated by both candidalysin-independent and -dependent mechanisms and both were GSDMD-dependent (Fig. 6c, d).

Collectively, our results show that both candidalysin-independent and -dependent mechanisms could trigger GSDMD activation and mediate *C. albicans* escape from macrophages. As a pore-forming toxin, candidalysin may facilitate *C. albicans*-induced rupture of the plasma membrane and macrophage lytic death, which enable *C. albicans* to escape from host cells (Fig. S4a). Indeed, due to the impaired escape, candidalysin-deficient *C. albicans* grew longer hyphae (>50 μm) inside macrophages (Fig. S4b). In agreement with the in vitro results, in the mouse systemic candidiasis model, mice infected with candidalysin-deficient *C. albicans* displayed higher survival rates (Fig. 7a), better clinical scores (Fig. 7b), less severe kidney enlargement (Fig. 7c), less kidney damage (Fig. 7d), and decreased fungal burden (Fig. 7e, f) than mice infected with WT *C. albicans*. These effects were observed in both the WT and GSDMD-deficient mice (Fig. S4c–i), again suggesting that

candidalysin also plays a role in GSDMD and pyroptosis-independent mechanisms.

The yeast to hyphae switch inside a macrophage can be easily observed in ex vivo Candida infection models. After being engulfed by a macrophage, *C. albicans* switches from yeast to hyphae, pierces the phagolysosome, and comes out of macrophage by killing it[66,67]. Although hyphae formation inside macrophages is considered an important factor for virulence, detecting this process in vivo in Candida-infected host is challenging. It was reported that, in a zebrafish model of *C. albicans* infection, formation of long hyphae inside macrophages is a relatively rare event[68,69]. To assess the role of candidalysin and GSDMD in controlling Candida escape in vivo, we developed an assay (Fig. S5a) in which macrophages from the kidney of Candida-infected mice were isolated and the internalized Candida cells were visualized based on their GFP expression or by staining with a fluorescent-labeled anti-Candida albicans antibody (Fig. S5b). Although *Gsdmd$^{-/-}$* mice displayed a significant decrease in kidney fungal burden as calculated by the total number of CFUs in kidneys (Fig. 1l), the percentage of kidney macrophages containing trapped Candida increased significantly compared to that in WT mice. Candidalysin-deficiency in *C. albicans* led to their augmented retention in macrophages (Fig. S5c–d). These results further confirm that GSDMD and candidalysin play critical roles in *C. albicans* escape from macrophages during Candida infection.

**C. albicans-elicited GSDMD activation is mediated by K$^+$ efflux-dependent canonical inflammasomes.** GSDMD needs to be cleaved by proteases such as caspases and neutrophil elastase (ELANE) to induce lytic cell death and IL-1β secretion. *C. albicans*-induced GSDMD cleavage was attenuated in *Casp1/11$^{-/-}$* compared to WT BMDMs (Fig. 2a). During *C. albicans* infection, *Casp1/11$^{-/-}$* BMDMs consistently released less LDH (Fig. 8a) and showed reduced plasma membrane perturbation (Fig. 8b, c) at each time point examined, further indicating that the caspase-1/11-GSDMD axis plays a critical role in *C. albicans*-induced macrophage pyroptotic death. GSDMD can be cleaved by both

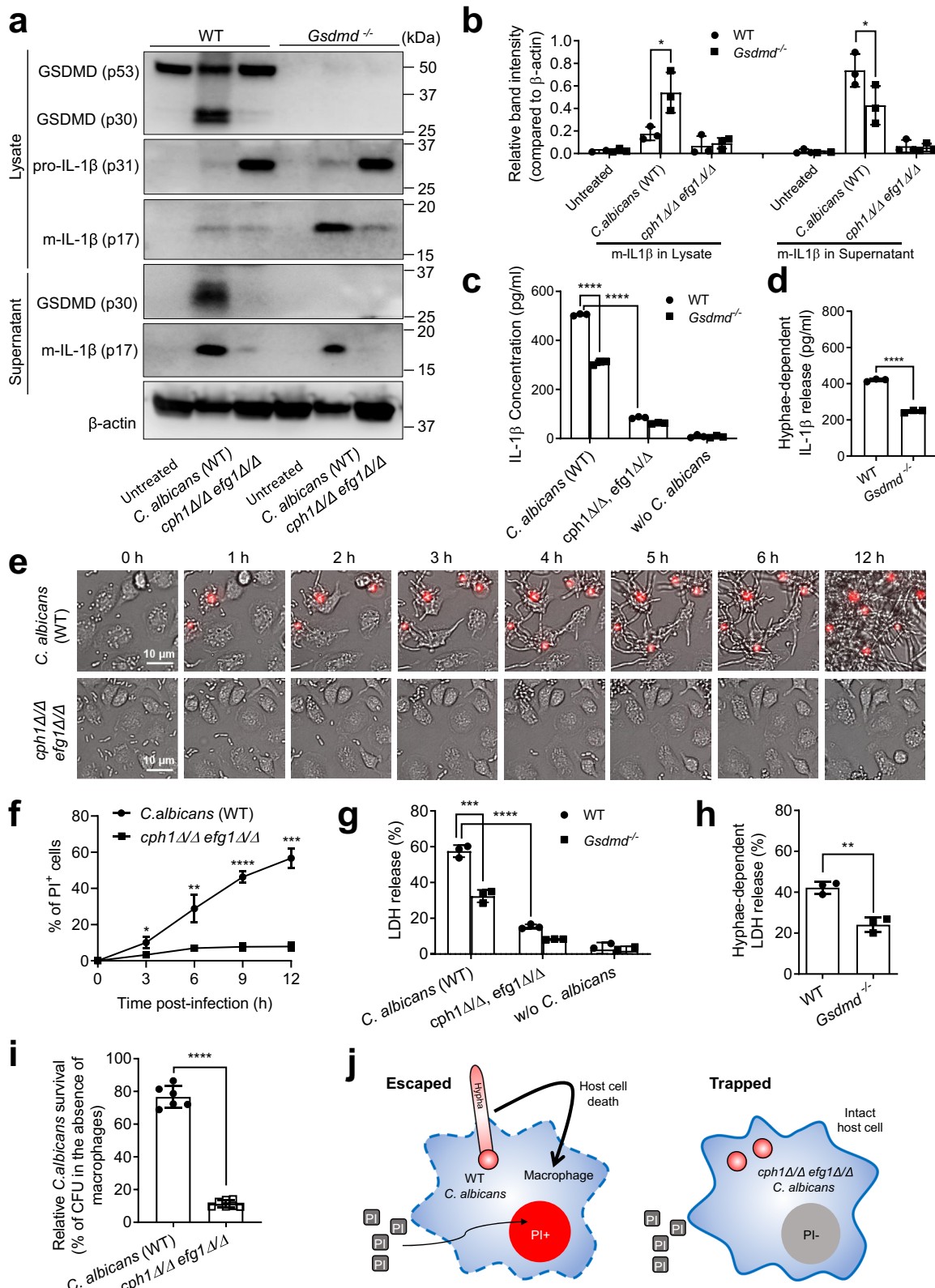

caspase-1-mediated canonical inflammasomes and caspase-11-mediated non-canonical inflammasomes. GSDMD cleavage was efficiently blocked by VX-765, a specific caspase-1/caspase-4 inhibitor (Fig. S6a), suggesting that *C. albicans*-induced GSDMD cleavage and activation were mainly mediated by canonical inflammasomes and caspase-1 (Fig. S6b). The serine protease inhibitor diisopropyl fluorophosphate (DFP) failed to inhibit

*C. albicans*-induced GSDMD cleavage in BMDMs (Fig. S6b). As a result of caspase-1 inhibition, treatment with VX-765 specifically suppressed *C. albicans*-induced production of IL-1β, but not other cytokines such as IL-6, IL-10, MIP2, and TNFα (Fig. S6C). Of note, *C. albicans*-induced macrophage death was only partially blocked in *Casp1/11*[-/-] BMDMs compared to untreated WT BMDMs (Fig. 8a–c and Supplementary Movie 4), suggesting

**Fig. 4 Hyphae formation is critical for *C. albicans*-induced GSDMD cleavage and *C. albicans* escape from macrophages. a** *C. albicans*-triggered GSDMD cleavage and IL-1β production. BMDMs from WT and *Gsdmd*$^{-/-}$ mice were infected with WT or hyphae-deficient (*cph1Δ/Δ, efg1Δ/Δ*) *C. albicans* for 6 h at MOI 50. GSDMD (cleaved p30 and full-length p53), pro-IL-1β, and mature IL-1β (m-IL1β) in the cell lysates and m-IL-1β in the supernatants were assessed by western blotting. Results are representative of data from at least three biological replicates. Source data are provided as a Source Data file. **b** The intensities of protein bands were quantified using ImageJ. Data shown are means ± SD (n = 3 biologically independent samples). Comparisons were conducted with a two-tailed unpaired student's t test. *p* = 0.0288 (m-IL1β in lysate), *p* = 0.0757 (m-IL1β in supernatant) *vs.* WT. **c** IL-1β secretion by BMDMs. BMDMs were infected (untreated BMDMs as controls) with WT or hyphae-deficient (*cph1Δ/Δ, efg1Δ/Δ*) *C. albicans* (MOI = 10) for 12 h. IL-1β in culture supernatants was measured by ELISA. Data shown are means ± SD (n = 3 biologically independent samples per group). Comparisons were conducted with a two-tailed unpaired student's t test. *p* < 0.0001 *vs.* WT. **d** Hyphae-dependent IL-1β release calculated by subtracting the level of IL-1β release induced by hyphae-deficient (*cph1Δ/Δ, efg1Δ/Δ*) mutants from that induced by WT *C. albicans*. Data shown are means ± SD (n = 3 biologically independent samples per group). Comparisons were conducted with a two-tailed unpaired student's t test. *p* < 0.0001 *vs.* WT. **e** Hyphae formation was critical for *C. albicans* escape from macrophages. BMDMs were incubated with WT or hyphae-deficient (*cph1Δ/Δ, efg1Δ/Δ*) *C. albicans* at MOI 10. Dying BMDMs were stained with PI (red). Images were acquired as described in Fig. 3c. Shown are representative images at the indicated time points. The related time-lapse video is included in the **Supplemental Materials**. **(f)** The percentage of PI-positive BMDMs calculated at each time point. At least 200 cells were assessed for each sample. Data shown are means ± SD (n = 3 biologically independent samples per group). Comparisons were conducted with a two-tailed unpaired student's t test. *p* = 0.0199 (3 h), *p* = 0.0079 (6 h), *p* < 0.0001 (9 h), *p* = 0.0001 (12 h) *vs.* WT. **g** *C. albicans*-induced BMDM death assessed by the LDH assay. WT and *Gsdmd*$^{-/-}$ BMDMs were incubated (untreated BMDMs as controls) with WT or hyphae-deficient (*cph1Δ/Δ, efg1Δ/Δ*) *C. albicans* at MOI 10 for 12 h. Relative LDH release was assessed as described in Fig. 3f. Data shown are means ± SD (n = 3 biologically independent samples per group). Comparisons were conducted with a two-tailed unpaired student's t test. *p* = 0.0008 *Gsdmd*$^{-/-}$ (*C. albicans*) *vs.* WT (*C. albicans*). *p* < 0.0001 (*C. albicans*) *vs.* WT (*cph1Δ/Δ, efg1Δ/Δ*). **(h)** Hyphae-dependent LDH release calculated by subtracting the level of LDH release induced by hyphae-deficient (*cph1Δ/Δ, efg1Δ/Δ*) mutants from that induced by WT *C. albicans*. Data shown are means ± SD (n = 3 biologically independent samples per group). Comparisons were conducted with a two-tailed unpaired student's t test. *p* = 0.0025 *vs.* WT. **i** Growth and survival of *C. albicans* assessed by colony formation assays. BMDMs were incubated with WT or hyphae-deficient (*cph1Δ/Δ, efg1Δ/Δ*) *C. albicans* at MOI 10 for 6 h. Relative *C. albicans* survival was measured as described in Fig. 3h. Data shown are means ± SD (n = 6 for *C. albicans*, n = 7 for *cph1Δ/Δ, efg1Δ/Δ*). Comparisons were conducted with a two-tailed unpaired student's t test. *p* < 0.0001 *vs.* WT. **(j)** A schematic diagram illustrating the role of hyphae in mediating macrophage death and facilitating *C. albicans* escape from macrophages.

the existence of inflammasome-independent death-promoting mechanisms.

Potassium (K$^+$) efflux is a major trigger of canonical inflammasome activation. The decrease in intracellular K+, which is likely mediated by TWIK2 potassium efflux channel[70], is an essential trigger for NLRP3 activation induced by ATP and other DAMPs[71–73]. K$^+$ efflux has also been implicated in Candida-induced inflammasome activation and pyroptosis in macrophages[74]. KCl treatment reduces fungal burden, suggesting fungi can use pyroptosis to evade killing by macrophages[74]. Consistently, we also observed that inhibition of K$^+$ efflux by increasing extracellular potassium concentration blocked caspase-1 activation (Fig. S7a, b) and supressed the processing and release of IL-1β, but not TNFα, IL-10, or IL-6 (Fig. S7c). *C. albicans*-induced GSDMD cleavage was completely inhibited in KCl-treated BMDMs, again confirming that GSDMD cleavage is mediated by canonical inflammasome activation (Fig. 8d). *C. albicans*-induced macrophage death is dictated by both candidalysin-independent and -dependent mechanisms (Fig. 6). To investigate the relative contribution of K$^+$ efflux to the two processes, we incubated untreated and KCl-treated BMDMs with WT, hyphae-deficient, and candidalysin-deficient *C. albicans* (Fig. 8e). *C. albicans*-triggered macrophage death, as assessed by LDH release, was significantly inhibited by blocking K$^+$ efflux. Similar to as observed in *Casp1/11*$^{-/-}$ BMDMs, significant macrophage death was still detected in KCl-treated macrophages, again indicating that *C. albicans*-induced macrophage death was likely mediated by both the inflammasome-dependent pyroptosis and inflammasome-independent mechanisms. Interestingly, on one hand, compared to WT *C. albicans*, hyphae-deficient and candidalysin-deficient *C. albicans* still induced less LDH release in KCl-treated macrophages, suggesting that inflammasome-independent macrophage death was also facilitated by candidalysin and hyphae formation (Fig. 8e). This may simply be mediated by the pore-forming activity of candidalysin and hyphae-driven physical peeling of the plasma membrane. On the other hand, KCl effectively inhibited macrophage death

triggered by candidalysin-deficient *C. albicans*, suggesting that K$^+$ efflux was critical for candidalysin-independent macrophage death (Fig. 8e). K$^+$ efflux also appeared to be essential for candidalysin-dependent macrophage death, with 50% of candidalysin-dependent LDH release inhibited by KCl treatment (Fig. 8f). Finally, time-lapse imaging (Supplementary Movie 5 and Fig. 8g) showed that inhibition of K$^+$ efflux did not affect engulfment of *C. albicans* by macrophages but reduced cell lytic death, preventing *C. albicans* escaping from macrophages by penetrating plasma membranes (Fig. 8h). Taken together, our results demonstrate that during *C. albicans* infection, K$^+$ efflux-dependent canonical inflammasomes trigger both the candidalysin-independent and -dependent GSDMD cleavage and macrophage pyroptosis, facilitating escape of *C. albicans* from macrophages.

**Pharmacologic inhibition of GSDMD alleviates *C. albicans* infection.** Our results have shown that inflammasome-mediated GSDMD activation could regulate *C. albicans* escape from macrophages. *C. albicans* infection was significantly reduced in *Gsdmd*$^{-/-}$ mice. Therefore, GSDMD inhibition is likely to be a promising therapeutic strategy for fungal sepsis. Importantly, *Gsdmd* knockout mice are similar to their wild-type littermates in body weight, survival rate, appearance, and behavior, and differential leukocyte counts in the bone marrow and peripheral blood are also normal in *Gsdmd* knockout mice[53]. Thus, targeting GSDMD for fungal sepsis may have few off-target effects. Small molecule inhibitors are ideal for clinical application, and several inhibitors have recently been developed to target the GSDMD pathway[75–80]. Rathkey et al. identified necrosulfonamide (NSA) as a potent inhibitor of GSDMD through direct binding to GSDMD to inhibit pyroptosis; it was also efficacious in sepsis models, suggesting that GSDMD inhibitors may be used clinically in infectious and inflammatory diseases[75]. In nigericin-stimulated LPS-primed BMDM cells, NSA does not inhibit other innate immune pathways and cleavage of GSDMD occurs normally, indicating that NSA does not inhibit caspase-1[75]. Interestingly, a

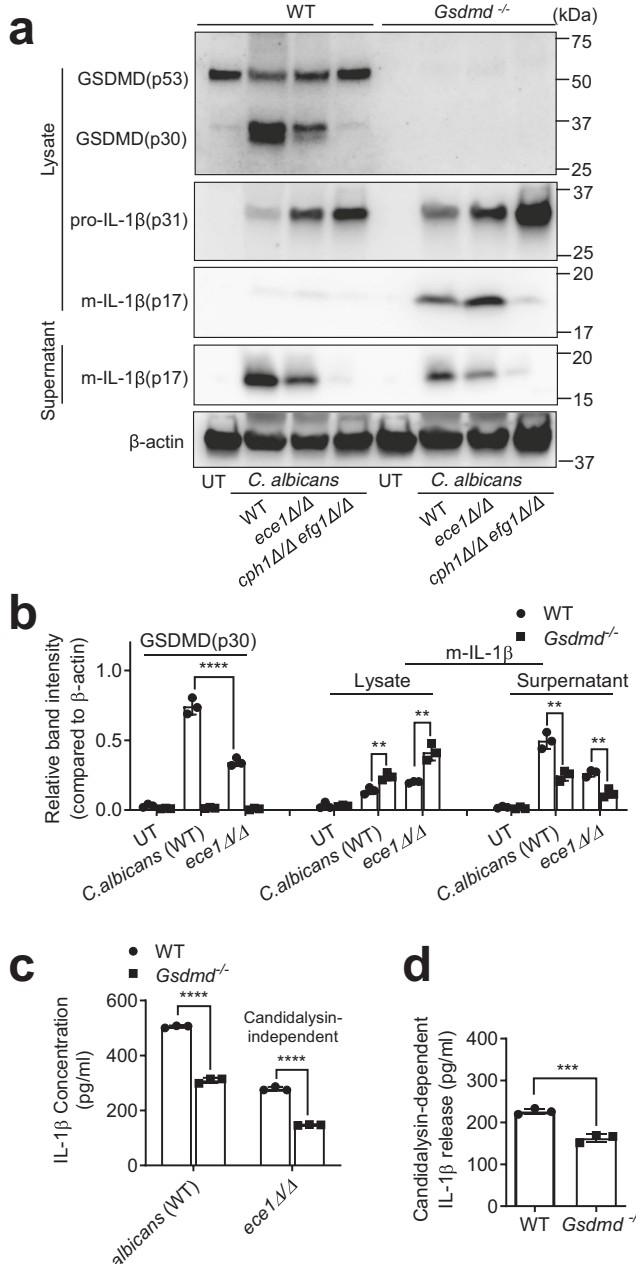

**Fig. 5 GSDMD plays critical roles in both candidalysin-dependent and -independent IL-1β secretion by C. albicans infected macrophage. a, b** C. albicans triggered GSDMD cleavage via both candidalysin-dependent and -independent mechanisms. **a** C. albicans-triggered GSDMD cleavage and IL-1β production. BMDMs from WT and Gsdmd⁻/⁻ mice were infected with WT or candidalysin-deficient (ece1Δ/Δ) C. albicans for 6 h at MOI 50. GSDMD (cleaved p30 and full-length p53), pro-IL-1β, and mature IL-1β (m-IL-1β) in the cell lysates and m-IL-1β in the supernatants were assessed by western blotting. Results are representative of data from three biological replicates. Source data are provided as a Source Data file. **b** The intensities of protein bands were quantified using ImageJ. Two-tailed unpaired Student's t test was used for statistical comparisons. Data shown are means ± SD (n = 3 biologically independent samples per group). p < 0.0001 (GSDMD p30), p = 0.0076 (m-IL-1β in lysate of WT C. albicans-infected BMDM), p = 0.0035 (m-IL-1β in lysate of ece1Δ/Δ-infected BMDM), p = 0.0035 (m-IL-1β in supernatant of WT C. albicans-infected BMDM), p = 0.0032 (m-IL-1β in supernatant of ece1Δ/Δ-infected BMDM) vs. WT. **c, d** GSDMD was required for both candidalysin-dependent and -independent IL-1β secretion. **c** IL-1β secretion by BMDMs. BMDMs were infected with WT or candidalysin-deficient (ece1Δ/Δ) C. albicans (MOI = 10) for 12 h. IL-1β in culture supernatants was measured by ELISA. Two-tailed unpaired Student's t test was used for statistical comparisons. Data shown are means ± SD (n = 3 biologically independent samples per group). Comparisons were conducted with a 2-tailed, unpaired, Student's t-test. p < 0.0001 vs. WT. **d** Candidalysin-dependent IL-1β release calculated by subtracting the level of IL-1β release induced by candidalysin-deficient (ece1Δ/Δ) mutants from that induced by WT C. albicans. Two-tailed unpaired Student's t test was used for statistical comparisons. Data shown are means ± SD (n = 3 biologically independent samples per group). Comparisons were conducted with a 2-tailed, unpaired, Student's t-test. p = 0.0007 vs. WT.

recent study revealed that, in monosodium urate (MSU) crystal–stimulated LPS-primed BMDMs, NSA may also inhibit inflammasomes upstream of GSDMD, thereby preventing pyroptosis independent of GSDMD targeting[81]. We investigated whether NSA could modulate host defense in Candida infection. Treatment with NSA blocked pro-caspase-1, pro-IL-1β and GSDMD cleavage in C. albicans-infected macrophages, consistent with the idea that it can inhibit pyroptosis upstream of GSDMD, in addition to inhibiting GSDMD p30 itself (Fig. S8a, b). Similar to what was observed in GSDMD deficient mice, NSA treatment significantly suppressed, but did not completely inhibit IL-1β production by infected macrophages (Fig. S8c). This effect was specific to IL-1β, with the production of IL-6, TNFα, and MIP2 being less affected (Fig. S8d). NSA effectively inhibited C. albicans-induced macrophage death, as indicated by fewer PI-positive cells being detected in NSA-treated BMDMs compared to untreated BMDMs (Fig. S8e, f). LDH release was also suppressed in NSA-treated BMDMs, with a 40% reduction detected 12 h after

C. albicans infection, again demonstrating that C. albicans-induced macrophage death could be partially blocked by NSA treatment (Fig. S8g).

We next treated mice with NSA before intravenous injection of C. albicans and then administered it daily during the course of C. albicans infection (Fig. 9a). In the absence of infection, NSA-treated mice had similar body weights, survival, appearance, and behavior to untreated mice, with no obvious abnormalities detected. NSA-treated mice displayed enhanced resistance to C. albicans infection, as evidenced by an increased survival rate (Fig. 9b), improved clinical scores (Fig. 9c), less severe kidney enlargement (Fig. 9d, e), and decreased fungal burden (Fig. 9f) compared to untreated mice. It is well documented that NSA can efficiently inhibit GSDMD-mediated pyroptosis. Thus, these results confirmed that inhibition of GSDMD and pyroptosis can induce resistance to Candida infection. As observed in Gsdmd⁻/⁻ mice, NSA-treated mice produced less IL-1β after C. albicans infection (Fig. 9g). Similarly, this modest IL-1β production appeared to be sufficient for initiating host anti-fungal immunity. Collectively, our results demonstrate that pharmacologic inhibition of GSDMD may offer a potential therapeutic strategy for C. albicans infection.

**Pharmacologic inhibition of GSDMD alleviates C. albicans-induced death of human macrophages.** C. albicans infection leads to lytic death of human monocyte derived macrophages (hMDMs). We next explored the role of human GSDMD (hGSDMD) in this process. Similar to what was observed in mouse macrophages, infection with C. albicans triggered robust cleavage of full-length hGSDMD and generated the active N-terminal fragment in hMDMs (Fig. 10a). NSA treatment significantly supressed C. albicans-induced pro-caspase-1 and

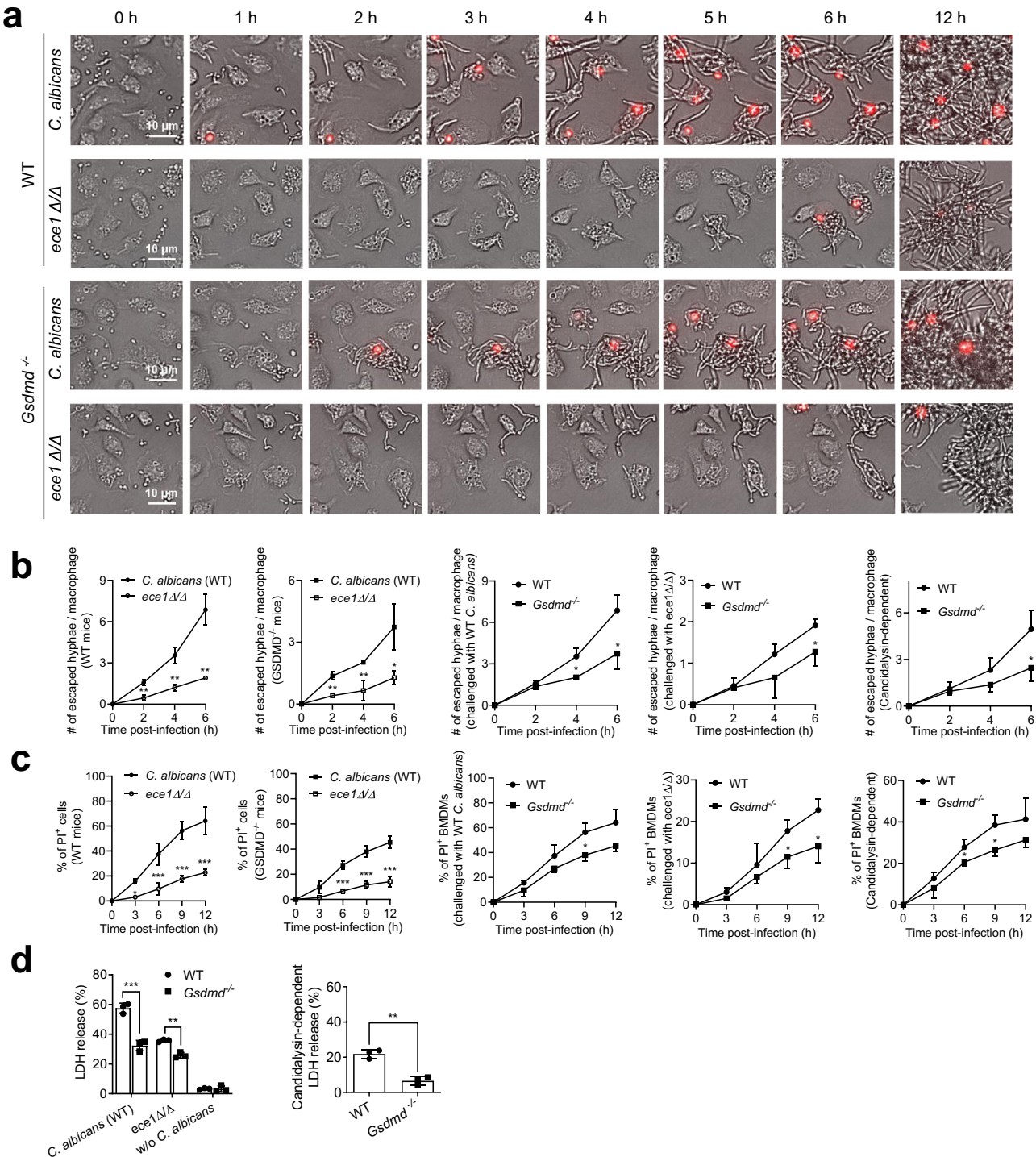

hGSDMD cleavage, as well as IL-1β production (Fig. 10a, b). In NSA-treated hMDMs, pro-IL-1β processing was not affected. In fact, more cleaved mature IL-1β was detected in hMDMs due to the inhibition of its release (Fig. 10a). Using PI to detect the loss of plasma membrane integrity caused by lytic cell death, the percentage of PI+ cells increased gradually after *C. albicans* infection, with NSA-treated hMDMs displaying significantly reduced levels (Fig. 10c, d). We also assessed macrophage lytic death through LDH release. Similarly, *C. albicans* infection induced robust LDH release from hMDMs over time. LDH release was suppressed by NSA treatment, with more than 50% reduction detected at each time points after *C. albicans* infection,

again demonstrating that *C. albicans*-induced human macrophage death could be blocked by GSDMD inhibition (Fig. 10e).

Infection with candidalysin-deficient (*ece1Δ/Δ*) *C. albicans* induced significantly less hMDM death, as indicated by fewer PI-positive cells being detected in (Fig. 10f, g) and less LDH release by (Fig. 10h) candidalysin-deficient *C. albicans*-treated hMDMs compared to WT *C. albicans*-treated hMDMs (Fig. 10f). *ece1Δ/Δ*-induced hMDM death could be further inhibited by NSA, confirming that NSA treatment suppressed candidalysin-independent death of *C. albicans* infected hMDMs (Fig. 10h). Based on the amounts of LDH release triggered by WT and *ece1Δ/Δ C. albicans*, we also calculated candidalysin-dependent

**Fig. 6 GSDMD mediates both candidalysin-dependent and -independent C. albicans escape from macrophages. a–c** Both candidalysin and GSDMD were required for the most efficient escape of engulfed *C. albicans* from macrophages. **a** *C. albicans* escape from macrophages. BMDMs from WT and *Gsdmd*[-/-] mice were incubated with WT or candidalysin-deficient (*ece1Δ/Δ*) *C. albicans* at MOI 10. Dying cells were stained with PI (red). Images were acquired as described in Fig. 3c. The related time-lapse video is included in the **Supplemental Materials**. **b** The number of escaped hyphae per macrophage was assessed as described in Fig. 3d. At least 200 cells were assessed for each sample. Two-tailed unpaired Student's t test was used for statistical comparisons. Data shown are means ± SD (n = 3 biologically independent samples per group). For escaped hyphae/macrophage in WT mice, $p = 0.0025$ (2 h), $p = 0.0029$ (4 h), $p < 0.0015$ (6 h) *vs. C. albicans* (WT). For escaped hyphae/macrophage in *Gsdmd*[-/-] mice, $p = 0.0012$ (2 h), $p = 0.0089$ (4 h), $p < 0.0212$ (6 h) *vs. C. albicans* (WT). For escaped hyphae/macrophage in mice challenged with *C. albicans* (WT), $p = 0.0101$ (4 h), $p = 0.0257$ (6 h) *vs.* WT. For escaped hyphae/macrophage in mice challenged with *ece1Δ/Δ*, $p = 0.0411$ (6 h) *vs.* WT. For escaped hyphae/macrophage (Candidalysin-dependent), $p = 0.0449$ (6 h) *vs.* WT. **c** The percentage of PI-positive BMDMs calculated at each time point. At least 200 cells were assessed for each sample. Two-tailed unpaired Student's t test was used for statistical comparisons. Data shown are means ± SD (n = 3 biologically independent samples per group). For PI[+] cells (WT mice), $p = 0.0485$ (3 h), $p = 0.0007$ (6 h), $p < 0.0009$ (9 h), $p < 0.001$ (12 h) *vs. C. albicans* (WT). For PI[+] cells (*Gsdmd*[-/-] mice), $p = 0.0007$ (6 h), $p = 0.0009$ (9 h), $p < 0.001$ (12 h) *vs. C. albicans* (WT). For PI[+] BMMCs (challenged with *C. albicans* (WT)), $p = 0.019$ (9 h) *vs.* WT. For PI[+] BMMCs (challenged with *ece1Δ/Δ*), $p = 0.0454$ (9 h), $p = 0.0358$ (12 h) *vs.* WT. For PI[+] BMMCs (Candidalysin-dependent), $p = 0.0391$ (6 h), $p = 0.018$ (9 h) *vs.* WT. **d** *C. albicans*-induced BMDM death assessed by the LDH assay. WT and *Gsdmd*[-/-] BMDMs were incubated (uninfected BMDMs as controls) with WT or candidalysin-deficient (*ece1Δ/Δ*) *C. albicans* at MOI 10 for 12 h. Relative LDH release was assessed as described in Fig. 3f. Candidalysin-dependent LDH release calculated by subtracting the level of LDH release induced by candidalysin-deficient (*ece1Δ/Δ*) mutants from that induced by WT *C. albicans*. Two-tailed unpaired Student's t test was used for statistical comparisons. Data shown are means ± SD (n = 3 biologically independent samples per group). $p = 0.0008$ (*C. albicans* (WT)), $p = 0.0011$ (*ece1Δ/Δ*) *vs.* WT.

hMDM death. This process was also suppressed by NSA treatment, demonstrating that GSDMD was important for both candidalysin-independent and -dependent hMDM death (Fig. 10h). We next measured the level of secreted IL-1β in the supernatant by ELISA. Similarly, IL-1β release was significantly reduced when hMDMs were treated with candidalysin-deficient *C. albicans*. However, candidalysin disruption in *C. albicans* did not completely abolish IL-1β release from hMDMs, indicating the presence of candidalysin-independent IL-1β production (Fig. 10i). IL-1β release induced by *ece1Δ/Δ* was significantly inhibited in NSA-treated hMDMs, confirming the essential role of GSDMD in both candidalysin-dependent and -independent IL-1β release (Fig. 10i).

Finally, similar to what was observed in mouse macrophages, *C. albicans*-induced hGSDMD activation and hMDM death were mediated by the caspase-1 inflammasome. Treatment with Caspase-1 inhibitor VX-765 significantly suppressed *C. albicans*-induced hGSDMD cleavage (Fig. S9a), IL-1β secretion (Fig. S9a, b), PI uptake (Fig. S9c, d), and LDH release (Fig. S9e). Additionally, VX-765 inhibited both candidalysin-independent and -dependent hMDM death (Fig. S9f) and IL-1β release (Fig. S9g). However, unlike NSA, which did not affect pro-IL-1β processing, VX765 significantly suppressed pro-IL-1β cleavage, with little m-IL-1β detected in hMDMs or supernatant (Fig. S9a, b and Fig. S9g).

## Discussion

The incidence of *C. albicans* infections has increased dramatically over the last decades: increased prevalence of chronic illnesses, antibiotic resistance, and numbers of immunocompromised patients due to surgery, chemotherapy, and organ transplantation, have created a large vulnerable population[13–15,82–85]. Despite an expanding antifungal armamentarium, treatment unfortunately is often unsuccessful and resistance can emerge[86–91]. Given the underlying morbidities of many patients, disseminated candidiasis remains a life-threatening infection and fungal sepsis is a high clinical priority[1,82,92–95]. Nevertheless, the precise mechanisms leading to *C. albicans*-induced sepsis remain unclear. Despite decades of exhaustive research, no significant breakthroughs have emerged and there are no FDA-approved drugs or effective cures for sepsis[4,6,16,19,96–102]. Thus, alternative therapeutic strategies are needed. The significance of developing personalized host-modulating therapies to counteract the growing threat of antifungal resistance is well recognized, because such

resistance cannot simply be overcome by using microbe-centered strategies. Here we find a key role for macrophage GSDMD in *C. albicans* infection and established it as a potential host-directed therapeutic target for *C. albicans*-induced sepsis.

Host cells are equipped with extracellular and cytoplasmic pathogen sensors such as Toll-like receptors (TLRs) and nucleotide-binding domain and leucine-rich repeat-containing family receptors (NLRs). Activation of these sensors triggers the assembly of the inflammasome, a multimeric protein complex containing active caspase-1, which converts pro-IL-1β into its biologically active form. IL-1β subsequently initiates and further amplifies innate and adaptive immune responses. Overwhelming and sustained IL-1 release contributes to exaggerated systemic inflammation and multiple organ damage in severe sepsis, making inhibition of inflammasome-mediated cytokine production a valid strategy to prevent sepsis[26,103,104]. Additionally, macrophage death is an important cellular host defense mechanism since it mediates the secretion of IL-1β. GSDMD is known to be a key mediator of inflammasome-mediated macrophage pyroptotic death and IL-1β secretion[47–49]. Macrophage pyroptosis, triggered by inflammasomes, caspase-1/11, and GSDMD, is a major type of macrophage lytic cell death. Thus, for bacterial sepsis where IL-1β facilitates septic shock, inhibition of GSDMD-mediated pore formation is considered an effective therapeutic strategy to prevent early shock caused by systemic hyperinflammation. GSDMD disruption indeed protects against lethal bacterial sepsis[47–49,53,54], and *Gsdmd*-deficient mice show significantly improved survival compared to WT mice in both LPS-induced and cecal ligation and puncture (CLP) sepsis models[47–49,53]. These results agree with observations in inflammasome (e.g., NLRP3, capsase-1/11, and ASC)-deficient mice, in which IL-1β production and bacterial sepsis are also suppressed[26,31–36,103,104].

However, inflammasomes appear to play a different role in fungal infection. As a key component of the innate immune response, inflammasome activation has been extensively investigated in *C. albicans* infection. Previous studies have suggested a critical role for innate immune receptors in mediating inflammatory responses against *C. albicans*. Upon *Candida* infection, NLRP3-mediated inflammasome assembly is induced by key pattern recognition receptors and their downstream molecules, including TLR2, Dectin-1, Syk and ZBP1[42–44,105]. Inflammasome activation plays essential roles in host defenses against *C. albicans* including inducing protective antifungal Th1/Th17 responses. Mice lacking key inflammasome components such as ASC,

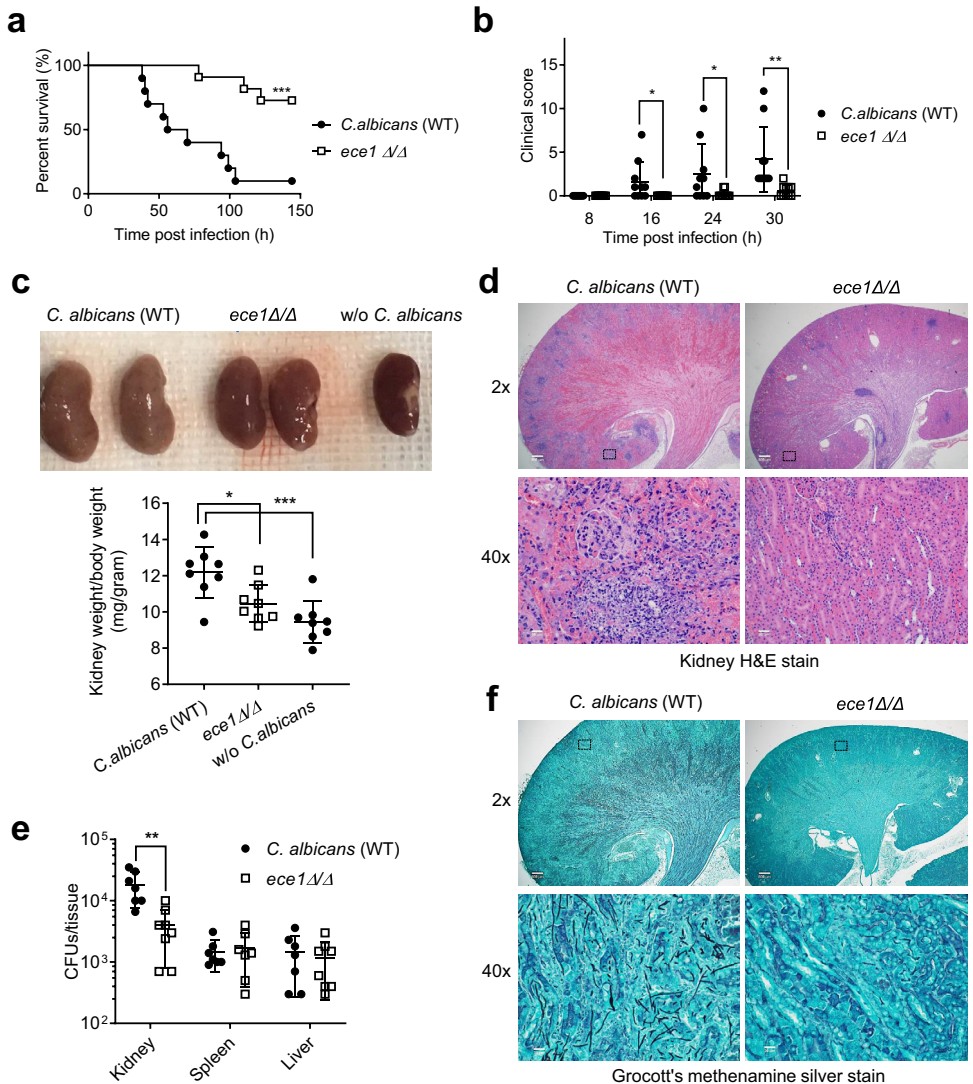

**Fig. 7 Candidalysin-deficient C. albicans cause less severe infection and tissue damage. a** Kaplan-Meier survival plots of mice infected with WT or candidalysin-deficient (*ece1Δ/Δ*) *C. albicans*. Mice were intravenously challenged with 1 × 10⁶ CFU *C. albicans*. Survival rates were analyzed using Kaplan-Meier survival curves and log-rank testing. $p = 0.0003$ vs. WT. **b** Clinical scores of mice challenged with WT or candidalysin-deficient (*ece1Δ/Δ*) *C. albicans*. Two-tailed unpaired Student's t test was used for statistical comparisons. Data shown are means ± SD ($n = 10$ mice per group). $p = 0.0388$ (16 h), $p = 0.0483$ (24 h), $p = 0.0074$ (30 h) vs. WT *C. albicans*. **c** Up panel, comparison of the kidneys of mice challenged with WT or candidalysin-deficient (*ece1Δ/Δ*) *C. albicans*. Mice were sacrificed 2 days after intravenous injection of 1 × 10⁶ CFU *C. albicans*. Shown are representative images. Lower panel, kidney weights of *C. albicans*-challenged mice. Two-tailed unpaired Student's *t* test was used for statistical comparisons. Data shown are means ± SD ($n = 8$ kidneys per group). $p = 0.0136$ (*ece1Δ/Δ*), $p = 0.0008$ (w/o *C. albicans*) vs. WT *C. albicans*. **d** Histopathologic assessment of the kidneys of mice challenged with WT or candidalysin-deficient (*ece1Δ/Δ*) *C. albicans*. Mice were sacrificed 2 days after intravenous injection of 1 ×10⁶ CFU *C. albicans*. Shown are representative H&E-stained sections of kidney tissues. Experiments were repeated three times. **e** Fungal burden of kidneys, livers, and spleens of mice challenged with WT or candidalysin-deficient (*ece1Δ/Δ*) *C. albicans*. Mice were sacrificed 2 days after intravenous injection of 1 × 10⁶ CFU *C. albicans*. Two-tailed unpaired Student's t test was used for statistical comparisons. Data shown are means ± SD ($n = 7$ for WT *C. albicans*-challenged mice, $n = 8$ for *ece1Δ/Δ*-challenged mice per group). $p = 0.0032$ (kidney) vs. WT *C. albicans*. **f** *C. albicans* in the kidney were identified by Grocott methenamine silver staining. Mice were sacrificed 2 days after intravenous injection of 1 × 10⁶ CFU *C. albicans*. Results are representative of data from at least three biological replicates.

NLRP3, and caspase-1/11 are hypersusceptible to *C. albicans* infection[45,46,106]. Of note, NLRP3-independent non-canonical inflammasome activation has also been implicated in *Candida* infection. Dectin-1 was identified as a key extracellular pathogen sensor for IL-1β induction and processing, directly activating a non-canonical caspase-8 inflammasome for pro-IL-1β processing and without the need for internalization of fungal pathogens[107,108]. Despite the complex mechanism, all these studies suggest that inflammasome activation confers essential protective immunity against *C. albicans* infection. Thus, unlike in bacterial sepsis, inhibition of inflammasome-mediated cytokine production is currently not considered a valid strategy to prevent *C. albicans*-induced sepsis.

Paradoxically, here we discovered that although disruption of caspase-1/11 aggravated *C. albicans* infection as previously reported[45,46,106], disruption of GSDMD, an effector molecule downstream of caspase-1/11, protected against *C. albicans*-induced sepsis. What causes this discrepancy? Fine-tuning of inflammasome-mediated IL-1β production is critical for *C. albicans*-induced pathology. Proinflammatory cytokines, including

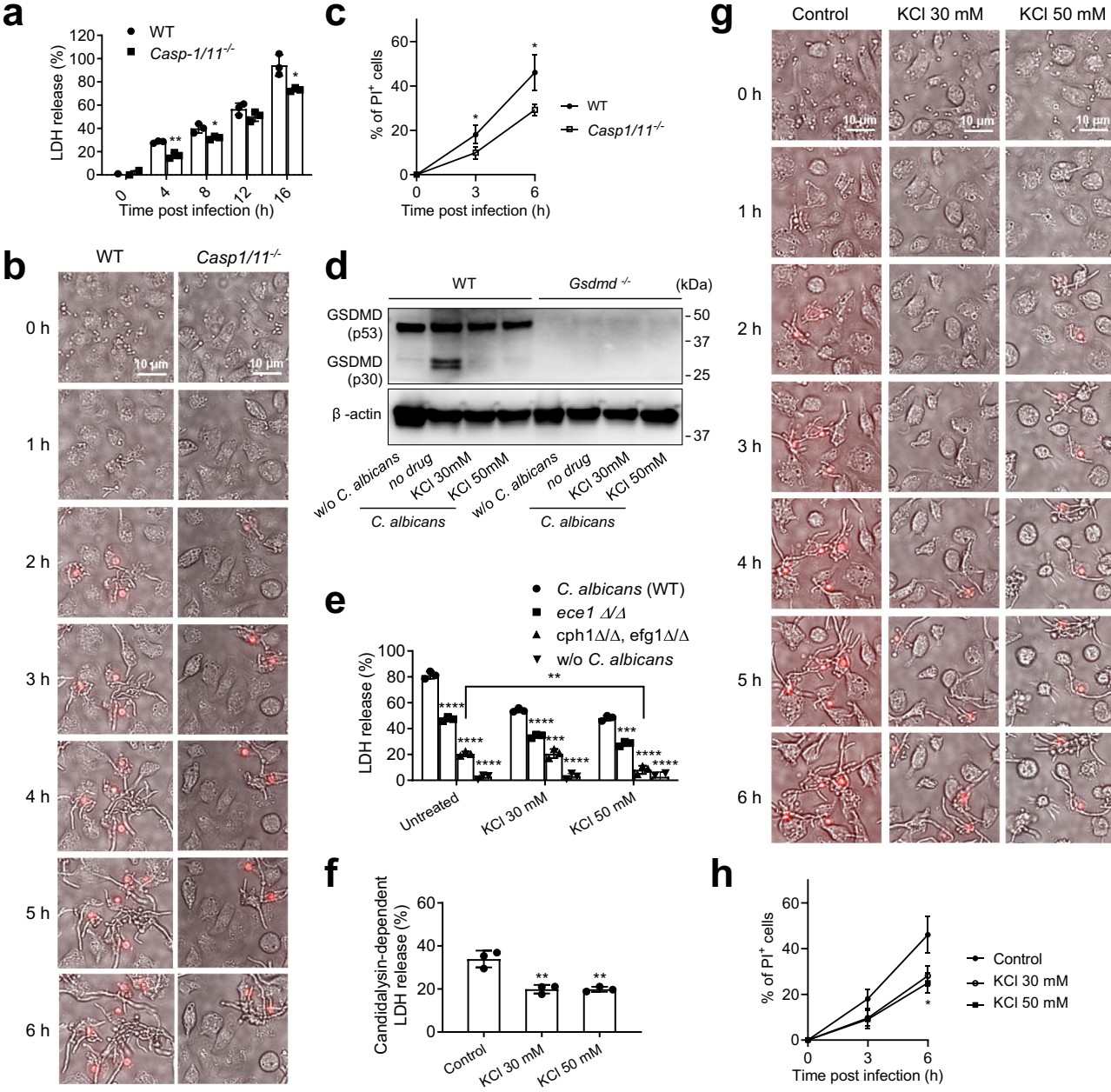

IL-1β are key initiators of sepsis-associated life-threatening organ dysfunction. Profound cytokine release is associated with *C. albicans*-induced immunopathology. However, IL-1β is also required for antifungal immunity. Inflammasome-mediated processing of pro-IL-1β is critical for the generation of mature IL-1β. *C. albicans*-induced IL-1β secretion is almost completely diminished in macrophages isolated from inflammasome-null mice (e.g., *Nlrp3, Asc, Casp1* knockouts)[45,46,106]. No mature IL-1β was detected inside caspase-1/11-deficient macrophages, confirming that the lack of IL-1β production was primarily caused by complete inhibition of caspase-mediated processing of pro-IL-1β. Caspase-1/11 also play a role in pyroptosis, a key mechanism mediating IL-1β release from macrophages, by mediating GSDMD cleavage and activation. However, GSDMD depletion only partially blocked IL-1β release upon *C. albicans* infection, indicating that IL-1β secretion can also be driven by mechanisms independent of GSDMD-mediated pyroptosis[109,110]. During *C. albicans* infection, mature IL-1β was generated via inflammasome activation even in GSDMD-deficient macrophages. Consistently,

*Gsdmd*[-/-] mice could still release significant amounts of mature IL-1β via GSDMD-independent mechanisms to mount a sufficient anti-*Candida* immune response without exaggerating tissue inflammation and multiorgan damage. Therefore, the aggravated *C. albicans* infection observed in inflammasome-deficient mice occurs as a result of abolished IL-1β production. GSDMD disruption only partially reduced IL-1β production in *C. albicans* infected hosts, with the modest IL-1β production appearing to be sufficient for initiating anti-fungal immunity. Indeed, in the presence of interleukin 1 receptor antagonist anakinra, the protective effect elicited by GSDMD disruption was abolished. The fact that GSDMD disruption alleviated and inflammasome disruption aggravated *C. albicans* infection indicates functional uncoupling of GSDMD and the inflammasome during *C. albicans*-induced IL-1β processing and secretion. This uncoupling is unique to *C. albicans* infection and was not observed in bacterial[47–49,53,54] or viral infection[111], where GSDMD and inflammasome disruption conferred the same phenotypes. In summary, this study demonstrates key functions for GSDMD in

**Fig. 8 C. albicans-induced GSDMD cleavage and C. albicans escape from macrophages relies on inflammasome activation. a** Casp1/11-mediated *C. albicans*-induced BMDM death. BMDMs from WT and *Casp-1/11$^{-/-}$* mice were infected with *C. albicans* at MOI 10. BMDM death at the indicated time points was measured as relative LDH release as described in Fig. 3f. Two-tailed unpaired Student's *t* test was used for statistical comparisons. Data shown are means ± SD (n = 3 biologically independent samples per group). $p = 0.0016$ (4 h), $p = 0.0282$ (8 h), $p = 0.0172$ (16 h) *vs.* WT. **b** Casp1/11-mediated *C. albicans* escape from macrophages. BMDMs from WT and *Casp-1/11$^{-/-}$* mice were infected with *C. albicans* at MOI 10. Dying cells were stained with PI (red). Experiments were repeated three times. The related time-lapse video is included in the **Supplemental Materials**. **c** The percentage of PI-positive BMDMs calculated at each time point. At least 100 cells were assessed for each sample. Two-tailed unpaired Student's t test was used for statistical comparisons. Data shown are means ± SD (n = 3 biologically independent samples per group). $p = 0.0484$ (3 h), $p = 0.0257$ (6 h) *vs.* WT. **d** Potassium efflux was essential for *C. albicans*-triggered GSDMD cleavage in BMDMs. BMDMs from WT or *Gsdmd$^{-/-}$* mice were cultured in the indicated concentrations of KCl for 1 h and then infected with WT *C. albicans* for 6 h at MOI 50. Full-length (p53) and cleaved (p30) GSDMD in the cell lysates were detected by western blotting. Results are representative of data from three biological replicates. Source data are provided as a Source Data file. **e** *C. albicans*-induced BMDM death assessed by LDH assay. BMDMs were cultured in the indicated concentrations of KCl for 1 h and then infected (uninfected BMDMs as controls) with WT, candidalysin-deficient (*ece1Δ/Δ*), or hyphae-deficient (*cph1Δ/Δ, efg1Δ/Δ*) *C. albicans* at MOI 10 for 12 h. Relative LDH release was assessed as described in Fig. 3f. Two-tailed unpaired Student's t test was used for statistical comparisons. Data shown are means ± SD (n = 3 biologically independent samples per group). For *cph1Δ/Δ, efg1Δ/Δ*-infected BMMCs, $p = 0.0016$ (KCl 50 mM) *vs.* KCl untreated macrophages. For KCl -untreated BMMCs, $p < 0.0001$ (*ece1Δ/Δ, cph1Δ/Δ, efg1Δ/Δ*, w/o *C. albicans*) *vs.* WT *C. albicans*. For KCl 30 mM -treated BMMCs, $p = 0.0001$ (*cph1Δ/Δ, efg1Δ/Δ*), $p < 0.0001$ (*ece1Δ/Δ*, w/o *C. albicans*) *vs.* WT *C. albicans*. For KCl 50 mM-treated BMMCs, $p = 0.0001$ (*ece1Δ/Δ*), $p < 0.0001$ (*cph1Δ/Δ, efg1Δ/Δ*, w/o *C. albicans*) *vs.* WT *C. albicans*. **f** Candidalysin-dependent LDH release calculated by subtracting the level of LDH release induced by candidalysin-deficient (*ece1Δ/Δ*) *C. albicans* from that induced by WT *C. albicans*. Two-tailed unpaired Student's t test was used for statistical comparisons. Data shown are means ± SD (n = 3 biologically independent samples per group). $p = 0.0052$ (KCl 30 mM), $p = 0.0036$ (KCl 50 mM) *vs.* KCl untreated macrophages. **g** Potassium efflux was critical for *C. albicans* escape from macrophages. BMDMs were cultured in the indicated concentrations of KCl for 1 h and then infected with WT *C. albicans* at MOI 10. Dying cells were stained with PI (red). Images were acquired as described in Fig. 3c. Shown are representative images at the indicated time points. The related time-lapse video is included in the **Supplemental Materials**. **h** The percentage of PI-positive BMDMs calculated at each time point. At least 100 cells were assessed for each sample. Two-tailed unpaired Student's t test was used for statistical comparisons. Data shown are means ± SD (n = 3 biologically independent samples per group). $p = 0.016$ (KCl 50 mM) *vs.* KCl untreated macrophages.

host defense again *Candida* infection and suggests that GSDMD may be a potential therapeutic target in *C. albicans*-induced sepsis.

GSDMD was originally identified as a key factor responsible for the inflammatory form of lytic pyroptotic death in macrophages[47–49]. However, this protein is also expressed in other cell types including various immune cells other than macrophages. We recently revealed that GSDMD is highly expressed in neutrophils and is a key regulator of neutrophil death[53]. GSDMD cleavage and activation in neutrophils can be caspase-independent and instead mediated by a neutrophil-specific serine protease, neutrophil elastase (ELANE), released from neutrophil granules into the cytosol during neutrophil death. ELANE-derived N-terminal fragment is fully active and can induce lytic cell death as efficiently as the caspase-derived N-terminal fragment[53,80]. It is intriguing to see whether ELANE-mediated activation of GSDMD in neutrophils plays a role in anti-Candida host defense. Similarly, the contribution of GSDMD in NK, DC, T, and B cells also remains elusive.

GSDMD is a member of the gasdermin family. There are five paralogue GSDMs (A-E) in humans, most of which have been shown to form pores that can activate pyroptosis when cleaved. While GSDMD is central to pyroptosis in myeloid cells, other gasdermins can also modulate cell death. For instance, Gasdermin E (GSDME) is turning out to be a critical GSDM in cancer cells. It is cleaved and activated to form pores by caspase-3 and classical apoptotic stimuli, and by killer lymphocyte granzyme B (GzmB) when cells are recognized as targets for immune elimination. GzmB both directly cleaves GSDME and indirectly activates GSDME because it activates caspase-3. GSDME activation converts noninflammatory apoptosis to inflammatory pyroptosis[112–116]. A recent study showed that intratumoral delivery of nanoparticles conjugated with a precleaved form of mouse gasdermin A (GSDMA3) could cause selective tumor cell pyroptosis[117]. Caspase-8, the caspase activated by the TNF death receptor to trigger apoptosis, can cleave gasdermins C and D (GSDMC, GSDMD) to convert apoptosis to pyroptosis[118–124]. Gasdermin B (GSDMB) is a substrate of granzyme A (GzmA), the

other abundant granzyme in killer T and NK cells[125]. Whether gasdermin family members other than GSDMD are also involved in anti-Candida host defense needs to be further investigated.

We also identified the mechanism augmenting antifungal immunity in *Gsdmd$^{-/-}$* mice. Host–pathogen interactions and their role in cell death are highly complex, involving a fine balance between pro- and antideath strategies for both host and pathogen. Pathogens can increase their success by hijacking hose defense system components. Programmed cell death is a central player in Candida infections[126–128], and membrane disruption is a key mechanism by which pathogens destroy immune cells for self-survival. *C. albicans* is known to trigger macrophage lytic death and use this as a mechanism to facilitate their escape from the macrophages[45,46]. Pyroptosis is an inflammatory form of programmed cell death that is triggered by invading pathogens. GSDMD-mediated macrophage pyroptosis has evolved as an efficient host defense mechanism, particularly against intracellular pathogens[47–49]. Thus, pyroptosis is usually an advantageous adaptation that removes the niche for intracellular pathogen replication. However, GSDMD disruption paradoxically improved host anti-*C. albicans* responses, suggesting that this particular pathogen may have hijacked this defense mechanism to improve its survival in infected hosts. *C. albicans*-triggered excessive activation of GSDMD-mediated pyroptosis led to widespread cell death, causing tissue damage, organ failure, and lethal septic shock, and this mechanism was abolished in *Gsdmd$^{-/-}$* mice. GSDMD-deficient macrophages became more resistant to *C. albicans*-induced pyroptosis, preventing the escape of engulfed *Candida* from macrophages, thereby reducing the fungal burden and improving outcomes in *C. albicans*-infected hosts.

At the cellular and molecular level, *C. albicans*-induced GSDMD cleavage was mainly mediated by K$^+$ efflux-dependent canonical inflammasomes and could be regulated by hyphae formation and candidalysin. *C. albicans* virulence is governed by its ability to readily interconvert between a singular budding yeast form and a filamentous hyphae form[44,129]. While the yeast form reproduces quickly and rapidly proliferates, the hyphal form invades and destroys host cells and tissues. Using key adhesin

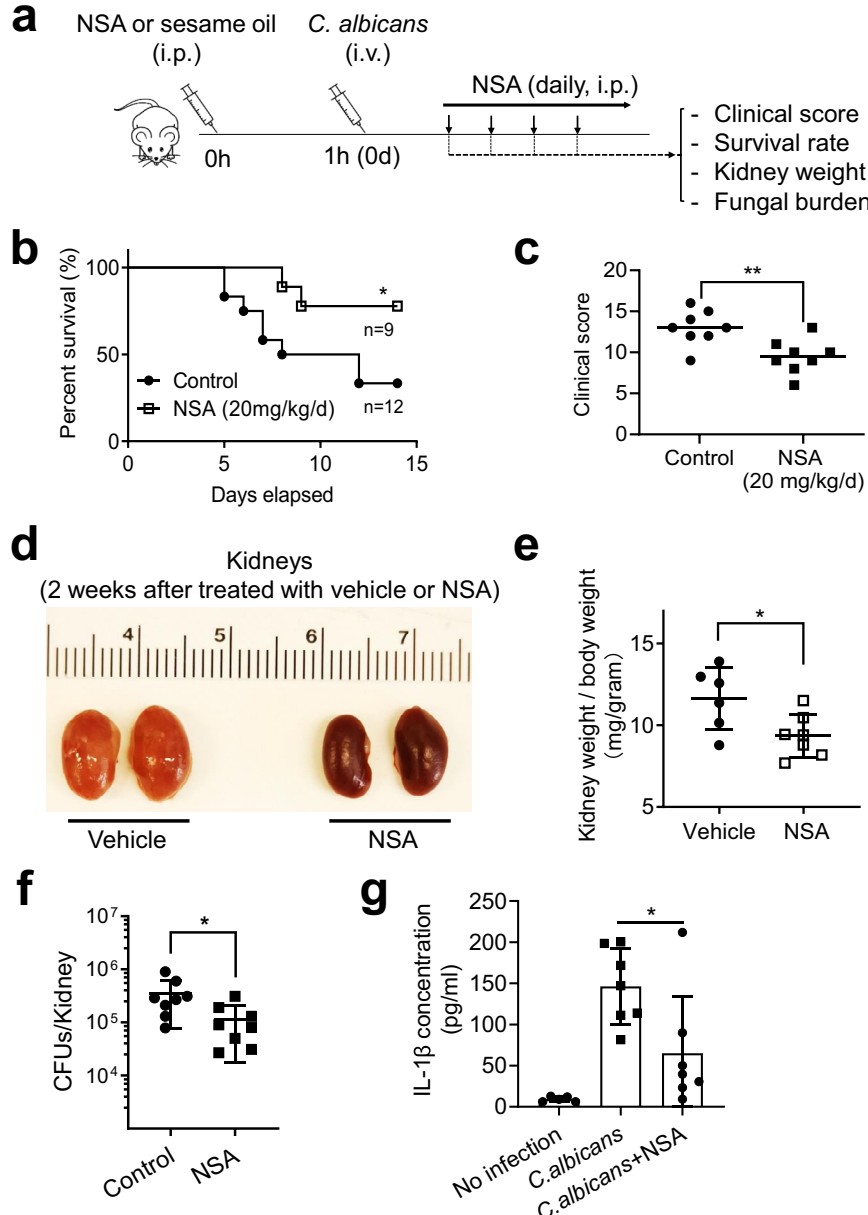

**Fig. 9 GSDMD antagonist NSA alleviates C. albicans infection in mice. a** Experimental scheme for assessing the effects of pharmacologic inhibition of GSDMD on *C. albicans* infection. Mice were intravenously challenged with $2 \times 10^5$ CFU *C. albicans* 1 h after intraperitoneal injection of sesame oil alone (control) or NSA (dissolved in sesame oil, 20 mg/kg body weight). The challenged mice received daily NSA treatment (20 mg/kg body weight) and were monitored for 15 days. Controls were injected with sesame oil only during the same time period. **b** Kaplan-Meier survival plots of *C. albicans*-challenged mice. Survival rates were analyzed using Kaplan-Meier survival curves and log-rank testing. $p = 0.0427$ *vs.* control. **c** Clinical score of *C. albicans*-challenged untreated and NSA-treated mice. Two-tailed unpaired Student's t test was used for statistical comparisons. Data presented are the means ± SD ($n = 8$ mice per data point) of three independent experiments. $p = 0.005$ *vs.* control. **d** Comparison of the kidneys of *C. albicans*-challenged untreated and NSA-treated mice. Mice were sacrificed 2 weeks after the *C. albicans* infection. Shown are the representative images from three independent experiments. **e** Kidney weights of *C. albicans*-challenged mice. Two-tailed unpaired Student's t test was used for statistical comparisons. Data presented are the means ± SD ($n = 6$ for Vehicle, $n = 7$ for NSA-treated mice per data point) of three independent experiments. $p = 0.0279$ *vs.* control. **f** Fungal burden of kidneys from untreated and NSA-treated mice 2 days after *C. albicans* infection. Two-tailed unpaired Student's t test was used for statistical comparisons. Data presented are the means ± SD ($n = 8$ mice per data point) of three independent experiments. $p = 0.0379$ *vs.* control. **g** *C. albicans*-induced IL-1β production in untreated and NSA-treated mice. Mice were intravenously challenged with $2 \times 10^5$ CFU *C. albicans*. Two-tailed unpaired Student's t test was used for statistical comparisons. Data shown are means ± SD ($n = 5$ for uninfected, $n = 7$ for *C. albicans* infected mice). $p = 0.0242$ *vs.* control (untreated mice).

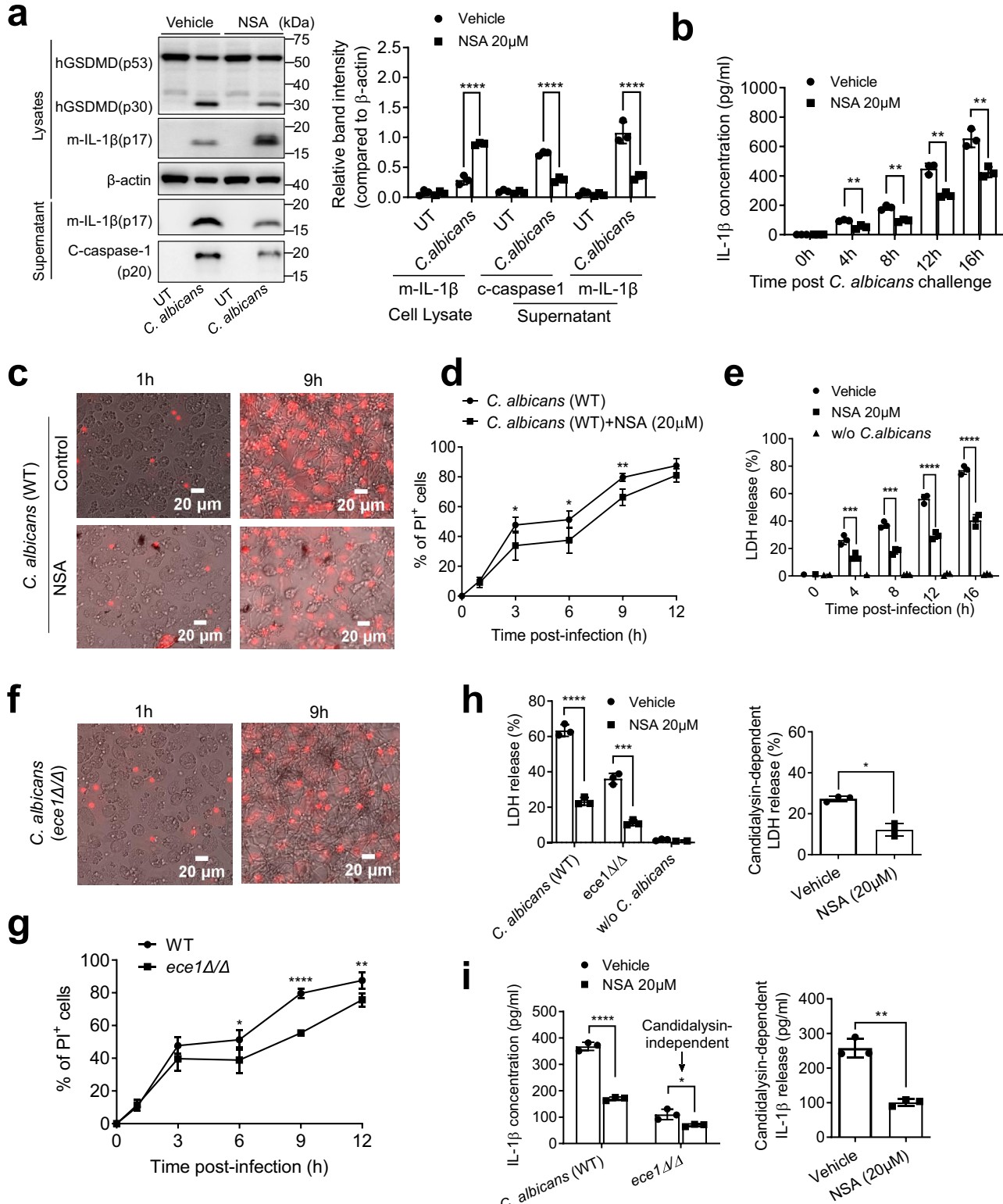

structures, hyphae strongly attach to host cells and modulate different invasion mechanisms including induced endocytosis and active intercalation[130]. The latter confers an additional benefit to virulent *C. albicans*, namely the ability to escape the phagolysosome and thus disrupt host defense mechanisms[131]. In this study, we demonstrated that hyphae formation was also critical for *C. albicans*-induced GSDMD cleavage and the subsequent death of host macrophages.

*C. albicans*-induced phagocyte damage was partially attributed to its ability to physically disrupt cellular integrity through conversion from the yeast to the filamentous form. Additionally, candidalysin, a hyphae-associated cytolytic pore-forming peptide toxin, was recently identified as a *C. albicans* virulence factor involved in mucosal and systemic infection[63–65,132–135]. Candidalysin is both a central trigger for NLRP3 inflammasome-dependent caspase-1 activation via K[+] efflux and a key driver of

**Fig. 10 GSDMD antagonist NSA alleviates C. albicans-induced death of human macrophages. a** NSA suppressed *C. albicans*-triggered hGSDMD cleavage in human monocyte-derived macrophages (hMDMs). NSA treated and untreated hMDMs were infected with WT *C. albicans* for 6 h at MOI 50. Full-length (p53) hGSDMD, cleaved (p30) hGSDMD, mature IL-1β (m-IL-1β), and actin in the cell lysates, as well as m-IL-1β and cleaved caspase-1 (C- caspase-1) in the supernatant, were detected by western blotting. The figure shows the result of a representative experiment that was repeated 3 times. The intensities of protein bands were quantified using ImageJ. UT, uninfected. Two-tailed unpaired Student's *t* test was used for statistical comparisons. Data shown are means ± SD. *p* < 0.0001 *vs.* untreated (Vehicle). Source data are provided as a Source Data file. **b** IL-1β secretion by hMDMs. hMDMs were infected with *C. albicans* (MOI = 10) for the indicated time periods. IL-1β in culture supernatants was measured by ELISA. Two-tailed unpaired Student's t test was used for statistical comparisons. Data shown are means ± SD (n = 3 biologically independent samples per group). *p* = 0.005 (4 h), *p* = 0.0012 (8 h), *p* = 0.0014 (12 h), *p* = 0.0048 (16 h) *vs.* untreated (Vehicle). **(c)** NSA treated and untreated hMDMs were incubated with *C. albicans* at MOI 10. Dying cells were stained with propidium iodide (PI) (red) (200 ng/mL). Images were acquired at indicated time points using 20× dry lens. Shown are representative images at the indicated time points. Scale bars represent 20 μm. Experiments were repeated three times. **(d)** The percentage of PI-positive hMDMs calculated at each time point. At least 200 cells were assessed for each sample. Two-tailed unpaired Student's t test was used for statistical comparisons. Data shown are means ± SD (n = 4 biologically independent samples per group). *p* = 0.044 (3 h), *p* = 0.384 (6 h), *p* = 0.049 (9 h) *vs.* NSA untreated. **e** *C. albicans*-induced hMDM death assessed by LDH cytotoxicity assay. NSA treated and untreated hMDMs were incubated with *C. albicans* at MOI 10. Relative LDH release was expressed as the percentage LDH activity in supernatants of cultured cells (medium) compared with total LDH (from the medium and cells) and used as an index of cytotoxicity. Two-tailed unpaired Student's t test was used for statistical comparisons. Data shown are means ± SD (n = 3). Results are representative of data from three biological replicates. *p* = 0.0007 (4 h), *p* = 0.0002 (8 h), *p* < 0.0001 (12 h), *p* < 0.0001 (16 h) *vs.* NSA untreated. **f, g Candidalysin partially mediated *C. albicans*-induced death of human macrophages. f** hMDMs were incubated with WT (Fig. 10c) or candidalysin-deficient (*ece1Δ/Δ*) *C. albicans* at MOI 10. Dying cells were stained with PI (red). Images were acquired as described in Fig. 10c. **g** The percentage of PI-positive hMDMs calculated at each time point. At least 200 cells were assessed for each sample. Data shown are means ± SD (n = 4 biologically independent samples per group). Two-tailed unpaired Student's t test was used for statistical comparisons. *p* = 0.0429 (6 h), *p* < 0.0001 (9 h), *p* < 0.0096 (12 h) *vs.* WT *C. albicans*. **(h)** *C. albicans*-induced hMDM death assessed by the LDH assay. NSA treated and untreated hMDMs were incubated with WT or candidalysin-deficient (*ece1Δ/Δ*) *C. albicans* at MOI 10 for 12 h. Relative LDH release was assessed as described in Fig. 10e. Candidalysin-dependent LDH release was calculated by subtracting the level of LDH release induced by candidalysin-deficient (*ece1Δ/Δ*) mutants from that induced by WT *C. albicans*. Two-tailed unpaired Student's t test was used for statistical comparisons. Data shown are means ± SD (n = 3 biologically independent samples per group). For LDH release, *p* < 0.0001 (WT *C. albicans*), *p* = 0.0002 (*ece1Δ/Δ*) *vs.* untreated (Vehicle). For candidalysin-dependent LDH release, *p* = 0.0015 *vs.* untreated (Vehicle). **(i)** GSDMD was required for both candidalysin-dependent and -independent IL-1β secretion. hMDMs were infected with WT or candidalysin-deficient (*ece1Δ/Δ*) *C. albicans* (MOI = 10) for 12 h. IL-1β in culture supernatants was measured by ELISA. Candidalysin-dependent IL-1β release was calculated by subtracting the level of IL-1β release induced by candidalysin-deficient (*ece1Δ/Δ*) mutants from that induced by WT *C. albicans*. Two-tailed unpaired Student's t test was used for statistical comparisons. Data shown are means ± SD (n = 3 biologically independent samples per group). For IL-1β concentration, *p* < 0.0001 (WT *C. albicans*), *p* = 0.0248 (*ece1Δ/Δ*) *vs.* untreated (Vehicle). For candidalysin-dependent IL-1β release, *p* = 0.0007 *vs.* untreated (Vehicle).

---

inflammasome-independent cytolysis of macrophages upon infection with *C. albicans*[64,65]. Consistent with previous studies, we also found that membrane disruption and cell death caused by candidalysin could be inflammasome-independent and that IL-1β was still released in a candidalysin-dependent manner through an inflammasome-independent pathway. Candidalysin is a pore-forming toxin that causes membrane perturbations, leading to K$^+$ efflux and subsequent inflammasome activation and pyroptosis. Candidalysin facilitates IL-1β processing and release via inflammatory caspases; however, macrophages infected with candidalysin-deficient mutants still released significant amounts of IL-1β, suggesting that other fungal components (e.g., β-D-glucans) could activate inflammasomes and process IL-1β during *C. albicans* infection. Additionally, we also observed significant GSDMD cleavage and cell death when macrophages were incubated with candidalysin-deficient *C. albicans*, supporting the existence of a candidalysin-independent death program. Presumably, during *C. albicans* infection, hyphal proteins other than candidalysin may facilitate GSDMD cleavage and plasma membrane perforation, leading to host cell death. Alternatively, in a candidalysin-independent manner, macrophages might be mechanically destroyed by hyphae inside the cells that physically pierce the host cell membrane. Notably, filamentation is necessary but not sufficient to trigger NLRP3 inflammasome-mediated pyroptosis, indicating that *C. albicans*-mediated macrophage damage is not solely due to hyphae-induced physical disruption of cellular integrity[46].

Collectively, our results suggest that the host response to *C. albicans* infection is complex and mediated by multiple factors and pathways (Fig. S10). **First**, both inflammasome-dependent and -independent mechanisms are involved. Upstream inflammasome and caspase-1 activation are essential for processing pro-

IL-1β to mature IL-1β as well as *C. albicans*-induced GSDMD cleavage. However, macrophage death and secretion of mature IL-1β can occur in the absence of inflammasome activation. **Second**, *C. albicans*-induced macrophage death and mature IL-1β secretion can be triggered by both the GSDMD-dependent pyroptosis and GSDMD-independent mechanisms. In the absence of GSDMD, secretion of caspase-processed mature IL-1β can still be supported by GSDMD-independent mechanisms. Thus, although inflammasome activation is absolutely required for IL-1β production and anti-*Candida* host defenses, significant amounts of IL-1β can still be produced in *Gsdmd*$^{-/-}$ mice to maintain host defenses against *C. albicans*, alleviating infection in these mice. **Third**, the response to *C. albicans* infection can be either candidalysin-dependent or -independent. Candidalysin can induce K$^+$ efflux through plasma membrane perforation, which in turn leads to canonical inflammasome activation, GSDMD cleavage, and macrophage pyroptosis. Thus, this candidalysin- and GSDMD-mediated relay mechanism enables *C. albicans* to escape host immunity via progressive plasma membrane permeabilization and host immune cell lysis. In the absence of candidalysin, hyphae still trigger rupture of the plasma membrane, inflammasome activation, and GSDMD cleavage. This is likely to be mediated by mechanical piercing or other hyphal proteins. It is noteworthy that, as a pore-forming toxin, candidalysin can also induce macrophage cell death through an inflammasome and K + -independent manner[65]. It is well documented that NLRP3-mediated pyroptosis contributes to *C. albicans*-induced damage of mBMDMs[45,46]. Macrophage lysis is dependent on caspase-1, ASC, and NLRP3[46]. Consistently, *C. albicans*-induced cell death is alleviated in NLRP3 inflammasome-deficient macrophages[65]. However, the cell death can not be completely inhibited by disrupting the NLRP3

inflammasome, suggesting that Candidalysin can trigger both the inflammasome-dependent and -independent cell death. Consistent with the role of candidalysin in Candida-induce macrophage lysis and Candida escape from macrophages, in the mouse systemic candidiasis model, we found that mice infected with the candidalysin-deficient (ece1Δ/Δ) *C. albicans* displayed higher survival rates and decreased fungal burden compared with WT *C. albicans*. A previous study showed that, in an oral candidiasis model, the fungal burden was also lower in ece1Δ/Δ *C. albicans* infection compared with wild type[63]. Interestingly, in another study, Swidergall et al. detected increased fungal burden with the candidalysin-deficient *C. albicans*[135]. The discrepancy is likely caused by different experimental conditions in their compared with our experiments. We both revealed that Candidalysin promoted mortality in mouse models of systemic fungal infection. Swidergall et al. focused on neutrophil recruitment. Their data indicated that candidalysin was required for neutrophil recruitment; thus when candidalysin-deficient *C. albicans* ($2 \times 10^5$) were used to infect mice, fungal burden of the kidney at 1 day increased significantly due to reduced neutrophil accumulation. This effect was diminished 4 days post-infection. Our study focuses on the escape of *C. albicans* from infected macrophages, and we inoculated mice with more *Candida albicans* cells ($1 \times 10^6$) and assessed fungal burden 2 days post-infection. In this setup, mice infected with the candidalysin-deficient *C. albicans* displayed decreased fungal burden compared with WT *C. albicans*. **Finally**, yeast-to-hyphal transition and hyphae-associated candidalysin induced both GSDMD-dependent and -independent cell death during *C. albicans* infection. It is plausible that direct membrane disruption by *C. albicans* could help IL-1β release independent of pyroptosis. Moreover, Monteleone et al. recently showed that pro-IL-1β cleavage results in relocation of mature IL-1β to the plasma membrane and its subsequent release via both GSDMD-dependent and -independent pathways[109]. These mechanisms may explain the GSDMD-independent release of mature IL-1β by *C. albicans*-challenged macrophages.

When inflammasome and pyroptosis axis is blocked, *C. albicans*-induced macrophage lysis[46] can not be completely inhibited, suggesting that *C. albicans* can escape from macrophages via both inflammasome-dependent and -independent mechanisms. A previous study suggests a defined order of candidalysin, inflammasome activation, pyroptosis, macrophage lysis, and *C. albicans* escape during *C. albicans*-macrophage interactions[45]. The early phase escape is likely mediated by a host-cell programmed death pathway including candidalysin, inflammasome activation, GSDMD, and pyroptosis. Inflammasome-independent macrophage lysis occurs at later time points and/or higher organism burdens[46]. The later phase *C. albicans* escape may be mediated by candidalysin-dependent but pyroptosis-independent cell death. Alternatively, the polarized growth of hyphae can directly lead to physical rupture of macrophages independent of candidalysin, by piercing of the fungal filaments through the plasma membrane. Finally, disruption of host glucose homeostasis may also contribute to massive macrophage death at the later stage[136,137]. Inflammasome activation can be triggered by glucose starvation in macrophages, which occurs when fungal load increases sufficiently to outcompete macrophages for glucose[136]. Despite the complexity of the host response to *C. albicans*, we clearly established GSDMD as a host component that exacerbates inflammatory responses during *C. albicans*-induced sepsis and can thus be targeted to combat *C. albicans* invasion (Fig. S10).

## Methods

**Materials and Resources**. Most reagents used in this application (e.g. antibodies, chemicals, media, etc.) were purchased from reputable commercial vendors. As such, these reagents have passed a quality control screening process, and this information was provided to us upon purchase. Key resources used in this study are listed in Table S1.

**Mouse strains**. $Gsdmd^{-/-}$ mice (in a C57BL/6 J background) were generated as previously described[53]. $Casp1/11^{-/-}$ (in a C57BL/6NJ background), C57BL/6 J WT, and C57BL/6NJ WT mice were purchased from The Jackson Laboratory (Bar Harbor, ME). Eight- to 12-week-old mice were used in all experiments. Corresponding sex- and age-matched WT mice were used as controls in all experiments performed with KO mice. The Boston Children's Hospital Animal Care and Use Committee approved and monitored all mouse procedures.

**C. albicans strains**. *C. albicans* ATCC MYA-2876 (SC5314)[138] was the wild type used in all experiments. The ece1Δ/Δ strain was a kind gift of William Fonzi[139]. The cph1Δ/Δ, efg1Δ/Δ strain was generated as previously described[61]. For construction of other strains, including the primers used in current study, please see Table S2–S4. *C. albicans* were grown overnight on YPD agar plates at 30 °C, and cells were harvested by centrifugation, washed twice, and resuspended in PBS. For in vitro experiments, *C. albicans* were opsonized with 20% mouse serum (autologous) in PBS for 20 min at 37 °C before infecting BMDMs.

**Primary Cell Culture**. Mouse bone marrow was isolated from femurs and tibias. Bone marrow-derived macrophages (BMDMs) were prepared as described previously[140]. Briefly, total bone marrow cells were isolated and resuspended in culture medium (DMEM supplemented with 10% FBS, 100 U/mL penicillin/streptomycin, and 30 ng/ml recombinant mouse M-CSF). Cells were seeded at $1.5 \times 10^6$/well in a 6-well plate with 2 ml culture medium per well and cultured for 6 days. The medium was changed on day 3, and floating non-adhesive cells were washed away during the medium change. Four to five million cells per mouse were routinely obtained. Microscopic examination confirmed that >95% attached cells were morphologically mature macrophages. For pharmacological treatments, the indicated chemicals or inhibitors were added to the culture medium 1 h prior to *C. albicans* infection. The GSDMD antagonist necrosulfonamide (NSA) was dissolved in dimethyl sulfoxide (DMSO), with DMSO alone used as control.

**C. albicans infection model**. Mice were injected intravenously with *C. albicans* yeast in a 100 μL volume of sterile pyrogen-free PBS ($2 \times 10^5$ or $1 \times 10^6$). Survival was assessed daily for 15 days. To evaluate *C. albicans*-induced sepsis, we used a well-established clinical scoring system[141] in which the appearance, activity, response to stimulus, and eyes were assessed.

**Infection of BMDMs with C. albicans**. Fresh overnight cultures of *C. albicans* yeast were resuspended in PBS and opsonized with 20% mouse serum (autologous) in PBS for 20 min at 37 °C. BMDMs from WT or KO mice ($0.5 \times 10^6$/well) were incubated with *C. albicans* at indicated multiplicities of infection (MOI) with intermittent shaking. After each time period, cells were lysed by adding distilled $H_2O$, and serially diluted aliquots were spread on YPD agar plates. CFUs were counted after incubating the plates overnight at 30 °C. *C. albicans* suspensions without any BMDMs were used as input control. In vitro *C. albicans* killing capabilities were reflected by the decrease in *C. albicans* CFUs after incubation.

**Preparation and infection of human monocyte-derived macrophages (hMDMs)**. Human peripheral blood mononuclear cells (hPBMC) were isolated using a Ficoll density gradient medium (LymphoprepTM, Stem cell technologies: Catalog #07801) following a protocol provided by the manufacturer. Subsequently, human monocytes were isolated from hPBMCs using EasySep™ Human Monocyte Isolation Kit (Stem cell technologies: Catalog #19359) according to the manufacturer's protocol. To differentiate monocytes into hMDMs, the isolated monocytes were resuspended in IMDM GlutaMAX Supplement culture medium (ThermoFisher: 31980-030) supplemented with 10% FBS, and 40 ng/ml recombinant human M-CSF. Cells were seeded at $1.5\times10^6$ in a 6-well plate with 2 ml culture medium per well and cultured for 6 days. The medium was changed on day 3, and floating non-adhesive cells were washed away during the medium change. hMDMs were collected from the plate 6 days after human M-CSF treatment and the cell densities were adjusted to $5 \times 10^5$ cells/ml. Finally, the cells were reseeded in 24 or 96-well plates at a final concentration of $2.5\times10^5$ or $5\times10^4$ cells/well, respectively in IMDM + FBS and incubated overnight. For pharmacological treatment, the indicated chemicals or inhibitors were added to the culture medium 1 h prior to *C. albicans* infection. Macrophage infection experiments were performed in Opti-MEM medium. All blood was drawn from healthy blood donors. All protocols were approved by the Children's Hospital Institutional Review Board and were subjected to annual review.

**Immunoblotting**. At each time point, BMDMs were lysed immediately with cell lysis buffer (2× Laemmli sample Buffer, Bio-Rad, Hercules, CA) containing 2-mercaptoethanol (Bio-Rad) and protease inhibitor cocktail (Cell Signaling). Proteins in the supernatants were precipitated with chloroform/methanol (1:4) and then dissolved in the same lysis buffer. Samples were denatured by boiling at 100 °C for 5 min and transferred to ice. After brief sonication (5–10 s), 15 μl of sample was

subjected to 4–15% gradient SDS-PAGE. Proteins were transferred to polyvinylidene difluoride or nitrocellulose membranes (Millipore, Bedford, MA). Blocking and antibody incubation were performed in PBS containing 0.05% Tween-20 and 5% Blotting-Grade Blocker (Bio-Rad). After blocking the membranes for 1 h, primary antibodies were added in blocking solution at the following dilutions: mouse anti-GSDMD antibody (ab209845, 1:500; Abcam, Cambridge, UK), mouse anti-IL-1-β/IL1F2 antibody (AB-401-NA, 1:1000; R&D Systems, Minneapolis, MI), antitotal IL-1β antibody (D3H1Z) (#12507, 1:1000; Cell Signaling Technology, Danvers, MA), anti-cleaved-IL-1β (Asp117) antibody (52718 S, 1:1000; Cell Signaling Technology), Anti-mouse NLRP3 /Cryopyrin antibody (ARG40539, 1:1000; Arigo), Anti-mouse GAPDH (D16H11) antibody (5174 S, 1:1000; Cell Signaling Technology), Anti-mouse Caspase-1 (E2Z1C) (24232 S, 1:1000; Cell Signaling Technology), Antimouse Cleaved Caspase-1 (Asp296) (E2G2I) (89332 S, 1:1000; Cell Signaling Technology), Antimouse Caspase-1 (p20) (AG-20B-0042-C100, 1:1000; AdipoGen), anti-human Gasdermin D (E9S1X) antibody (39754 S, 1:1000; Cell Signaling Technology), anti-human Cleaved Gasdermin D (Asp275) (E7H9G) (36425 S, 1:1000; Cell Signaling Technology), anti-human Cleaved-IL-1β (Asp116) (D3A3Z) (83186 S, 1:1000; Cell Signaling Technology), anti-human Cleaved Caspase-1 (Asp297) (D57A2) (4199 S, 1:1000; Cell Signaling Technology), or anti-β-actin antibody (A2228, 1:1000; Sigma Aldrich, St. Louis, MO). Primary antibody incubations were at 4 °C overnight. After washing, goat HRP-conjugated secondary antibodies (Santa Cruz Biotechnology Inc., Dallas, TX) were added (1:10,000) in blocking solution and the membrane incubated at room temperature (RT) for 1 h. Immunoreactivity was detected with Super Signal West Femto substrate (Thermo Scientific, Rockford, IL) using an ImageQuant LAS-4000 (GE, Fairfield, CT). Densitometry was performed using ImageJ software Gel Analyzer plug-in[142].

**Time-lapse imaging.** For analysis of lytic macrophage death induced by *C. albicans,* as well as escape of *C. albicans* from macrophages, BMDMs were seeded in 6-well plates at $5 \times 10^5$ cells/well. To stain non-viable cells, propidium iodide (PI; 200 ng/ml; Sigma Aldrich) was added before BMDMs were infected with WT or mutant (*ece1Δ/Δ, cph1Δ/Δ efg1Δ/Δ*) *C. albicans* at MOI 10. Plates were placed under a Nikon Ti Eclipse inverted microscope. At least three independent fields/well were imaged using a brightfield and DsRed channel every 5 min at 20× magnification for a maximal time span of 12 h. DsRed filter channel images were processed using NIS-Elements software. The total number of macrophages per field was calculated in the picture of the first time point. The number of PI-positive cells was determined by counting macrophages manually.

**PI staining of hMDMs.** To count the percentage of PI-positive macrophages after infection, propidium iodide (PI; 200 ng/ml; Sigma Aldrich) was added before hMDMs were infected with WT or mutant (*ece1Δ/Δ*) *C. albicans* at MOI 10. Pictures were taken at 1 h, 3 h, 6 h, 9 h, and 12 h with fluorescence microscope. At least three independent fields/well were imaged. The total number of macrophages per field was calculated in the picture of the first time point (1 h). The number of PI-positive cells was determined by counting macrophages manually.

**Lactate dehydrogenase (LDH) cytotoxicity assay.** LDH release was assayed using the Pierce™ LDH Cytotoxicity Assay Kit (Thermo Fisher Scientific) following the manufacturer's protocol. LDH release was assayed using the CytoTox 96® Non-Radioactive Cytotoxicity Assay (Promega, Madison, WI) following the manufacturer's protocol. Culture medium was collected and centrifuged at $400 \times g$ for 5 min to remove cell debris. LDH release into the medium was measured at OD 490. Relative LDH release was expressed as the percentage LDH activity in supernatants of cultured cells (medium) compared to total LDH (from media and cells) and used as an index of cytotoxicity.

**IL-1β production by *C. albicans*-challenged mBMDMs and hMDMs.** Mouse BMDMs and hMDMs were infected with *C. albicans* (MOI = 10) for the indicated time periods. IL-1β in culture supernatants was measured with a mouse IL-1β/IL-1F2 Quantikine ELISA Kit or human IL-1 beta/IL-1F2 DuoSet ELISA Kit (R&D Systems). To measure intracellular IL-1β, BMDMs were collected and lysed with PBS/0.5% Triton X-100. Proteins in the supernatants were precipitated with chloroform/methanol (1:4). Unattached cells (<5%) were removed by washing twice with PBS. The levels of IL-1β in each sample were measured using the ELISA kit as described above.

**IL-1β and other cytokine levels in *C. albicans*-challenged mice.** WT, *Gsdmd*[-/-], and *Casp-1/11*[-/-] mice were intravenously challenged with $1 \times 10^6$ CFU *C. albicans*, and the level of IL-1β in the serum was assessed at the indicated time points. For assessment of in vivo IL-1β, whole blood was drawn by cardiac puncture and collected in Eppendorf tubes. The blood was incubated at RT for 15–30 m to allow a clot to form. The clot was removed by centrifuging at $2,000 \times g$ for 10 m. The resulting serum was collected and filtered through a 0.22 μm filter. Serum IL-1β levels were analyzed by multiplex through Eve Technologies. IL-6, TNF-α, MIP-2, and IL-10 levels were detected by multiplex using MILLIPLEX® MAP Mouse High Sensitivity T Cell Magnetic Bead Panel (MHSTCMAG-70K, Millipore)

**Tissue fungal burden.** Kidney, lung, liver, and spleen samples were homogenized using 40 μm cell strainers, diluted with saline, and quantitatively cultured overnight on YPD agar at 30 °C. The results are expressed as log CFUs per organ.

**Histopathology.** WT, *Gsdmd*[-/-], and *Casp-1/11*[-/-] mice were intravenously challenged with $1 \times 10^6$ CFU WT or *ece1Δ/Δ C. albicans*. Mice were euthanized by asphyxiation by inhalation of $CO_2$ 2 days after *C. albicans* injection. Kidneys were dissected, fixed in 10% neutral buffered formalin (Sigma Aldrich), and then embedded in paraffin. Paraffin-embedded sections (~6 μm thick) were stained with hematoxylin and eosin (H&E) and examined by light microscopy.

**Grocott's methenamine silver stain and Periodic Schiff-Methenamine Silver (PASM) stain.** Grocott's methenamine silver (GMS) staining was used to visually identify fungi in infected kidneys and was performed at the HMS Rodent Histopathology Core facility. Briefly, kidney sections were oxidised with 4% chromic acid at RT for 1 h, treated with 1% sodium metabisulphite at RT for 1 min, and stained with 60 °C silver solution for 20 min or until sections turned yellowish-brown. Fungi were stained dark brown. PASM staining was also used to identify fungi in infected kidneys and was performed using a PASM staining kit (Servicebio, G1059) following the manufacturer's protocol. Kidney sections were treated with periodic acid-silver metheramines (PASM) staining solution A overnight, acidized with PAS staining solution B for 15-20 mins, and stained with 60 °C silver solution until sections turned yellowish-brown. Fungi were stained black or dark brown.

**Detection of macrophages containing trapped *C. albicans* in kidneys of infected mice.** Mice (WT and *Gsdmd*[-/-]) were intravenously challenged with $3 \times 10^5$ CFU WT (GFP[+]) or candidalysin-deficient (*ece1Δ/Δ*) *C. albicans*. Three days after *C. albicans* infection, kidneys were homogenized using 40 μm cell strainers. Red blood cells were lysed with 1 ml ACK lysis buffer (Gibco) for 5 min at room temperature. Cells were then stained with PE-Cy7-conjugated CD45 (BioLegend), PE-conjugated F4/80 (BioLegend), and APC-conjugated CD11b (BioLegend) antibodies for 30 min. For cells isolated from WT (GFP[+]) *C. albicans*-infected mice, macrophages containing trapped *C. albicans* were recognized by GFP fluorescence. For cells isolated from candidalysin-deficient (*ece1Δ/Δ*) *C. albicans*-infected mice, the samples were fixed and permeabilized (BD Biosciences) after surface marker staining, and then stained with FITC-conjugated anti-*C. albicans* antibody (thermofisher) for 1 h. Flow cytometry was performed on the LSRFortessa (BD Biosciences) instrument. Flow cytometry data were analyzed with FlowJo software.

**C. albicans infection of NSA-treated mice.** Mice were intravenously challenged with $2 \times 10^5$ CFU *C. albicans* 1 h after intraperitoneal injection of sesame oil alone (control) or NSA (dissolved in sesame oil, 20 mg/kg body weight). Challenged mice received daily NAS treatment (20 mg/kg body weight) and were monitored for 15 days. Controls were injected with sesame oil during the same time period. Survival rates were analyzed using Kaplan-Meier survival curves and log-rank testing. Clinical scores of untreated and NSA-treated mice were assessed 24 h after *C. albicans* challenge. Kidneys (weight and size) were examined 2 weeks after *C. albicans* infection. Histopathologic assessment of the kidneys, GMS staining of *C. albicans* in the kidney, and measurement of fungal burden in the kidneys were all conducted 2 days after *C. albicans* injection. Shown are representative H&E stained sections of kidney tissues.

**C. albicans infection of anakinra-treated mice.** Mice were intraperitoneally injected with 2 doses of anakinra (diluted in PBS, 100 mg/kg body weight per dose) administered 24 h apart. The first dose was applied 1 h before and the second dose was applied 23 h after *C. albicans* infection. Challenged mice were monitored for 15 days. Survival rates were analyzed using Kaplan-Meier survival curves and log-rank testing. Clinical scores of untreated and anakinra-treated mice were assessed 24 h after *C. albicans* challenge. Kidneys (weight and size) examination, histopathologic assessment of the kidneys, PASM staining of *C. albicans* in the kidney, and measurement of fungal burden were all conducted 2 days after *C. albicans* injection.

**Statistical analyses.** For most experiments, comparisons were conducted with a two-tailed, unpaired, Student's *t*-test or two-way ANOVA analysis. Values shown in each figure represent mean ± SD. A p-value<0.05 was considered statistically significant. *$p < 0.05$, **$p < 0.01$, ***$p < 0.001$, ****$p < 0.0001$. In vitro experiments were repeated at least three times. For in vivo experiments, for reliable statistical analysis, at least five mice from each genotype or treatment group were utilized for each data point. This number was chosen based on a power analysis conducted using Simple Interactive Statistical Analysis (SISA; http://home.clara.net/sisa/sampshlp.htm). Based on our preliminary data and experience, we set a standard deviation (SD) to 10% of the average. To detect a 20% difference (average 1 = 100, average 2 = 80, SD1 = 10, SD2 = 8, allocation ratio=1) with a power level of 90% (90% chance to discover a real difference in the sample) and an alpha of 0.05, we needed to use $n = 5$ mice from each genotype/treatment group for double-sided power (direction of effect unknown). Animal experiments were performed independently at least three times and the data

were pooled and analyzed together. No samples or animals subjected to successful procedures and/or treatments will be excluded from the analysis. For survival analyses, we used survival data to generate Kaplan-Meier survival curves and performed comparisons between groups by log-rank analysis using Prism (GraphPad Software, La Jolla, CA).

**Reporting Summary**. Further information on research design is available in the Nature Research Reporting Summary linked to this article.

## Data availability

All data needed to evaluate the conclusions in the paper, including data associated with main figures and supplementary figures, are available within the article or in the Supplementary Information. Additional data related to this paper may be requested from the authors. Source data are provided with this paper.

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

## Acknowledgements

We thank Drs. Li Chai, Leslie Silberstein, John Manis for helpful discussions. We also thank the Dana-Farber/Harvard Cancer Center in Boston, MA, for the use of the Rodent Histopathology Core, which provided paraffin embedding and staining services. The Dana-Farber/Harvard Cancer Center is supported in part by an NCI Cancer Center Support Grant # NIH 5 P30 CA06516. H. Luo is supported by National Institutes of Health grants (R01 AI142642, R01 AI145274, R01 AI141386, R01HL092020, and P01 HL095489).

## Author contributions

Conceptualization was inputted from H.R.L., H.K., X.D. and F.M.; Methods were designed by H.R.L., H.K., X.D., W.Q., R.G. J.L. and J.R.K.; Experiments were conducted by X.D., H.K., R.G., F.L., A.K., M.A., W.Q., J.L., T.B., X.X., W.H., N.L., C.Z., X.Z., H.Y. and L.Z.; Analysis was performed by H.R.L., F.M., J.R.K., F.L., H.K., X.D., A.K., M.A., T.B., X.X., W.H., N.L., W.Q., J.L., C.Z., X.Z., H.Y. and L.Z.; Work was supervised by H.R.L., J.R.K., Y.J. and F.M.; Original manuscript was written by H.R.L., H.K., X.D. and A.K. and was revised by H.R.L., H.K., X.D., J.R.K., M.A. and F.M. All coauthors read, reviewed, and approved the manuscript.

## Competing interests

All authors declare no competing interests.
