## [Peer Review File · Nature Communications]

Inflammasome-mediated GSDMD activation facilitates escape of *Candida albicans* from macrophagesREVIEWER COMMENTS

Reviewer #1 (Remarks to the Author):

Kambara et al. describe an interesting observation that GSDMD-dependent pyroptosis may promote *Candida* escape from macrophages and contributes to disseminated candidiasis. Incubation of murine macrophages with *C. albicans* leads to IL-1 β release in a GSDMD-dependent and -independent fashion, and disruption of GSDMD prevents fungal escape while maintaining IL-1 β -mediated anti-*Candida* immune defences. The yeast-to-hypha morphogenetic transition and K⁺ efflux is crucial for GSDMD activation and cell death. They also address the role of the toxin candidalysin in driving both GSDMD-dependent pyroptosis and GSDMD-independent lytic cell death. Finally, they show that administration of the GSDMD inhibitor necrosulfonamide (NSA) alleviates *C. albicans* infections in mice. Overall, the work is interesting and well written. While most of the approaches/findings have been published elsewhere regarding macrophage pyroptosis in the context of *C. albicans* (and candidalysin) infections, this is the first work that looks at GSDMD in detail. However, the ms lacks several important controls, lacks a coherent/complete mechanism, is inadequately discussed in places, and is speculative in several areas. As such, while the work has great promise, the ms currently provides only an incremental advance rather than a step-wise change in our understanding of *C. albicans*-induced inflammasomes/pyroptosis/cell death. One could also summarise by saying the data do not currently provide a clear, new picture but do provide the foundations to obtain a clear, new picture. The detailed review that follows is so the authors can convert this interesting paper into something more solid and substantial. I hope the authors take this on board and undertake the necessary work and modifications.

1. Controls and data interpretation

The *ece1* null from Bill Fonzi was used. Candidalysin is generated from the processing of the parent protein Ece1 into 8 (or more) peptides (including candidalysin). Thus, some of the responses observed could be driven by these peptides and not candidalysin. The authors need to undertake experiments using both the candidalysin-only null *C. albicans* strain and with synthetic candidalysin to confirm that all these responses are due to candidalysin. The authors also do not fulfil Koch's postulates and should also undertake experiments using the ECE1 revertant strain. All these mutants/strains and synthetic candidalysin are readily available from the Hube and Naglik labs. This alone requires a considerable amount of work but is essential to verify many of the ms's conclusions. Furthermore, the authors themselves state that "hyphal proteins other than candidalysin may facilitate GSDMD cleavage and plasma membrane perforation, leading to host cell death etc." Hence, the above experiments will also exclude (or more excitingly, identify) a role for the other Ece1 peptides in the observed responses.

The ms states: "Yeast-locked *C. albicans* failed to trigger GSDMD cleavage and IL-1 β production by BMDMs (Fig.4A-B) (lanes 229-230)". Again, in Table Fig.S4: "Pro-IL-1 β is expressed but cannot be processed". Two concerns here. The data in Figure 4A show a band corresponding to the mature IL-1 β in both cell lysate and supernatants. Perhaps, the total amount of processed IL-1 β (intracellular and extracellular) in Fig.4A-B is significant compared to untreated macrophages? Also, Fig. 4C shows that the yeast-locked mutant induces a moderate secretion of IL-1 β (50-100 pg/ml) compared to hyphae. However, untreated macrophages are not shown and it's difficult to determine/understand whether the amount of IL-1 β release following stimulation with the yeast-locked mutant is significant. Similarly, in Fig. 4G LDH release in untreated macrophages is missing. Does KCl 50 mM decrease LDH release in macrophages stimulated with the yeast-locked strain in Fig.6E? Furthermore, LDH release from untreated macrophages in the presence or absence of KCl (Fig.6E) and from untreated WT and *gsdmd*^{-/-} macrophages (Fig.5H) needs inclusion. Thus, several controls need to be added.

Kasper et al. recently shown that candidalysin induces macrophage cell death through an inflammasome and K⁺-independent manner (PMID: 30323213). However, the authors show that candidalysin participates in pyroptosis (Fig.5) and candidalysin-dependent cell death is also partially K⁺ efflux-dependent (Fig.6E-F). How can the authors reconcile their data with the previous observations? Additional discussion is required and differences and similarities between studies need

highlighting to make the message clearer.

The authors show that IL-1 β processing and GSDMD cleavage are dependent on caspase-1/11 using caspase-1/11^{-/-} mice and the caspase-1 inhibitor VX765. However, they do not show caspase-1/11 expression and activation (for instance, in WT vs. KO macrophages). Although *gsdmd*^{-/-} macrophages show an increase in the intracellular level of mature IL-1 β , suggesting that IL-1 β cleavage still occurs, the detection of caspase-1/11 activity or activation would be helpful in both Figure 1 and 2. Also, IL-1 β release was impaired in caspase-1/11^{-/-}, *gsdmd*^{-/-} macrophages or when the VX765 chemical inhibitors were used. The specificity or the global effect of the treatment should be addressed, for instance by looking at other cytokines.

The authors show that KCl treatment (Fig.6) inhibits GSDMD cleavage and cell death. Does KCl inhibit caspase-1 activation or IL-1 β processing and release? Is the effect specific for IL-1 β ? More data are required to interpret the findings properly.

Dynamics (time course data) are required. Previous, initial studies have simplified the message that inflammasome activation = pyroptosis. More recent work including this submission indicates that this message is clearly wrong. The relationship between inflammasome activation, pyroptosis and GSDMD activation in *Candida* infection is complex. It appears that inflammasome activation can be independent of pyroptosis but that a key event of pyroptosis is GSDMD activation. There is also a combination of two pore forming events: GSDMD and candidalysin. These may act separately or together. However, all the above depends on the time point of analysis (as demonstrated by the Hube and Traven groups) and what is knocked out. The dynamics studies show that there is a defined order of inflammasome/pyroptosis/cell death during *C.alb*-macrophage interactions (early/late). Thus, undertaking time course assays are important in data interpretation. Note: two publications from the Traven group have not been cited or discussed – see below.

Conceptual. A schematic diagram is also required. Regarding this, the authors should consider that the lack of pyroptosis induction via GSDMD may relate to the early phase escape route by *C. albicans* (see Kasper et al). The two later phase escape routes may be mediated by candidalysin and physical forces of hyphae (associated with metabolism re: Traven group data). Thus, if you inhibit/KO GSDMD, you may remove one escape route, while at the same time the macrophage can better deal with candidalysin-mediated damage, while protective IL-1 β driven inflammation is still induced by candidalysin. Note: candidalysin can induce the inflammasome and pyroptosis-independent cell lysis, and pyroptosis can also be inflammasome-independent (Kasper et al and others). Also note that if you KO the pyroptosis axis by knocking down the inflammasome, you will still see lysis – this means that lysis induced by candidalysin will happen independent of pyroptosis. The Traven lab stressed that this depends on the time point: early phase = pyroptosis-mediated cell death and inflammasome activation; later phase = candidalysin-mediated cell death and inflammasome activation; very late phase = escape by physical forces/cell death and cell death by glucose consumption. Thus, as mentioned, this is complex, and the authors need to discuss their findings more fully and in context regarding these different processes. This is not possible unless they undertake time course studies (see dynamics issue above).

Although the authors show that NSA treatment alleviates *Candida* infection and therefore the inflammasome response should not be dramatically impaired, a recent publication (PMID: 31209100, Rashidi et al, - also not cited in this submission) has demonstrated that NSA pre-treatment blocks inflammasome priming and caspase-1 activation independent of GSDMD. The Rashidi findings conflict with the original paper (Rathkey et al. 2018), which is the reference cited in this ms as the main justification for using NSA. Perhaps, the authors should address this point and verify whether NSA is affecting any priming and/or activation event in their system following *Candida* infection. More controls required.

The authors used opsonised *C. albicans* for their experiments. This is different to other publications since opsonised and unopsonised *C. albicans* can give different results due to the role of complement in fungal immunity (see multiple publications by the Zipfel lab). The authors should undertake multiple

experiments with unopsonised *C. albicans* to confirm they obtain the same data as with opsonised *C. albicans* to address the role of complement. A better discussion of this is also required.

2. Mechanism and novelty

Most of the non-GSDMD data has previously been published (for *C. albicans* and candidalysin). Regarding the GSDMD data, the authors need to discuss/determine how GSDMD fits functionally and mechanistically in the recognised inflammasome/pyroptosis processes (see above).

Although K⁺ efflux has been demonstrated as an important event for GSDMD activation, the manuscript lacks depth with respect to possible mechanisms of inflammasome/Caspase-1/GSDMD activation in response to *C. albicans*. As the mechanisms by which *C. albicans* regulate inflammasome and GSDMD activation in their model is not described, the manuscript will greatly benefit from at least discussing how this might happen (see Point 1 comments).

In vitro stimulation of macrophages with the GSDMD inhibitor NSA specifically inhibits IL-1 β release and pyroptosis following LPS+nigericin stimulation. Does NSA inhibit macrophage cell death and *Candida* escape from macrophages? What about the inflammatory response following *Candida* stimulation in the presence of NSA (does NSA specifically control IL-1 β ? What about other inflammatory cytokines?). Also, the authors show that NSA contributes to increased resistance to *Candida* infection in vivo. A discussion on this finding is missing and the overall conclusion based on the use of NSA lacks a mechanism. This needs to be addressed.

Work is with GSDMD. What about the role of other Gasdermins? Discussion.

The authors use a full *gsdmd* KO mouse. Therefore, apart from macrophages, other cells (immune or otherwise) may well be important for the readouts observed in the in vivo work, given that most hematopoietic and non-hematopoietic cells tested have a role in *C. albicans* infection in vivo. Discussion.

Authors show that significant macrophage death still occurred in the absence of GSDMD and conclude that *C. albicans*-induced macrophage death could be mediated by both GSDMD-dependent pyroptosis and GSDMD-independent mechanisms. OK, this is new, but no real mechanism is provided.

Candidalysin disruption in *C. albicans* did not completely abolish IL-1 β release from BMDMs, indicating the presence of candidalysin independent IL-1 β production. Yes, but no real mechanism is provided.

They indicate that GSDMD disruption paradoxically improved host anti-*C. albicans* responses, suggesting that *C. albicans* may have hijacked this defense mechanism to improve its survival in infected hosts. OK, but no mechanism, rather speculation.

The main novelty of the ms is the notion that GSDMD facilitates escape from macrophages. However, *C. albicans* escape has previously been quantified (Kasper et al) and no difference in piercing or hyphal length was observed in human primary macrophages. All data in the current submission is based on mouse macrophages (where differences in hyphal length was observed), but mouse and human macrophages behave differently. Conclusions/findings should be supported using human macrophages. Furthermore, the authors indicate that candidalysin mediates escape via GSDMD activation and pyroptosis, but that candidalysin also does this via direct membrane lysis. Are the authors indicating these processes are connected or independent? More discussion required.

3. References and lack of discussion

One of the authors (Kanneganti) has previously published data on GSDMD cleavage by *C. albicans* (JBC PMID: 33109609). However, this publication is not cited or discussed in the current submission, which reduces novelty somewhat and raises the question of why wasn't this publication cited? Irrespective, in Fig.2 of the JBC paper, it appears that activation of Casp-1 (p20) is slightly reduced in

gsdmd KO mice (however, no quantification is available). Therefore, in the current submission, the authors should also show Caspase-1 activation where relevant as controls (also relevant for Point 1 above).

The authors show that KCl supplementation reduces GSDMD activation and macrophage cell death (Fig.6), but inhibition of macrophage pyroptosis following KCl supplementation has been already shown following *Candida* infection (PMID: 30131363). Importantly, treatment of KCl also reduces ASC speck formation and the number of CFUs, thus suggesting fungi can use pyroptosis to evade killing by macrophages. However, the authors do not cite or discuss this paper. Needs including with proper discussion.

Important publications and concepts (dynamics) from the Traven Lab have been omitted (see above). "Metabolic competition between host and pathogen dictates inflammasome responses to fungal infection" (PMID: 32750090), and "Glucose homeostasis is important for immune cell viability during candida challenge and host survival of systemic fungal infection" (PMID: 29719235). Should be included in context and discussed.

In the mouse systemic candidiasis model, the authors found that mice infected with the *ece1* deficient *C. albicans* displayed higher survival rates and decreased fungal burden compared with WT *C. albicans*. While the Swidergall reference is cited, the data are in contract with that ms, which found increased fungal burden with the *ece1* deficient *C. albicans*. This is not discussed. How do the authors explain this?

C. albicans-dependent activation of the inflammasome/pyroptosis is complex. The authors should consider recent reviews on the topic and determine how their work fits into this (e.g.: PMID: 33293167; PMID: 24967821; PMID: 33119702), none of which have been cited.

4. Minor

The title is rather misleading and needs modifying, as they find that cell death is promoted by both inflammasome-dependent and -independent mechanisms. Also, significant macrophage death occurred in the absence of GSDMD. Furthermore, they find candidalysin-dependent and independent mechanisms, suggesting GSDMD is not the factor 'dictating' escape of *C. albicans*.

Abstract. Candidalysin is not ON hyphae, it's secreted. Change to "expressed by hyphae".

The ms states: "For in vitro experiments, the GSDMD antagonist necrosulfonamide (NSA) was dissolved in dimethyl sulfoxide (DMSO), with DMSO alone used as control (lanes 554-555)". Did the authors mean in vivo? Consider amending.

The authors suggest that pharmacologic inhibition of GSDMD may offer a new therapeutic strategy for *C. albicans* infection. This is way too early to state. Given the lack of mechanism and the significant experimentation the authors clearly need to undertake, all such comments should to be removed throughout the ms.

5. Direct comment to sentences of the paper

"Mice lacking key inflammasome components are hypersusceptible to *Candida* infection"
That depends on the type of *Candida* infection. Inflammasome activation can be beneficial in candidiasis, and detrimental i.e. during VVC (immunopathology). The authors need to be more precise. Furthermore, every candidemia patient is different as will be the 'mechanisms' of immunity, thus personalized medicine is key.

"Our results have shown that inflammasome-mediated GSDMD activation dictated *C. albicans* escape from macrophages"

Disagree, and it contradicts some of the data and many statements in the ms. This conclusion should be altered as does the title.

"IL-1b release was significantly reduced when macrophages were treated with candidalysin-deficient *C. albicans*" and "However, candidalysin disruption in *C. albicans* did not completely abolish IL-1b release from BMDMs, indicating the presence of candidalysin-independent IL-1b production"
That should be pyroptosis. This is likely to be mediated by any other factors of the fungus that provide both signals to the inflammasome.

Many of the key findings/statements in the ms have been shown before, predominantly in the Kasper et al paper. This is not always recognized by the authors. These include:

"Infection with candidalysin-deficient *C. albicans* also induced significantly less LDH release, again suggesting that candidalysin partially contributed to macrophage death"

"Similarly, *C. albicans* escape from macrophages and *C. albicans*-induced macrophage death were mediated by both candidalysin-independent and -dependent mechanisms ..."

"As a pore-forming toxin, candidalysin may facilitate *C. albicans*-induced rupture of the plasma membrane and macrophage lytic death, which enable *C. albicans* to escape from host cells "

"The hyphae from candidalysin-deficient *C. albicans* could push and stretch the macrophage plasma membranes, and yet these infected macrophages were PI negative, suggesting that candidalysin was indeed critical for facilitating disruption of host plasma membrane integrity"

"It is noteworthy that candidalysin-deficient *C. albicans* could eventually break the plasma membrane, contributing to candidalysin-independent *C. albicans* escape. This process may be mediated by mechanical piercing or other hyphal proteins, ..."

"*C. albicans*-induced macrophage death likely mediated by both inflammasome-dependent pyroptosis and inflammasome-independent mechanisms"

"...inflammasome-independent macrophage death was also facilitated by candidalysin and hyphae formation"

Regarding "This process was also suppressed in *Gsdmd* $-/-$ BMDMs, demonstrating that GSDMD was important for both candidalysin-independent and -dependent IL-1b release (Fig.5D). " There maybe candidalysin-mediated, GSDMD-independent IL-1b release due to candidalysin pore formation, but if you have both, IL-1 β release is much higher likely because the other stimuli provided by *C. albicans* are poor inducers of inflammasome activation while candidalysin will cause rapid K efflux. Note: escape due to physical forces are independent of GSDMD.

" however, macrophages infected with candidalysin-deficient mutants still released significant amounts of IL-1b, suggesting that other fungal components (e.g., β -glucans) could activate inflammasomes and process IL-1b during *C. albicans* infection."

See above: the time point is important. Early events = pyroptosis = candidalysin independent.

Reviewer #2 (Remarks to the Author):

This is a great manuscript from Kambara and colleagues showing a novel link between GSDMD activation and the intracellular escape of *Candida albicans*. The data are very clear and convincing, the observation novel and important and I have almost no comments. However, my only reservation is the in vivo relevance of this phenotype. The authors show a clear in vivo phenotype in the GSDMD mice and then go onto to reveal the mechanism using BMDMs. In vivo support for their hypothesis is given at the end with the GSDMD inhibitors (but essentially this is reproducing the the KO phenotype). What is not clear is whether the mechanism derived from the in vitro studies explains the in vivo phenotype. For example, there is evidence from several models (including zebrafish..see work from Rob Wheeler) that *Candida* hyphenation inside macrophages in vivo is a rare event. Also the impact of tissue matrix on macrophages responses etc. (also shown in in vitro work using 2d vs 3d systems). Thus I would like to see some formal in vivo evidence supporting the hyphae, candidalysin, GSDMD model. Also, the survival of GSDMD deficient mice infected with the ece mutant should be shown (surprising this was not, as this would also help validate their system),

Reviewer #3 (Remarks to the Author):

This study reports that unlike caspase-1/11 loss, deletion of the inflammatory caspase substrate GSDMD protects against Candida-induced sepsis and this likely occurs as a consequence of reduced host cell death facilitating increased killing of intracellular Candida and subsequent decreased fungal dissemination. This finding is unexpected as deletion of caspase-1/11, which cleave GSDMD, has the opposite effect and increases animal susceptibility to Candida infection. It is also of significant pharmaceutical interest, as it points towards GSDMD as being a bona fide therapeutic target in this condition. The experiments are well performed using relevant genetically targeted mice or GSDMD inhibitor (NSA), the results clear, and the conclusions supported by the data presented.

I only have a few relatively minor questions for the authors;

1. Figure 1 and Figure 7. The number of times the in vivo experiments were independently repeated to confirm the results presented should be clearly noted in the figure legend for readers.
2. Does the reduced killing capacity of GSDMD deficient macrophages allow them to generate more inflammatory cytokines (e.g. TNF, IL-6), and if so, could this also contribute to anti-fungal responses?
3. The authors suggest that caspase-1 mediated IL-1beta activation and release, which still occurs in GSDMD deficient cells, explains why caspase-1/11 and GSDMD loss have the opposite phenotype, despite both being required for pyroptosis. Can the authors prove this by, for example, using neutralising IL-1beta antibodies in infected GSDMD KO mice to reverse the protective phenotype?
4. Although NSA targets GSDMD it can also act independent of GSDMD to block pyroptotic killing (e.g. DOI: 10.4049/jimmunol.1900228), and the authors should acknowledge this.

Point-by-point responses NCOMMS-21-03664-T

We appreciate the conceptual recognition of the importance of our work by the reviewers and are pleased with the conclusions that "the work is interesting and well written", "this is the first work that looks at GSDMD in detail" (Reviewer #1), "this is a great manuscript" "the data are very clear and convincing, the observation novel and important and I have almost no comments" (Reviewer #2), "The experiments are well performed, the results clear, and the conclusions supported by the data presented" (Reviewer #3). We thank the reviewers for raising insightful comments that helped us improve the study and the manuscript substantially. We revised our manuscript by closely following these suggestions. We performed additional experiments/analyses in the past 5 months and added a significant number of new results (Fig.1a, Fig.2h-j, Fig.4c and g, Fig.6d, Fig.8e, Fig.9g, Fig.10, Fig.S1a and g, Fig.S2, Fig.S3, Fig.S4c-i, Fig.S5, Fig.S6, Fig.S7, Fig.S8, and Fig.S9) to address the concerns raised by the reviewers. Specific responses are as follows (changes/new sections are marked red in the revised manuscript):

Reviewer 1

Kambara et al. describe an interesting observation that GSDMD-dependent pyroptosis may promote *Candida* escape from macrophages and contributes to disseminated candidiasis. Incubation of murine macrophages with *C. albicans* leads to IL-1b release in a GSDMD-dependent and -independent fashion, and disruption of GSDMD prevents fungal escape while maintaining IL-1b-mediated anti-*Candida* immune defences. The yeast-to-hypha morphogenetic transition and K⁺ efflux is crucial for GSDMD activation and cell death. They also address the role of the toxin candidalysin in driving both GSDMD-dependent pyroptosis and GSDMD-independent lytic cell death. Finally, they show that administration of the GSDMD inhibitor necrosulfonamide (NSA) alleviates *C. albicans* infections in mice. Overall, the work is interesting and well written. While most of the approaches/findings have been published elsewhere regarding macrophage pyroptosis in the context of *C. albicans* (and candidalysin) infections, this is the first work that looks at GSDMD in detail. However, the ms lacks several important controls, lacks a coherent/complete mechanism, is inadequately discussed in places, and is speculative in several areas. As such, while the work has great promise, the ms currently provides only an incremental advance rather than a step-wise change in our understanding of *C. albicans*-induced inflammasomes/pyroptosis/cell death. One could also summarise by saying the data do not currently provide a clear, new picture but do provide the foundations to obtain a clear, new picture. The detailed review that follows is so the authors can convert this interesting paper into something more solid and substantial. I hope the authors take this on board and undertake the necessary work and modifications.

1. Controls and data interpretation

The *ece1* null from Bill Fonzi was used. Candidalysin is generated from the processing of the parent protein Ece1 into 8 (or more) peptides (including candidalysin). Thus, some of the responses observed could be driven by these peptides and not candidalysin. The authors need to undertake experiments using both the candidalysin-only null *C. albicans* strain and with synthetic candidalysin to confirm that all these responses are due to candidalysin. The authors also do not fulfil Koch's postulates and should also undertake experiments using the ECE1 revertant strain. All these mutants/strains and synthetic candidalysin are readily available from the Hube and Naglik labs. This alone requires a considerable amount of work but is essential to verify many of the ms's conclusions. Furthermore, the authors themselves state that "hyphal proteins other than candidalysin may facilitate GSDMD cleavage and plasma membrane perforation, leading to host cell death etc." Hence, the above experiments will also exclude (or more excitingly, identify) a role for the other Ece1 peptides in the observed responses.

- We appreciate these great suggestions. However, we feel this set of experiments is outside the scope of this study, which focuses on the unexpected role of GSDMD (compared to caspases1/11) in candidiasis. The editor agreed and instructed us not to pursue these experiments at this stage.

The ms states: "Yeast-locked *C. albicans* failed to trigger GSDMD cleavage and IL-1b production by BMDMs (Fig.4A-B) (lanes 229-230)". Again, in Table Fig.S4 (Fig.S10 in the revised manuscript): "Pro-IL-1b is expressed but cannot be processed". Two concerns here. The data in Figure 4A show a band corresponding to the mature IL-1b in both cell lysate and supernatants. Perhaps, the total amount of processed IL-1b (intracellular and extracellular) in Fig.4A-B is significant compared to untreated macrophages? Also, Fig. 4C shows that the yeast-locked mutant induces a moderate secretion of IL-1b (50-100 pg/ml) compared to hyphae. However, untreated macrophages are not shown and it's difficult to determine/understand whether the amount of IL-1b release following stimulation

with the yeast-locked mutant is significant. Similarly, in Fig. 4G LDH release in untreated macrophages is missing. Does KCl 50 mM decrease LDH release in macrophages stimulated with the yeast-locked strain in Fig.6E (Fig.8e in the revised m.s.)? Furthermore, LDH release from untreated macrophages in the presence or absence of KCl (Fig.6E) and from untreated WT and *gsdmd*^{-/-} macrophages (Fig.5H) (Fig.6d in the revised m.s.) needs inclusion. Thus, several controls need to be added.

- a) It is a good observation by the reviewer. Thanks! We modified the related sentences in the revised manuscript (Page 9 and Fig.S10). Pro-IL-1 β was not expressed in untreated macrophages, thus we could not detect any m-IL-1 β (intracellular or extracellular). We agree that, based on our results, yeast-locked *C. albicans* can still release a very small amount of m-IL-1 β (Fig.4a-b and Fig.4c) and induce a small level of cell death (Fig.4G) in macrophages.
- b) Sorry for the omission. We added the "untreated macrophages" controls in Fig.4c and Fig.4g. Untreated WT and *Gsdmd*^{-/-} macrophages did not produce pro-IL-1 and thus the secretion of m-IL-1 β was almost undetectable.
- c) Although *C. albicans*-induced LDH release was largely hyphae dependent. Yeast-locked *C. albicans* could still trigger a small LDH release. We reanalysed the data in Fig.8e as suggested. Indeed, KCl (50 mM) also decreased LDH release in macrophages stimulated with the yeast-locked strain (# $p < 0.05$ vs. untreated macrophages).
- d) As suggested, LDH release from untreated macrophages in the presence or absence of KCl (Fig.8e) and from untreated WT and *Gsdmd*^{-/-} macrophages (Fig.6d) are included in the revised manuscript.

Kasper et al. recently shown that candidalysin induces macrophage cell death through an inflammasome and K⁺-independent manner (PMID: 30323213). However, the authors show that candidalysin participates in pyroptosis (Fig.5) and candidalysin-dependent cell death is also partially K⁺ efflux-dependent (Fig.6E-F). How can the authors reconcile their data with the previous observations? Additional discussion is required and differences and similarities between studies need highlighting to make the message clearer.

- a) Our results are largely consistent with what were reported by Kasper et al.:
 - Candidalysin can induce the NLRP3 inflammasome and caspase-1 activation.
 - Candidalysin activates the inflammasome via K⁺ efflux.
 - Although *C. albicans*-induced cell death could not be completely inhibited by disrupting NLRP3 inflammasome, it was clearly alleviated in NLRP3 inflammasome-deficient macrophages (Fig.10g in PMID: 30323213). These results are similar to what we observed. It is noteworthy that the cell death induced by exogenous synthetic candidalysin might be different (1-fold increase in hMDMs, Fig.10a in PMID: 30323213) from what was induced by WT *C. albicans*.
- b) Previous reports from several other labs also demonstrated that NLRP3-mediated pyroptosis contributes to *C. albicans*-induced damage of mBMDMs (Uwamahoro et al. 2014; and Wellington et al., 2014). *C. albicans*-induced macrophage lysis is dependent on caspase-1, ASC, and NLRP3 (Wellington et al., 2014).
- c) Based on these results, we conclude that candidalysin can trigger both caspase-1-dependent and -independent cell death. The "candidalysin-induced macrophage cell death" described Kasper et al. is also partially mediated by the "GSDMD-dependent pyroptosis" elucidated in our study. We further discuss and clarify this point in the revised manuscript (Page 19).

The authors show that IL-1 β processing and GSDMD cleavage are dependent on caspase-1/11 using caspase-1/11^{-/-} mice and the caspase-1 inhibitor VX765. However, they do not show caspase-1/11 expression and activation (for instance, in WT vs. KO macrophages). Although *gsdmd*^{-/-} macrophages show an increase in the intracellular level of mature IL-1 β , suggesting that IL-1 β cleavage still occurs, the detection of caspase-1/11 activity or activation would be helpful in both Figure 1 and 2. Also, IL-1 β release was impaired in caspase-1/11^{-/-}, *gsdmd*^{-/-} macrophages

or when the VX765 chemical inhibitors was used. The specificity or the global effect of the treatment should be addressed, for instance by looking at other cytokines.

We conducted these experiments as suggested:

- a) We confirmed that GSDMD disruption does not affect the cleavage and activation of caspase 1 (Fig.1a and Page 6)
- b) The Caspase-1/11^{-/-} mice and the caspase-1 inhibitor VX765 are commonly used, and the resulting caspase-1 inhibition has been demonstrated by multiple labs including my own lab. We added the related results in Fig.S1a and Fig.S6a in the revised manuscript.
- c) As suggested, we measured the amount of TNF α and IL6 secreted by stimulated macrophages and confirmed that the inhibitory effect elicited by caspase-1/11 inhibition (Fig.S6c and Page 12) or GSDMD disruption (Fig.S3c and Page 9) is specific for IL-1 β secretion.

The authors show that KCl treatment (Fig.6) (Fig.8 in the revised manuscript) inhibits GSDMD cleavage and cell death. Does KCl inhibit caspase-1 activation or IL-1 β processing and release? Is the effect specific for IL-1 β ? More data are required to interpret the findings properly.

- The role of K⁺ efflux in inflammation/caspase-1 activation is well-documented. We conducted the requested experiments and integrated them in the revised manuscript:

- a) KCl inhibited caspase-1 activation (Fig.S7a-b and Page 12).
- b) KCl inhibited IL-1 β processing and release (Fig.S7c and Page 12).
- c) KCl did not suppress TNF α or IL6 secretion (Fig.S7c and Page 12).

Dynamics (time course data) are required. Previous, initial studies have simplified the message that inflammasome activation = pyroptosis. More recent work including this submission indicates that this message is clearly wrong. The relationship between inflammasome activation, pyroptosis and GSDMD activation in Candida infection is complex. It appears that inflammasome activation can be independent of pyroptosis but that a key event of pyroptosis is GSDMD activation. There is also a combination of two pore forming events: GSDMD and candidalysin. These may act separately or together. However, all the above depends on the time point of analysis (as demonstrated by the Hube and Traven groups) and what is knocked out. The dynamics studies show that there is a defined order of inflammasome/pyroptosis/cell death during C.alb-macrophage interactions (early/late). Thus, undertaking time course assays are important in data interpretation. Note: two publications from the Traven group have not been cited or discussed – see below.

- a) We conducted some of the "time course assays" as suggested (Fig.2d, Fig.3c-f, Fig.4e-f, Fig.6a-c, Fig.8a-c and g-h, Fig.S1g, Fig.S8e-g). However, as acknowledged by the reviewer, the relationship between inflammasome activation, candidalysin, pyroptosis, and GSDMD activation in Candida infection is complex. We tried not to overinterpret these results. In the discussion, we focused on the main findings of this study **1**) the unexpected role of GSDMD in anti-candida host defense and **2**) the interactions between GSDMD and the known factors/pathways/mechanisms (Pages 20-21). We also discussed the potential "defined order of inflammasome/pyroptosis/cell death" as suggested (Page 21-22).
- b) We defined *C. albicans*-induced lytic cell death as "macrophage lysis". We adopted this term from Wellington et al (Wellington et al., 2014). When inflammasome and pyroptosis axis is blocked, *C. albicans*-induced macrophage lysis can not be completely inhibited, suggesting that *C. albicans* can escape from macrophages via both inflammasome-dependent and -independent mechanisms. Inflammasome-independent macrophage lysis may occur at later time points and/or higher organism burdens.
- c) We are familiar with the seminal work reported by the Traven lab. Their two recent publications on the role of metabolic interactions between host and pathogen in modulating Candida-induced

inflammasome activation and macrophage viability have now been cited (Tucey et al., 2018; Tucey et al., 2020) and discussed (Page 21-22).

Conceptual. A schematic diagram is also required. Regarding this, the authors should consider that the lack of pyroptosis induction via GSDMD may relate to the early phase escape route by *C. albicans* (see Kasper et al). The two later phase escape routes may be mediated by candidalysin and physical forces of hyphae (associated with metabolism re: Traven group data). Thus, if you inhibit/KO GSDMD, you may remove one escape route, while at the same time the macrophage can better deal with candidalysin-mediated damage, while protective IL-1b driven inflammation is still induced by candidalysin. Note: candidalysin can induce the inflammasome and pyroptosis-independent cell lysis, and pyroptosis can also be inflammasome-independent (Kasper et al and others). Also note that if you KO the pyroptosis axis by knocking down the inflammasome, you will still see lysis – this means that lysis induced by candidalysin will happen independent of pyroptosis. The Traven lab stressed that this depends on the time point: early phase = pyroptosis-mediated cell death and inflammasome activation; later phase = candidalysin-mediated cell death and inflammasome activation; very late phase = escape by physical forces/cell death and cell death by glucose consumption. Thus, as mentioned, this is complex, and the authors need to discuss their findings more fully and in context regarding these different processes. This is not possible unless they undertake time course studies (see dynamics issue above).

- We appreciate all these suggestions and modified Fig.S10 (a schematic diagram) accordingly. The model and the related results from other labs were further discussed on Pages 20-22. We focused on the main findings of this study - 1) the unexpected role of GSDMD in anti-candida host defense and 2) the interactions between GSDMD and the known factors/pathways/mechanisms.

Although the authors show that NSA treatment alleviates *Candida* infection and therefore the inflammasome response should not be dramatically impaired, a recent publication (PMID: 31209100, Rashidi et al, - also not cited in this submission) has demonstrated that NSA pre-treatment blocks inflammasome priming and caspase-1 activation independent of GSDMD. The Rashidi findings conflict with the original paper (Rathkey et al. 2018), which is the reference cited in this ms as the main justification for using NSA. Perhaps, the authors should address this point and verify whether NSA is affecting any priming and/or activation event in their system following *Candida* infection. More controls required.

- a) We are familiar with the report by Rashidi et al. and added a discussion in the revised manuscript (Page 13). In nigericin-stimulated LPS-primed BMDM cells, NSA does not inhibit other innate immune pathways. Cleavage of GSDMD occurred normally in NSA-treated BMDMs, indicating that NSA does not inhibit caspase-1 under this condition (Fig.5A in Rathkey et al., 2018). Rashidi et al reported that, in monosodium urate (MSU) crystal-stimulated LPS-primed BMDMs, NSA may also inhibit inflammasomes upstream of GSDMD, thereby preventing pyroptosis independent of GSDMD targeting (Rashidi et al., 2019).
- b) NSA directly inhibits GSDMD as reported by Rathkey et al. In this study, we used NSA to demonstrate that pharmacological inhibition of GSDMD is an effective therapeutic strategy for treating candidiasis (although the inhibitor may not be completely specific). Currently, there is no absolutely GSDMD-specific inhibitor.
- c) We added a control as suggested. As observed in *Gsdmd*^{-/-} mice, NSA-treated mice produced less IL-1β after *C. albicans* infection (Fig.S9g). Similarly, this modest IL-1β production appeared to be sufficient for driving anti-fungal immunity (Fig.9a-f). NSA did not affect priming (NLRP3 expression) or activation (caspase-1 cleavage) event (Fig.S8a-b). The effect on IL-1β production was specific; *C. albicans*-induced production of TNFα and IL6 was unaltered (Fig.S8c-d). These results were discussed on Pages 13-14 in the revised manuscript.

The authors used opsonised *C. albicans* for their experiments. This is different to other publications since opsonised and unopsonised *C. albicans* can give different results due to the role of complement in fungal immunity (see multiple publications by the Zipfel lab). The authors should undertake multiple experiments with unopsonised *C. albicans* to confirm they obtain the same data as with opsonised *C. albicans* to address the role of complement. A better discussion of this is also required.

- a) In this study, we focused on the candida cells engulfed by macrophages. Compared to unopsonized *C. albicans*, opsonized *C. albicans* were engulfed by the macrophages more efficiently, which is consistent with the significance of opsonization in phagocytosis (Kagaya and Fukazawa, 1981; Luo et al., 2013; Marodi et al., 1991; Pereira and Hosking, 1984; Wellington et al., 2003). We added this result in the manuscript (Fig.S3a-b). Thus, to facilitate phagocytosis and most importantly to mimic the serum-containing physiological condition, we used opsonised *C. albicans* in our experiments. This discussion was included in the revised manuscript on Page 8.
- b) We appreciate the studies conducted by the Zipfel lab and agree that complements play critical roles in host defense against Candida infection. However, the focus of current study is GSDMD. We feel the experiments aiming to "address the role of complement" are outside the scope of this study.

2. Mechanism and novelty

Most of the non-GSDMD data has previously been published (for *C. albicans* and candidalysin). Regarding the GSDMD data, the authors need to discuss/determine how GSDMD fits functionally and mechanistically in the recognised inflammasome/pyroptosis processes (see above).

Although K⁺ efflux has been demonstrated as an important event for GSDMD activation, the manuscript lacks depth with respect to possible mechanisms of inflammasome/Caspase-1/GSDMD activation in response to *C. albicans*. As the mechanisms by which *C. albicans* regulate inflammasome and GSDMD activation in their model is not described, the manuscript will greatly benefit from at least discussing how this might happen (see Point 1 comments).

- a) As correctly pointed by this reviewer, the key finding of our study is the unexpected effect of GSDMD disruption. The role of K⁺ efflux, inflammasome, and Caspase-1 in candida infection has been investigated in previous studies. In the revised manuscript, we further clarified the significance of our study and emphasized the role of GSDMD in *C. albicans* host defense (Pages 17-18).
- b) As suggested, we included a paragraph to discuss the complex mechanisms by which *C. albicans* regulate inflammasome and GSDMD activation in macrophages (Pages 20-21) and also included a schematic diagram in the revised manuscript (Fig.S10).
- c) We further discussed the role of K⁺ efflux in inflammasome/Caspase-1 activation. Studies have shown that the decrease in intracellular K⁺, which is likely mediated by the TWIK2 potassium efflux channel (Di et al., 2018), is an essential trigger for NLRP3 activation induced by ATP and other DAMPs (Franchi et al., 2007; Muñoz-Planillo et al., 2013; Pétrilli et al., 2007) (Page 12).

In vitro stimulation of macrophages with the GSDMD inhibitor NSA specifically inhibits IL-1b release and pyroptosis following LPS+nigericin stimulation. Does NSA inhibit macrophage cell death and Candida escape from macrophages? What about the inflammatory response following Candida stimulation in the presence of NSA (does NSA specifically control IL-1b? What about other inflammatory cytokines?). Also, the authors show that NSA contributes to increased resistance to Candida infection in vivo. A discussion on this finding is missing and the overall conclusion based on the use of NSA lacks a mechanism. This needs to be addressed.

- a) We further discussed the effect of NSA on inflammasome and caspase 1 in the revised manuscript (Page 13). In nigericin-stimulated LPS-primed BMDM cells, NSA does not inhibit other innate immune pathways. Cleavage of GSDMD occurred normally in NSA-treated BMDMs, indicating that NSA does not inhibit caspase-1 under this condition (Fig.5A in Rathkey et al., 2018).
- b) We conducted the requested experiments and showed that NSA could indeed inhibit macrophage cell death and Candida escape from macrophages (Fig.S8e-g). NSA treatment led to reduced production of IL-1 β , but not other cytokines (Fig.S8d).
- c) We show that NSA contributes to increased resistance to Candida infection *in vivo*. A discussion on this finding was added in the revised manuscript (Pages 13-14). It is well established that NSA can efficiently inhibit GSDMD-mediated pyroptosis. Thus, our results confirmed that inhibition of

GSDMD can induce resistance to *Candida* infection. As observed in *Gsdmd*^{-/-} mice, NSA-treated mice produced less IL-1 β after *C. albicans* infection (Fig.9g). Similarly, this modest IL-1 β production appeared to be sufficient for initiating anti-fungal immunity.

Work is with GSDMD. What about the role of other Gasdermins? Discussion.

- As suggested, we discussed the role of other Gasdermins in the revised manuscript (Page 18).

The authors use a full *gsdmd* KO mouse. Therefore, apart from macrophages, other cells (immune or otherwise) may well be important for the readouts observed in the in vivo work, given that most hematopoietic and non-hematopoietic cells tested have a role in *C. albicans* infection in vivo. Discussion.

- As suggested, we discussed this point in the revised manuscript (Page 18).

Authors show that significant macrophage death still occurred in the absence of GSDMD and conclude that *C. albicans*-induced macrophage death could be mediated by both GSDMD-dependent pyroptosis and GSDMD-independent mechanisms. OK, this is new, but no real mechanism is provided.

Candidalysin disruption in *C. albicans* did not completely abolish IL-1b release from BMDMs, indicating the presence of candidalysin independent IL-1b production. Yes, but no real mechanism is provided.

They indicate that GSDMD disruption paradoxically improved host anti-*C. albicans* responses, suggesting that *C. albicans* may have hijacked this defense mechanism to improve its survival in infected hosts. OK, but no mechanism, rather speculation.

- (Also see above) We agree. As pointed out by this reviewer, the relationship between inflammasome activation, candidalysin, pyroptosis, and GSDMD activation in *Candida* infection is complex. Several labs have made major contributions to the growth of understanding of the underlying mechanisms. It took many years and tremendous efforts to accomplish this. Here we do not intend to reveal all the remaining mechanisms and/or to establish a unified model. The major discovery of this study is that GSDMD disruption protects against *C. albicans*-induced sepsis, contrary to observations in inflammasome-deficient (e.g., *Casp1/11*^{-/-}) mice. We further emphasized the significance of this finding in the revised manuscript (Pages 17-18). Uncovering the GSDMD- and/or candidalysin-independent mechanisms will extend beyond the current study.
- As suggested by the reviewer, we added a paragraph to further discuss these potential mechanisms (Pages 20-21) and also included a schematic diagram in the revised manuscript (Fig.S10). After discussing with the co-authors and colleagues, we all feel that we should focus on what we did and avoid overinterpreting our results. As correctly pointed out by this reviewer, the main finding of this study is the unexpected role of GSDMD in anti-candida host defense. The role of inflammasome, macrophage lysis, candidalysin, and the "two-phased model" have all been previously reported. Thus, instead of trying to establish a unified model, we focused on the interactions between GSDMD and the known factors/pathways/mechanisms (Pages 20-21 and Fig.S10).

The main novelty of the ms is the notion that GSDMD facilitates escape from macrophages. However, *C. albicans* escape has previously been quantified (Kasper et al) and no difference in piercing or hyphal length was observed in human primary macrophages. All data in the current submission is based on mouse macrophages (where differences in hyphal length was observed), but mouse and human macrophages behave differently. Conclusions/findings should be supported using human macrophages. Furthermore, the authors indicate that candidalysin mediates escape via GSDMD activation and pyroptosis, but that candidalysin also does this via direct membrane lysis. Are the authors indicating these processes are connected or independent? More discussion required.

- a) Thanks for the comments. We conducted the suggested experiments using human monocyte-derived macrophages (hMDMs). Both WT and candidalysin-deficient *C. albicans* were used. The human GSDMD and caspase-1 were pharmacologically inhibited by NSA (Fig.10) and VX765 (Fig.S9), respectively. We confirmed that GSDMD and candidalysin also played a critical role in clearance of *C. albicans* by human macrophages (Pages 14-15).

- b) Kasper et al previously showed that "there was no difference (between WT and candidalysin-deficient *C. albicans*) in piercing or hyphal length in human primary macrophages" (Fig.2f in Kasper et al PMID: 30323213). The purpose of these experiments was to demonstrate that candidalysin did not affect the hyphal growth. Thus, what they measured was the average hyphal length in the cells (10-20 μm) and they measured it at the relatively early stage of infection (3 h p.i.) when hyphae were still growing. We were not able to find other details (e.g the number of cells/data point and the number of hyphal/cell examined) regarding these experiments. The conclusion of this experiment is that the decreased inflammasome activation in the absence of ECE1 was not due to reduced uptake of fungal cells or hyphal defects. In contrast, the purpose of our experiments is to assess the constraint of *Candida* growth in intact macrophages. Thus, we measured the maximum length of intracellular hyphae in WT and *Gsdmd*^{-/-} macrophages (>25 μm), and we measured it at 6 h after the infection. Under this condition, due to the impaired escape, candidalysin-deficient *C. albicans* grew longer hyphae (>50 μm) inside macrophages (Fig.S4b). These hyphae from candidalysin-deficient *C. albicans* could push and stretch the macrophage plasma membranes, and yet the infected macrophages were PI negative, suggesting that candidalysin was indeed critical for facilitating disruption of host plasma membrane integrity (Fig.S4a). In this experiment, at least 100 cells were assessed for each sample. We also counted the number of escaped hyphae per macrophage (Fig.3d). After 6 h, there were too many hyphae to be counted accurately.
- c) Our results suggest that candidalysin can trigger both GSDMD-dependent and -independent macrophage cell death. Candidalysin can induce K⁺ efflux through plasma membrane perforation, which in turn leads to canonical inflammasome activation, GSDMD cleavage, and macrophage pyroptosis. As a pore-forming toxin, candidalysin can also induce macrophage lysis in the absence of GSDMD. These are two parallel and independent mechanisms. We discussed this point in the revised manuscript (Page 19).

3. References and lack of discussion

One of the authors (Kanneganti) has previously published data on GSDMD cleavage by *C. albicans* (JBC PMID: 33109609). However, this publication is not cited or discussed in the current submission, which reduces novelty somewhat and raises the question of why wasn't this publication cited? Irrespective, in Fig.2 of the JBC paper, it appears that activation of Casp-1 (p20) is slightly reduced in *gsdmd* KO mice (however, no quantification is available). Therefore, in the current submission, the authors should also show Caspase-1 activation where relevant as controls (also relevant for Point 1 above).

- Sorry for the confusion. Dr. Thirumala-Devi Kanneganti, the corresponding author of the above-mentioned paper (JBC PMID: 33109609), is a professor at the St. Jude Children's Research Hospital. Apurva Kanneganti, a Harvard undergraduate student and a co-author of this paper, is a different person. The paper from the Kanneganti lab had not been published when we prepared our manuscript. We now cited this paper in the revised manuscript (Page 17).
- (Also see above) As suggested, we assessed Caspase-1 activation in the in the revised manuscript.

The authors show that KCl supplementation reduces GSDMD activation and macrophage cell death (Fig.6), but inhibition of macrophage pyroptosis following KCl supplementation has been already shown following *Candida* infection (PMID: 30131363). Importantly, treatment of KCl also reduces ASC speck formation and the number of CFUs, thus suggesting fungi can use pyroptosis to evade killing by macrophages. However, the authors do not cite or discuss this paper. Needs including with proper discussion.

- This is a great point! We cited the paper (O'Meara et al, 2018) and discussed the related results in the revised manuscript (Page 12).

Important publications and concepts (dynamics) from the Traven Lab have been omitted (see above). "Metabolic competition between host and pathogen dictates inflammasome responses to fungal infection" (PMID: 32750090),

and “Glucose homeostasis is important for immune cell viability during candida challenge and host survival of systemic fungal infection” (PMID: 29719235). Should be included in context and discussed.

- (Also see above) We cited and discussed these studies in the revised manuscript.

In the mouse systemic candidiasis model, the authors found that mice infected with the *ece1* deficient *C. albicans* displayed higher survival rates and decreased fungal burden compared with WT *C. albicans*. While the Swidergall reference is cited, the data are in contract with that ms, which found increased fungal burden with the *ece1* deficient *C. albicans*. This is not discussed. How do the authors explain this?

- This is a great observation. Thanks for pointing this out. Both my lab and Naglik lab (Swidergall et al., 2019) reported that candidalysin promotes mortality in murine models of systemic fungal infection. However, Naglik lab focused on neutrophil recruitment. Their data indicated that candidalysin was required for neutrophil recruitment; thus when *ece1* deficient *C. albicans* (2×10^5) were used to infect mice, fungal burden of the kidney at 1 day increased significantly due to reduced neutrophil accumulation. This effect was diminished 4 days post-infection. Our study focuses on the escape of *C. albicans* from macrophages, and we inoculated mice with more *Candida albicans* cells (1×10^6) and assessed fungal burden 2 days post-infection. In this setup, mice infected with the *ece1* deficient *C. albicans* displayed decreased fungal burden compared with WT *C. albicans*.
- Thus, the discrepancy is likely caused by different experimental conditions in their compared with our experiments. Swidengall et al sacrificed the mice at 1 and 4 days, whereas we sacrificed them at 2 days. Our inoculum was 1×10^6 cells, theirs was 2×10^5 cells for immunocompetent mice and 5×10^4 for neutropenic mice. Additionally, experiments of Swidergall et al and ours were performed using different strains. We used a strain received from the lab who first cloned and deleted ECE1 (Birse et al., 1993). Swidergall et al used different strains, that were differently constructed, first published in Moyes et al paper (Moyes et al., 2016).
- Of note, in an oral candidiasis model (Moyes et al., 2016)(another paper published by the Naglik lab), the fungal burden was also lower in *ece* Δ/Δ infection compared with wild type, in line with our own findings. In the Moyes et al study, it was revealed that "tongue tissue from *ece1* Δ/Δ -infected animals (n = 17/20) showed no invasive fungi and no inflammatory infiltrates or damage (Fig.1g)" and "very low numbers of *ece1* Δ/Δ cells in only 3/20 mice (Extended Data Fig. 2a) were detected". This discrepancy was not explained or mentioned in the 2019 Swidengall et al paper (Swidergall et al., 2019).
- In summary, Swidergall et al. saw higher organ fungal burdens, while Moyes et al and we saw concordant results of lower fungal burdens and decreased virulence in *ece* Δ/Δ infection compared with wild type. We discussed these results in the revised manuscript (Page 21).

***C. albicans*-dependent activation of the inflammasome/pyroptosis is complex. The authors should consider recent reviews on the topic and determine how their work fits into this (e.g.: PMID: 33293167; PMID: 24967821; PMID: 33119702), none of which have been cited.**

- We are familiar with these reviews and now cited and discussed them in the revised manuscript (Page 19).

4. Minor

The title is rather misleading and needs modifying, as they find that cell death is promoted by both inflammasome-dependent and -independent mechanisms. Also, significant macrophage death occurred in the absence of GSDMD. Furthermore, they find candidalysin-dependent and independent mechanisms, suggesting GSDMD is not the factor ‘dictating’ escape of *C. albicans*. (**Title**)

Abstract. Candidalysin is not ON hyphae, it’s secreted. Change to “expressed by hyphae”. (**Abstract**)

The ms states: “For in vitro experiments, the GSDMD antagonist necrosulfonamide (NSA) was dissolved in dimethyl sulfoxide (DMSO), with DMSO alone used as control (lanes 554-555)”. Did the authors mean in vivo? Consider amending. (Pages 25 and 27)

- Thanks for pointing these out. We modified the manuscript and made the suggested changes.

The authors suggest that pharmacologic inhibition of GSDMD may offer a new therapeutic strategy for *C. albicans* infection. This is way too early to state. Given the lack of mechanism and the significant experimentation the authors clearly need to undertake, all such comments should to be removed throughout the ms.

- We modified the related sentences and emphasized that it is a "potential" new therapeutic strategy (Abstract, Page 5, Page 16).

5. Direct comment to sentences of the paper

"Mice lacking key inflammasome components are hypersusceptible to *Candida* infection" (Page 4-5)

That depends on the type of *Candida* infection. Inflammasome activation can be beneficial in candidiasis, and detrimental i.e. during VVC (immunopathology). The authors need to be more precise. Furthermore, every candidemia patient is different as will be the 'mechanisms' of immunity, thus personalized medicine is key.

"Our results have shown that inflammasome-mediated GSDMD activation dictated *C. albicans* escape from macrophages" (Page 13)

Disagree, and it contradicts some of the data and many statements in the ms. This conclusion should be altered as does the title.

- Again, thanks for all these great comments and suggestions. We have revised the manuscript accordingly.

"IL-1b release was significantly reduced when macrophages were treated with candidalysin-deficient *C. albicans*" and "However, candidalysin disruption in *C. albicans* did not completely abolish IL-1b release from BMDMs, indicating the presence of candidalysin-independent IL-1b production"

That should be pyroptosis. This is likely to be mediated by any other factors of the fungus that provide both signals to the inflammasome.

- We further discussed the potential mechanisms (Pages 20-21) and also included a schematic diagram in the revised manuscript (Fig.S10). Based on our results (Fig 5c), both GSDMD-dependent (pyroptosis) and -dependent mechanisms contribute to candidalysin-independent IL-1 β production.

Many of the key findings/statements in the ms have been shown before, predominantly in the Kasper et al paper. This is not always recognized by the authors. These include:

"Infection with candidalysin-deficient *C. albicans* also induced significantly less LDH release, again suggesting that candidalysin partially contributed to macrophage death"

"Similarly, *C. albicans* escape from macrophages and *C. albicans*-induced macrophage death were mediated by both candidalysin-independent and -dependent mechanisms ..."

"As a pore-forming toxin, candidalysin may facilitate *C. albicans*-induced rupture of the plasma membrane and macrophage lytic death, which enable *C. albicans* to escape from host cells "

"The hyphae from candidalysin-deficient *C. albicans* could push and stretch the macrophage plasma membranes, and yet these infected macrophages were PI negative, suggesting that candidalysin was indeed critical for facilitating disruption of host plasma membrane integrity"

"It is noteworthy that candidalysin-deficient *C. albicans* could eventually break the plasma membrane, contributing to candidalysin-independent *C. albicans* escape. This process may be mediated by mechanical piercing or other hyphal proteins, ..."

"*C. albicans*-induced macrophage death likely mediated by both inflammasome-dependent pyroptosis and inflammasome-independent mechanisms"

"...inflammasome-independent macrophage death was also facilitated by candidalysin and hyphae formation"

- We agree. We acknowledged and further discussed the work reported in the Kasper et al paper (Pages 10, 19, and 20).

- As discussed above, the focus of the current study is GSDMD. To show the role of candidalysin in GSDMD-mediated processes, we needed to repeat some of their experiments and use them as controls. We are not claiming these are new discoveries.

Regarding "This process was also suppressed in *Gsdmd* -/- BMDMs, demonstrating that GSDMD was important for both candidalysin-independent and -dependent IL-1b release (Fig.5D). " There maybe candidalysin-mediated, GSDMD-independent IL-1b release due to candidalysin pore formation, but if you have both, IL-1 β release is much

higher likely because the other stimuli provided by *C. albicans* are poor inducers of inflammasome activation while candidalysin will cause rapid K efflux. Note: escape due to physical forces are independent of GSDMD.

" however, macrophages infected with candidalysin-deficient mutants still released significant amounts of IL-1b, suggesting that other fungal components (e.g., b-glucans) could activate inflammasomes and process IL-1b during *C. albicans* infection."

See above: the time point is important. Early events = pyroptosis = candidalysin independent.

- (Also see above) The relationship between inflammasome activation, candidalysin, pyroptosis, and GSDMD activation in *Candida* infection is complex. We added a paragraph to further discuss these potential mechanisms (Pages 20-21) and also included a schematic diagram in the revised manuscript (Fig.S10).

Reviewer 2

This is a great manuscript from Kambara and colleagues showing a novel link between GSDMD activation and the intracellular escape of *Candida albicans*. The data are very clear and convincing, the observation novel and important and I have almost no comments. However, my only reservation is the *in vivo* relevance of this phenotype. The authors show a clear *in vivo* phenotype in the GSDMD mice and then go onto to reveal the mechanism using BMDMS. *In vivo* support for their hypothesis is given at the end with the GSDMD inhibitors (but essentially this is reproducing the the KO phenotype). What is not clear is whether the mechanism derived from the *in vitro* studies explains the *in vivo* phenotype. For example, there is evidence from several models (including zebrafish..see work from Rob Wheeler) that *Candida* hyphenation inside macrophages *in vivo* is a rare event. Also the impact of tissue matrix on macrophages responses etc. (also shown in *in vitro* work using 2d vs 3d systems). Thus I would like to see some formal *in vivo* evidence supporting the hyphae, candidalysin, GSDMD model. Also, the survival of GSDMD deficient mice infected with the *ece* mutant should be shown (surprising this was not, as this would also help validate their system).

- We are pleased with the complimentary comment of the reviewer "This is a great manuscript ...The data are very clear and convincing, the observation novel and important and I have almost no comments".

- As suggested by this reviewer, we conducted several additional experiments to provide more *in vivo* evidence supporting our hyphae-candidalysin-GSDMD model.

- a) We examined candida hyphaenation inside macrophages *in vivo* in WT and GSDMD KO mice challenged with the either WT or candidalysin-deficient (*ece1* Δ/Δ) *C. albicans*. These data are included in Fig.S5 in the revised manuscript. The results are described and discussed on Page 11. The related references were also discussed.
- b) As suggested, we measured the survival, body weight change, and clinical score of GSDMD-deficient mice infected with *ece1* Δ/Δ *C. albicans*. These data are included in Fig.S4c-i. The related results are described and discussed on Page 11.

Reviewer 3

This study reports that unlike caspase1/11 loss, deletion of the inflammatory caspase substrate GSDMD protects against *Candida*-induced sepsis and this likely occurs as a consequence of reduced host cell death facilitating increased killing of intracellular *Candida* and subsequent decreased fungal dissemination. This finding is unexpected as deletion of caspase-1/11, which cleave GSDMD, has the opposite effect and increases animal susceptibility to *Candida* infection. It is also of significant pharmaceutical interest, as it points towards GSDMD as being a bona fide therapeutic target in this condition. The experiments are well performed using relevant genetically targeted mice or GSDMD inhibitor (NSA), the results clear, and the conclusions supported by the data presented.

- We are pleased with the conclusion of this reviewer that "The experiments are well performed using relevant genetically targeted mice or GSDMD inhibitor (NSA), the results clear, and the conclusions supported by the data presented". The reviewer only has "a few relatively minor questions for the authors":

1. Figure 1 and Figure 7 (Figure 9 in the revised manuscript). The number of times the in vivo experiments were independently repeated to confirm the results presented should be clearly noted in the figure legend for readers.

- Thanks for pointing this out. We clarified these details in the figure and figure legends (Fig.1 and Fig.9).

2. Does the increased killing capacity of GSDMD deficient macrophages allow them to generate more inflammatory cytokines (e.g. TNF, IL-6), and if so, could this also contribute to anti-fungal responses?

- We measured the levels of TNF α and IL-6 produced by infected WT and GSDMD deficient macrophages as suggested. *Candida* infection could directly trigger TNF α and IL-6 production in macrophages. However, WT and GSDMD deficient macrophages generated the same amount of TNF α and IL-6 after *Candida* infection, supporting the notion that expression and secretion of these two inflammatory cytokines are GSDMD-independent. Thus, the elevated anti-fungal response observed in the GSDMD-deficiency mice was unlikely mediated by overall upregulation of inflammatory cytokine production. These data are included in Fig.S3c in the revised manuscript. The results are described and discussed on Page 9.

3. The authors suggest that caspase-1 mediated IL-1beta activation and release, which still occurs in GSDMD deficient cells, explains why caspase-1/11 and GSDMD loss have the opposite phenotype, despite both being required for pyroptosis. Can the authors prove this by, for example, using neutralising IL-1beta antibodies in infected GSDMD KO mice to reverse the protective phenotype?

- This is a great suggestion! We performed this experiment using anakinra, a commonly used interleukin 1 receptor antagonist. In the presence of anakinra, the protective effect elicited by GSDMD disruption was completely inhibited. Anakinra-treated WT and GSDMD-deficiency mice displayed worse clinical score and reduced survival after *C. albicans* infection compared to untreated GSDMD-deficient mice (Fig.2h-j, Fig.S2, and Page 7).

4. Although NSA targets GSDMD it can also act independent of GSDMD to block pyroptotic killing (e.g. DOI: 10.4049/jimmunol.1900228), and the authors should acknowledge this.

- Thanks for pointing this out. We acknowledged and discussed this point in the revised manuscript (Page 13).

REFERENCES

- Birse, C.E., Irwin, M.Y., Fonzi, W.A., and Sypherd, P.S. (1993). Cloning and characterization of ECE1, a gene expressed in association with cell elongation of the dimorphic pathogen *Candida albicans*. *Infect Immun* 61, 3648-3655.
- Di, A., Xiong, S., Ye, Z., Malireddi, R.K.S., Kometani, S., Zhong, M., Mittal, M., Hong, Z., Kanneganti, T.D., Rehman, J., *et al.* (2018). The TWIK2 Potassium Efflux Channel in Macrophages Mediates NLRP3 Inflammasome-Induced Inflammation. *Immunity* 49, 56-65 e54.
- Franchi, L., Kanneganti, T.-D., Dubyak, G.R., and Núñez, G. (2007). Differential Requirement of P2X7 Receptor and Intracellular K⁺ for Caspase-1 Activation Induced by Intracellular and Extracellular Bacteria*. *Journal of Biological Chemistry* 282, 18810-18818.
- Kagaya, K., and Fukazawa, Y. (1981). Murine defense mechanism against *Candida albicans* infection. II. Opsonization, phagocytosis, and intracellular killing of *C. albicans*. *Microbiology and immunology* 25, 807-818.
- Luo, S., Skerka, C., Kurzai, O., and Zipfel, P.F. (2013). Complement and innate immune evasion strategies of the human pathogenic fungus *Candida albicans*. *Molecular immunology* 56, 161-169.

- Marodi, L., Korchak, H.M., and Johnston, R.B., Jr. (1991). Mechanisms of host defense against *Candida* species. I. Phagocytosis by monocytes and monocyte-derived macrophages. *J Immunol* *146*, 2783-2789.
- Moyes, D.L., Wilson, D., Richardson, J.P., Mogavero, S., Tang, S.X., Wernecke, J., Hofs, S., Gratacap, R.L., Robbins, J., Runglall, M., *et al.* (2016). Candidalysin is a fungal peptide toxin critical for mucosal infection. *Nature* *532*, 64-68.
- Muñoz-Planillo, R., Kuffa, P., Martínez-Colón, G., Smith, Brenna L., Rajendiran, Thekkelnaycke M., and Núñez, G. (2013). K⁺ Efflux Is the Common Trigger of NLRP3 Inflammasome Activation by Bacterial Toxins and Particulate Matter. *Immunity* *38*, 1142-1153.
- Pereira, H.A., and Hosking, C.S. (1984). The role of complement and antibody in opsonization and intracellular killing of *Candida albicans*. *Clin Exp Immunol* *57*, 307-314.
- Pétrilli, V., Papin, S., Dostert, C., Mayor, A., Martinon, F., and Tschopp, J. (2007). Activation of the NALP3 inflammasome is triggered by low intracellular potassium concentration. *Cell Death & Differentiation* *14*, 1583-1589.
- Rashidi, M., Simpson, D.S., Hempel, A., Frank, D., Petrie, E., Vince, A., Feltham, R., Murphy, J., Chatfield, S.M., Salvesen, G.S., *et al.* (2019). The Pyroptotic Cell Death Effector Gasdermin D Is Activated by Gout-Associated Uric Acid Crystals but Is Dispensable for Cell Death and IL-1 β Release. *The Journal of Immunology*, ji1900228.
- Swidergall, M., Khalaji, M., Solis, N.V., Moyes, D.L., Drummond, R.A., Hube, B., Lionakis, M.S., Murdoch, C., Filler, S.G., and Naglik, J.R. (2019). Candidalysin Is Required for Neutrophil Recruitment and Virulence During Systemic *Candida albicans* Infection. *The Journal of infectious diseases* *220*, 1477-1488.
- Tucey, T.M., Verma, J., Harrison, P.F., Snelgrove, S.L., Lo, T.L., Scherer, A.K., Barugahare, A.A., Powell, D.R., Wheeler, R.T., Hickey, M.J., *et al.* (2018). Glucose Homeostasis Is Important for Immune Cell Viability during *Candida* Challenge and Host Survival of Systemic Fungal Infection. *Cell Metab* *27*, 988-1006.e1007.
- Tucey, T.M., Verma, J., Olivier, F.A.B., Lo, T.L., Robertson, A.A.B., Naderer, T., and Traven, A. (2020). Metabolic competition between host and pathogen dictates inflammasome responses to fungal infection. *PLOS Pathogens* *16*, e1008695.
- Wellington, M., Bliss, J.M., and Haidaris, C.G. (2003). Enhanced phagocytosis of *Candida* species mediated by opsonization with a recombinant human antibody single-chain variable fragment. *Infect Immun* *71*, 7228-7231.
- Wellington, M., Koselny, K., Sutterwala, F.S., and Krysan, D.J. (2014). *Candida albicans* triggers NLRP3-mediated pyroptosis in macrophages. *Eukaryotic cell* *13*, 329-340.

REVIEWER COMMENTS

Reviewer #1 (Remarks to the Author):

The authors have taken on board many of the comments raised and have produced an excellent piece of work that will significantly advance the field. While not all issues were fully addressed, the major issues have been addressed with significant additional discussion.

Reviewer #2 (Remarks to the Author):

I am happy with the responses provided by the authors, and have no further comments

Reviewer #3 (Remarks to the Author):

The authors have addressed all my queries. This is a very nice story which I'm confident will generate significant interest in the field. I only have one additional request for the authors;

It is stated in the response letter that "NSA did not affect priming (NLRP3 expression) or activation (caspase-1 cleavage) event (Fig.S8a-b)." This is incorrect, as the authors data (Fig. S8a) clearly show that NSA blocks caspase-1 cleavage, GSDMD cleavage to the p30 fragment and also pro-IL-1beta levels. These data are consistent with the findings of Rashidi et al demonstrating that NSA limits IL-1beta levels and also pyroptosis upstream of targeting the GSDMD pore-forming p30 fragment. If Rathkey et al were correct, then GSDMD p30 levels should not be impacted by NSA. Therefore, in the manuscript text the statement "Treatment with NSA blocked both pro-caspase-1 and GSDMD cleavage in *C. albicans*-infected macrophages without affecting Candida-induced NLRP3 expression (Fig.S8a-b)." should be re-written to note that "Treatment with NSA blocked both pro-caspase-1, pro-IL-1beta and GSDMD cleavage in *C. albicans*-infected macrophages consistent with the idea it can inhibit pyroptosis upstream of GSDMD, in addition to inhibiting GSDMD p30 itself (Fig.S8a-b)."

Point-by-point response to the reviewers' comments

RE: Manuscript number NCOMMS-21-03664A

Reviewer #3 has a minor suggestion:

The authors have addressed all my queries. This is a very nice story which I'm confident will generate significant interest in the field. I only have one additional request for the authors;

It is stated in the response letter that “NSA did not affect priming (NLRP3 expression) or activation (caspase-1 cleavage) event (Fig.S8a-b).” This is incorrect, as the authors data (Fig. S8a) clearly show that NSA blocks caspase-1 cleavage, GSDMD cleavage to the p30 fragment and also pro-IL-1beta levels. These data are consistent with the findings of Rashidi et al demonstrating that NSA limits IL-1beta levels and also pyroptosis upstream of targeting the GSDMD pore-forming p30 fragment. If Rathkey et al were correct, then GSDMD p30 levels should not be impacted by NSA. Therefore, in the manuscript text the statement “Treatment with NSA blocked both pro-caspase-1 and GSDMD cleavage in *C. albicans*-infected macrophages without affecting Candida-induced NLRP3 expression (Fig.S8a-b).” should be re-written to note that “Treatment with NSA blocked both pro-caspase-1, pro-IL-1beta and GSDMD cleavage in *C. albicans*-infected macrophages consistent with the idea it can inhibit pyroptosis upstream of GSDMD, in addition to inhibiting GSDMD p30 itself (Fig.S8a-b).”

- We modified this sentence as suggested.